geology/climatology/palaeontology

mid-Devensian, Vale of York, plant macrofossils, Trichoptera, Coleoptera, periglaciation

**Author for correspondence:**
Philip I. Buckland
e-mail: philip.buckland@umu.se

# Mid-Devensian climate and landscape in England: new data from Finningley, South Yorkshire

Philip I. Buckland[1], Mark D. Bateman[2], Ole Bennike[3], Paul C. Buckland[4], Brian M. Chase[5], Charles Frederick[6], Malcolm Greenwood[7], Julian Murton[8], Della Murton[9] and Eva Panagiotakopulu[10]

[1]Environmental Archaeology Lab, Umeå University, Umeå 901 87, Sweden
[2]Department of Geography, University of Sheffield, Sheffield S10 2TN, UK
[3]GEUS Geological Survey of Denmark and Greenland, ØsterVoldgade 10, Copenhagen 1350, Denmark
[4]Independent Researcher, 20 Den Bank Close, Sheffield S10 5PA, UK
[5]Institut des Sciences de l'Evolution-Montpellier (ISEM), Université de Montpellier, CNRS, EPHE, IRD, Bat 22, CC061, Place Eugène Bataillon, 34095 Montpellier cedex 5, France
[6]Department of Geography and the Environment, The University of Texas at Austin, Austin, TX, USA
[7]Department of Geography, Loughborough University, Leics LE11 3TU, UK
[8]Department of Geography, University of Sussex, Brighton BN1 9RH, UK
[9]Department of Zoology, University of Cambridge, Downing Street, Cambridge CB2 3EJ, UK
[10]Institute of Geography, University of Edinburgh, Drummond Street, Edinburgh EH8 9XP, UK

PIB, 0000-0002-2430-0839; MDB, 0000-0003-1756-6046;
OB, 0000-0002-5486-9946; BMC, 0000-0001-6987-1291;
CF, 0000-0002-5955-2411; JM, 0000-0002-9469-5856

While there is extensive evidence for the Late Devensian, less is known about Early and Middle Devensian (approx. 110–30 ka) climates and environments in the UK. The Greenland ice-core record suggests the UK should have endured multiple changes, but the terrestrial palaeo-record lacks sufficient detail for confirmation from sites in the British Isles. Data from deposits at Finningley, South Yorkshire, can help redress this. A channel with organic silts, dated 40 314–39 552 cal a BP, contained plant macrofossil and insect remains showing tundra with dwarf-shrub heath and bare ground. Soil moisture conditions varied from free draining to riparian, with ponds and wetter vegetated areas. The climate was probably low arctic with snow cover during the winter. Mutual climatic range (MCR), based on Coleoptera, shows the mean monthly winter temperatures of −22 to −2°C and summer ones of 8–14°C. Periglacial structures within the basal gravel deposits and beyond the

glacial limits indicate cold-climate conditions, including permafrost. A compilation of MCR reconstructions for other Middle Devensian English sites shows that marine isotope stage 3—between 59 and 28 ka—experienced substantial variation in climate consistent with the Greenland ice-core record. The exact correlation is hampered by temporal resolution, but the Finningley site stadial at approximately 40 ka may correlate with the one of the Greenland stadials 7–11.

## 1. Introduction

During the Late Quaternary, the UK was subjected to multiple cold stages, during which both periglaciation and glaciation sculpted the landscape. Recent research has started to address systematically the pattern and chronology of the British and Irish Ice Sheet (BIIS), particularly during the Last Glacial Maximum (LGM, approx. 21 ka) and Lateglacial (*ca* 13–10 ka; e.g. [1–3]). During the LGM (although not necessarily synchronously), the BIIS extended to the Scilly Isles [4] in the southwest of the UK and to Norfolk in the southeast (e.g. [5]). Less is known about the climate and environments of the UK during the Early and Middle Devensian (approx. 110–30 ka). Rasmussen *et al.* [6], based on Greenland ice-core records, showed 25 stadials and 24 interstadials between 110 ka and the start of the Holocene (approx. 11.7 ka), but the extent and duration of Devensian glacial ice in the UK prior to the LGM is less certain. While Straw [7,8] has argued for two distinct Devensian advances of ice as far south as Lincolnshire and Yorkshire (figure 1), one LGM and another at some time pre-LGM, evidence for this was not found by Evans *et al.* [10] in Yorkshire or Lincolnshire, where the two Marsh tills of Straw were reinterpreted as two oscillations within the LGM ice advance. Carr *et al.* [11] showed extensive ice across northern England and Scotland during the Ferder episode approximately 70 ka in marine isotope stage (MIS) 4, although this limit is not well constrained by dating evidence. Clark *et al.* [2] showed that by 27 ka (MIS 2), once again much of northern England and Scotland were covered by ice, suggesting that initiation must have occurred sometimes before that in MIS 3. Roberts *et al.* [5] showed ice advanced southward down the North Sea to Dogger Bank, withdrew by approximately 23.1 ka before advancing to its maximal position on the Norfolk coast approximately 22.8–21.5 ka. During the Devensian, periglaciation took place beyond the ice limits, as demonstrated by regional patterned ground and involutions dated to 60–55 ka (MIS 3), 35–31 ka (MIS 3), 22–20 ka (MIS 2) and 12–11 ka in eastern England [12]. The Devensian cold periods were interspersed with warmer interstadials ([13], table 3.4), most notably the Chelford Interstadial (MIS 5c, approx. 108–92 ka [14–16]), the Brimpton (MIS 5a, approx. 86–72 ka [14,17]) and the Upton Warren (early during MIS 3 [18]; possibly approx. 42.5–38.5 $^{14}$C ka, according to Catt *et al.* [19]). However, the chronologies, environments and climatic conditions associated with these interstadials remain not well understood. For example, aminostratigraphy suggests that the Upton Warren Interstadial significantly pre-dates MIS 3 [20]. In summary, while the Greenland ice-core record would suggest the UK landscape should have endured multiple climatic and environmental changes, at present, studies of the terrestrial palaeo-record lack sufficient detail to confirm this.

This study reports observations from one of the UK's main onshore sedimentary basins (Glacial Lake Humber), which contain Pleistocene deposits laid down in a proglacial setting, in order to elucidate further the Middle to Late Devensian terrestrial palaeo-record and landscape change in eastern England. The study takes a detailed multi-proxy approach to both sediments and preserved fossils found at a site close to, but outside of, the Devensian ice limit at Finningley, South Yorkshire. Results are evaluated in the context of other dated Middle Devensian sites for which beetle-based palaeoclimatic or palaeoenvironmental information exists.

## 2. Regional setting

East of Doncaster the Sherwood Sandstone Group dips gently beneath the Quaternary deposits of the Humberhead Levels (figure 1). The detailed mapping by Gaunt [21–23] provided the basis for a regional Quaternary stratigraphy (figure 2), with the oldest deposits being ascribed to the Anglian (MIS 12) represented by heavily cryoturbated sands and gravels found on the north–south ridge between Bawtry and Rossington (cf. [25,26]). The overlying Older River Gravel (ORG; [22,27]) forms a gently undulating surface sloping northwards from Austerfield through to Hatfield, where it meets a similar fan of gravels relating to the Don drainage ([22], fig. 41). A flora and insect fauna from within

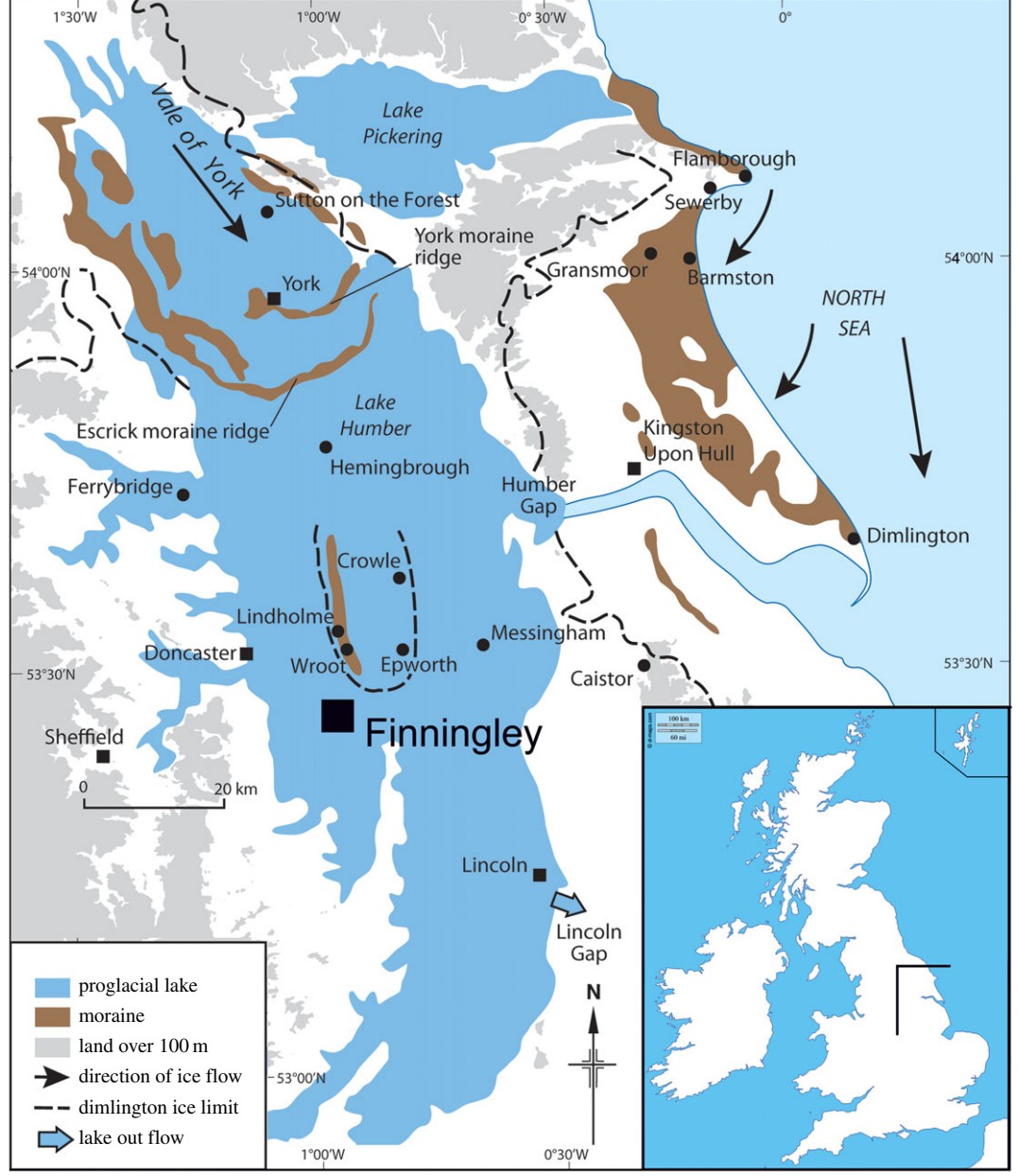

**Figure 1.** Map showing location of Finningley and LGM features in the East Midlands and East and South Yorkshire (redrawn and modified from [9]).

the ORG at Austerfield indicates an MIS 5e age [28], although Coope [29] raised the possibility of an MIS 9 age. It seems probable that the ORG may represent deposits from several glacial–interglacial cycles.

More recent general reviews of the regional Quaternary stratigraphy are provided in Bateman *et al.* [30] and Gaunt *et al.* [24], but eastwards and northwards, these deposits are overlain by those of the '25 ft Drift', littoral sands and profundal (deep-water, lake) clay–silts of proglacial Lake Humber (Hemingbrough Glaciolacustrine Formation; [31]). Lake Humber is thought to have formed in the Vale of York when North Sea ice blocked the Humber Gap at Ferriby. Gaunt [21,22] interpreted the regional stratigraphy in terms of a two-stage model, with shorelines at approximately 33 and 7 m (25 ft Drift). While the existence of a high level lake has been challenged [32–34], its transient nature, leading to limited deep-water deposition, has been stressed by Bateman *et al.* [35] and its relationship with a short-lived extension of a western tongue of the Vale of York glacier has recently been considered by Friend *et al.* [36]. Recent optically stimulated luminescence (OSL) ages from both lacustrine and regional glacial sediments [9,35,37–39] indicate an LGM age for both phases, although the sequence appears more complex than Gaunt [22] originally suggested [40,41].

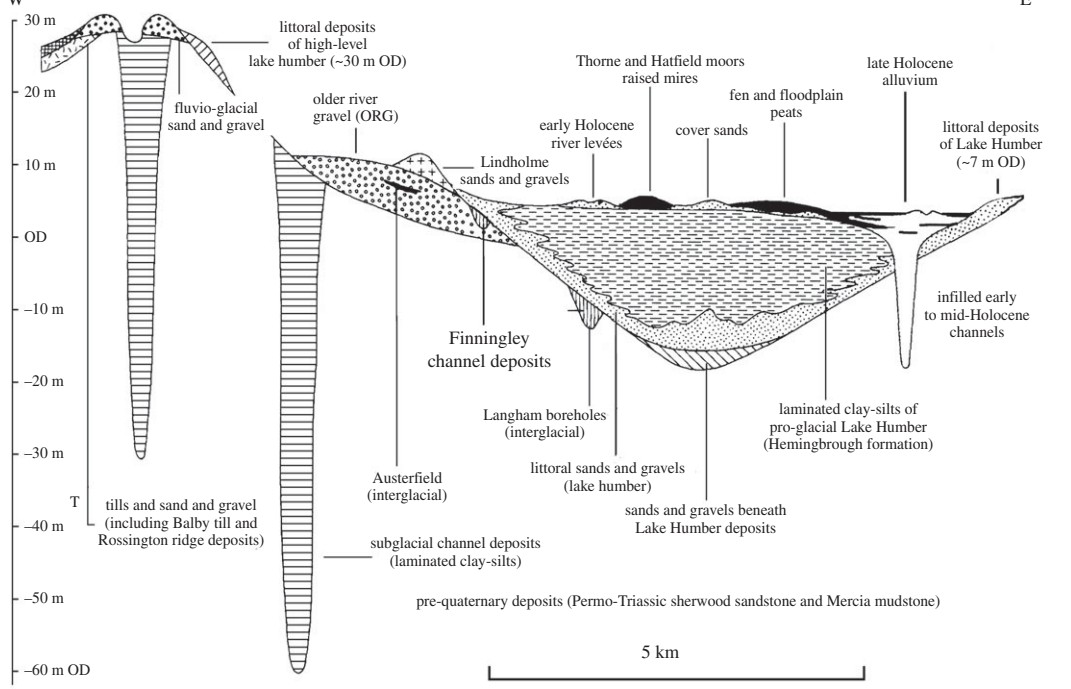

**Figure 2.** Schematic geological cross-section of Quaternary deposits in the southern part of the Vale of York projected onto a west–east line near Rossington (redrawn and modified from [24]).

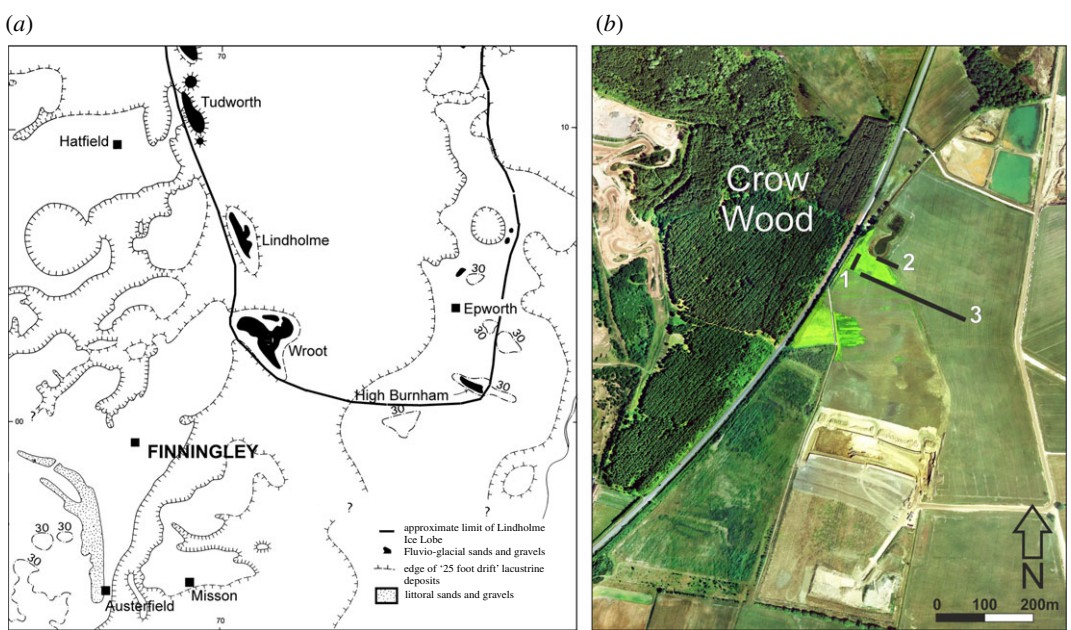

**Figure 3.** (*a*) Map of Finningley in relation to the projected shoreline of 'low-level' Lake Humber (from [22]) and the Lindholme lobe of the Vale of York glacier (modified from [36]). (*b*) Satellite image of the Finningley Quarry in December 2002 showing the location of the sections (Map data: Google Earth: Image @ The GeoInformation Group. Image NASA).

Deposits exposed in the quarry at Finningley, as it was worked as a series of west–east faces (figures 3 and 4), allowed further examination of the stratigraphy of the region. The relationship between these small outcrops, the ORG and 25 ft Drift was progressively revealed in 2001–2007 as quarrying proceeded southwards (figure 4). The site itself lies east of the Finningley to Austerfield road at Crow Wood (figure 3*a,b*; Lat. 53° 28′ 20″N; Long 00 59′ 02″ W).

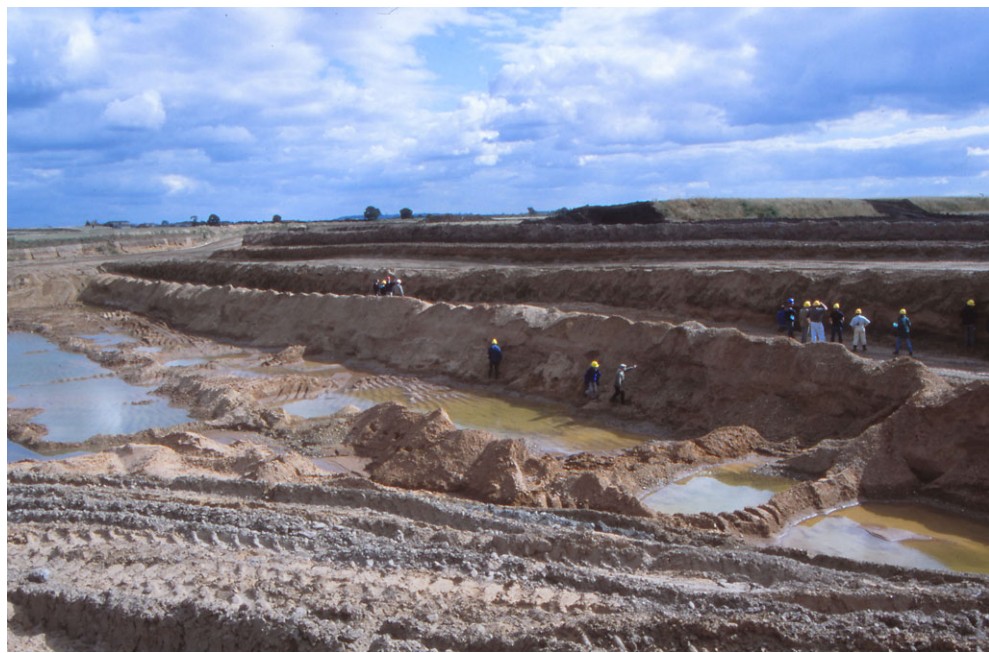

**Figure 4.** Section at the Finningley Quarry, looking southeast, September 2001. The base of the pit is in the ORG (unit 1), the working levels in Littoral Sands and Gravels (unit 4) and the upper level of the pit in the clay–silts of the Hemingbrough Formation (unit 5). Photo: P.C.B.

# 3. Methods

## 3.1. Section logging, sediment characterization and dating

A number of vertical sections were cleaned back, recorded and scaled photomontages prepared in 2001 (figures 5 and 6; electronic supplementary material figures S1 and S2). Included on these were sediment texture and colour (using Munsell colours), primary and secondary sedimentary structures, bed contacts, sediment body geometry as well as observations on clast form and lithology, as per Gale & Hoare [42] and Evans *et al.* [10].

To characterize the texture of the sedimentary units and support interpretation of depositional environments, key units were sampled for particle-size analysis using a CILAS 940 laser diffraction particle analyser. For each measurement, samples retaining their field moisture were dispersed in de-ionized water and underwent ultrasound prior to measurement. Results were used to determine the mean particle size and sorting as per Gale & Hoare [42].

Samples for OSL dating were collected by hammering lightproof PVC tubes into freshly cleaned exposures. In the laboratory, the samples were prepared to clean and extract quartz as per Bateman & Catt [43], using the grain-size fractions 90–180 μm. Samples were measured in a Risø DA-12 luminescence reader. Equivalent doses ($D_e$) were determined using the SAR protocol [44] with an experimentally determined preheat of 240°C for 10 s. Results show the rapid decay of OSL dominated by a fast component and good growth with laboratory dose. Replicate Palaeodose (De) values for sample Shfd02010 had a very low overdispersion (OD) (7%) and so the De value used for age calculation was based on the Common Age Model [45]. Dosimetry was calculated via ICP with elemental concentrations converted to dose using data from Guérin *et al.* [46], taking into account attenuation factors relating to sediment grain sizes used, density and palaeomoisture. The latter was set at 23 ± 5% based on estimations of sediment saturation potential, and assumptions of groundwater levels based on the region's palaeoenvironmental history as it is presently understood. The Prescott & Hutton [47] algorithm was used to calculate the cosmogenic-derived dose rate.

## 3.2. Sampling and analysis of organic sediments

Three 3 l samples were collected from organic sediments and processed for plant and animal macrofossils. Two samples (S1 and S2) were collected in 2001 from a channel (figure 7) close to the western edge of the pit, and another (S3) was revealed in 2006 further to the northeast. Samples of the organic silts were placed directly into polythene bags and sealed. Each was disaggregated in water over a 300 μm sieve and the

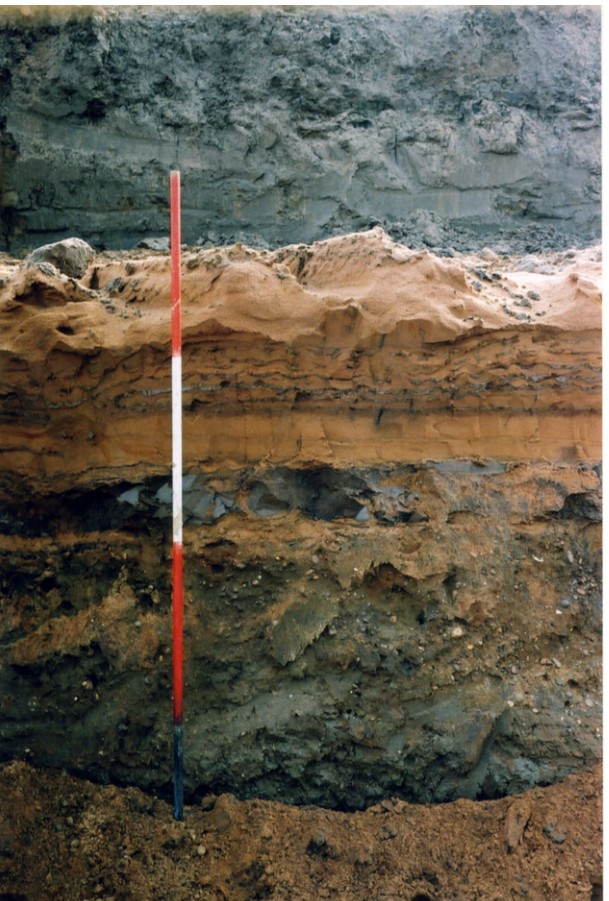

**Figure 5.** Vertical face through units 3 – 5 in the south section at the Finningley Quarry. From base, upwards is diamict (unit 3, dark grey), stratified sand (unit 4) and laminated clay – silt (unit 5, grey). Symmetrical ripple forms with rounded crests and overlain with silt – clay drapes in unit 4 are interpreted as wave ripples. Two metres high pole for scale. Photo: C.F.

residue retained on the mesh was placed in a bowl and paraffin (kerosene) added. This adsorbs onto the surface of insect remains, and when water is added the light oil and insects float and can be decanted off. The float was then washed in detergent to remove the paraffin, cleaned with methanol and sorted under a low-power binocular microscope (cf. [48]). Preservation in sample 1 was insufficient to warrant further work and samples 2 and 3 were used for plant macrofossil and insect identification. Pollen preservation in these samples was considered too sparse to merit further analysis within the scope of the project. As the amount of plant material was slight, only a few 10s of grams, the material was recombined and the easily identified seeds of crowberry, *Empetrum nigrum/hermaphroditum*, were separated for material for accelerator mass spectrometry (AMS) dating. Taxonomy for plant macrofossils follows The Plant List [49], that for Coleoptera Böhme [50] and for Trichoptera Graf *et al.* [51].

# 4. Results

## 4.1. Stratigraphy

At the west end of the pit (approx. 4 m OD), a north–south section, parallel to and close to the Finningley to Austerfield road, exposed over 3 m of sands and gravels, shallowly cross-bedded to the north (1 in figures 3*b* and 5). The series of west–east sections exposed as quarrying progressed southwards between 2001 and 2007 (2 and 3 in figures 3*b* and 4) provided more complete sequences. The stratigraphic sequence (figure 6) is numbered and described from the quarry floor upwards.

### 4.1.1. Unit 1: sand and gravel

Unit 1, as exposed, comprised at least 2 m of reddish-yellow (7.5YR 6/6) sands and gravels (Gm) directly overlying the subhorizontal eroded upper surface of the Sherwood Sandstone (figure 6). This unit is

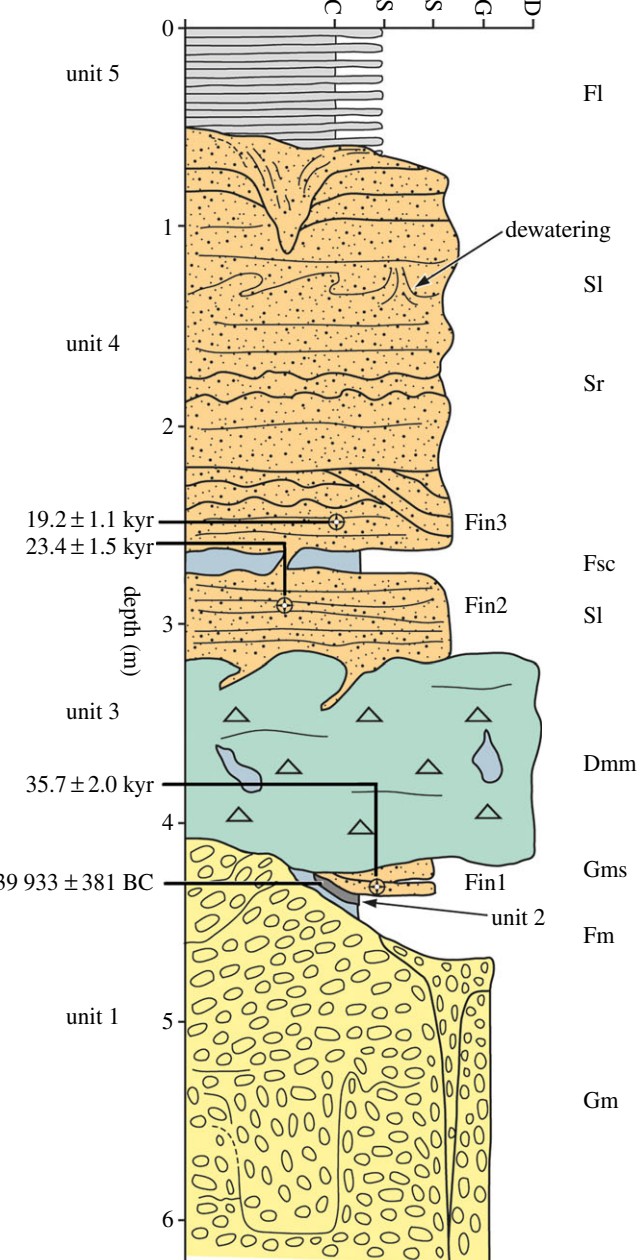

**Figure 6.** Schematic stratigraphy of the Finningley Quarry. Unit 1, ORG (s.l.) of Gaunt [22]; 2, channel deposits with lens of organic silts; 3, diamict; 4, Littoral Sands and Gravels ([22]); 5, Lake Humber clay–silts (Hemingbrough Formation).

massive, although occasionally stratified to crudely stratified with localized cross-beds. Clasts comprising rounded to subrounded quartzite pebbles and a few cobbles with some imbrication were observed where not disturbed (see below). The unit appears completely decalcified and the upper part is penetrated to 1.5 m depth by involutions, festoons and wedge features. Ice-wedge pseudomorphs up to 28 cm wide were infilled with sandy gravel or open-work gravel with vertically aligned elongate pebbles (figure 8). The downturned beds of the host material suggest that this fill was secondary, once ice had melted. Given that the wedge tops are found within unit 1, these ice-wedge pseudomorphs are considered intraformational. This unit is interpreted as being the ORG of Gaunt [22].

### 4.1.2. Unit 2: infilled channel

Unit 2 has been eroded into the underlying sands and gravels of unit 1, forming an infilled channel (figure 7). Within this channel was a series of cut-and-fill structures, sometimes bedded lenses of sandy gravel, sands (Gms, brown, 7.5YR 4/5) and clay–silts (Fm, dark greyish brown, 10YR 4/2). Of particular note was a 25 cm thick lens of thin-bedded and slightly contorted organic silt with sandy

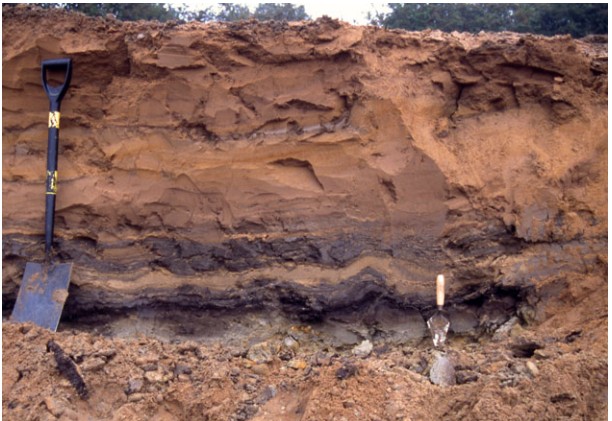

**Figure 7.** Vertical section through unit 2 at the Finningley Quarry: channel deposits with the contorted organic silt sampled for plant and invertebrate remains towards base. Photo: P.C.B.

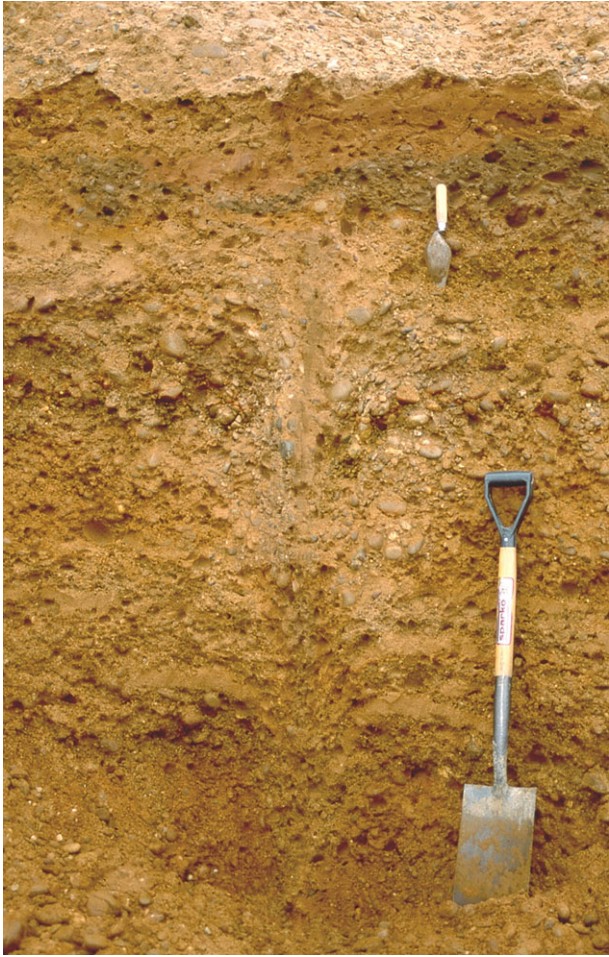

**Figure 8.** Ice-wedge pseudomorph within sand and gravel (unit 1). The wedge is greater than 1.45 m high and has a maximum true width (orthogonal to the axial plane of the wedge) of 28 cm. Note smooth downturning of host strata adjacent to the wedge and vertically aligned elongate pebbles within sandy gravel infill of the wedge. The top of the wedge is overlain by a continuous layer of grey sand and gravel at the top of the trowel, indicating that the wedge is intraformational within unit 1. October 2001. Photo: J.M.

partings (figure 7) which was sampled for plant macrofossils, insects and radiocarbon dating (samples 1 and 2).

Given that the channel infill contains gravels through to fine-grained material, unit 2 is interpreted as indicating a range of fluvial flow regimes with the clay and organic lenses representing low flow, possibly

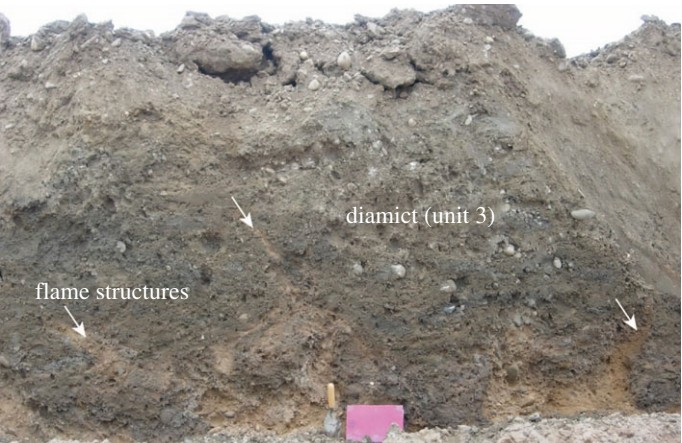

**Figure 9.** Diamict of unit 3. Involuted lower contact of diamict shows flame structures (indicated by arrows), inclined in various directions, of underlying sand and gravel extending up for as much as 0.5 m into the diamict. Diamict is texturally heterogeneous, with pockets of sand and dark grey silt–clay, and some clayey bands. Trowel for scale. August 2007. Photo: J.M.

in overbank or channel marginal settings. While the contortion of the organic and clay lenses could be due to cryoturbation, it probably resulted from post-depositional dewatering. The unit indicates multiple cycles of cutting and infilling of small channels which switched position through time, typical of a braided-stream environment with peaked flows associated with a periglacial environment.

### 4.1.3. Unit 3: diamict

Unit 3 comprises a massive 1–1.5 m thick matrix-supported poorly sorted silty-clay diamict (Dmm, on average 50% gravel, 25% sand, 25% silt–clay). It showed considerable lateral variability in colour and texture. Where the deposit sits on higher ground, it is thinner and more heavily involuted, appearing more red (strong brown 7.5YR 4/6). Where the diamict sits on sandier facies of the underlying sands and gravels, it too is sandier (strong brown 7.5YR 5/6) and where sitting on gravels, it is more clast-rich. In places, the uppermost part of this unit appears laminated and contains occasional dropstones. Sections showed some discontinuous traces of bedding and deformed U-shaped lenses of dark greyish brown (10YR4/2) silty sand or greyish-brown (10YR 5/2) silty-clay lenses aligned parallel or subparallel to the lower contact. The clasts, up to cobble size, are dominated by subrounded to rounded quartzites ultimately derived from the Sherwood Sandstone. The lower boundary of the unit, where it overlies unit 1, appears sharp, erosive and irregular, infilling low points in the underlying unit. Flame structures of sand and gravel extend up to 50 cm into the diamict in variable directions (figure 9). Where unit 3 overlies unit 2 folds and sheared bodies of the silty sand within unit 2 were observed. The upper boundary is also irregular with apparent downward intrusions of sand (figure 9).

The incorporation of rounded quartzite clasts is interpreted as indicating reworking of unit 1 and, therefore, a local rather than regional derivation for this diamict. Given the apparent limits of the LGM ice in the Humberhead Levels to the north [36], a glacial origin is improbable, leaving either solifluction or a subaqueous mass flow similar to that inferred in Klassen & Murton [52] as possibilities. The absence of erratics, beyond those derived from either the ORG or underlying Fluvio-Glacial Sands and Gravels (*sensu* Gaunt [22]), can be accounted for by solifluction of locally derived sediments and fits with the periglacial environment at the time of deposition as judged from the associated involutions. The fines contained within the diamict (25% silt–clay) would have made it frost-susceptible and these along with any permafrost could have created saturated conditions at the base of the active layer. However, while the diamict thins considerably as it rises to the west up an interfluve, suggesting an association with topography, present-day relief to allow either form of flow is limited. Gaunt [22] mapped similar deposits as 'Head' (= solifluction deposit) in the shallow valley to the north of the quarry and there are more extensive deposits, which were perhaps continuous with these before the area was levelled for the expansion of Finningley (now Doncaster Sheffield Airport) airfield during the Second World and Cold Wars. It has been observed that subaqueous flows are often overlain by a drape of silty clay resulting from suspension settling from the water column. This is similar to the deposits that are found above or involuted into the top of the diamict. This clay drape, where it has not been modified by subsequent cryoturbation, has ripples on the

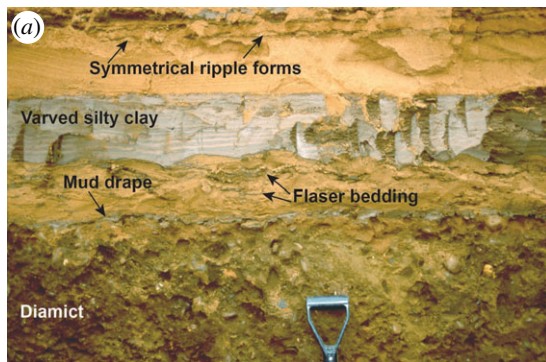
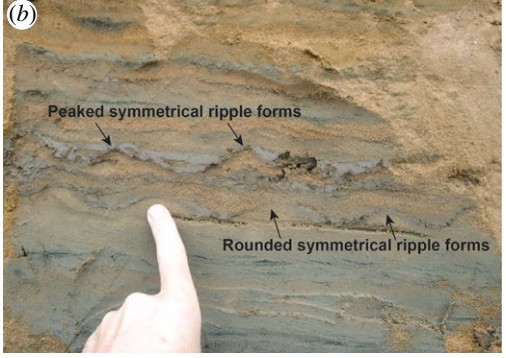

**Figure 10.** (*a*) Sand and varved silty clay of unit 4 overlying diamict of unit 3. Note mud drape (dark grey) directly overlying diamict and flaser bedding within the overlying sand. Above this is a prominent layer 16–20 cm thick of varved silty clay. Ripple form sets are visible in the top of the overlying sand. October 2001. (*b*) Symmetrical ripple forms (wavy bedding) in sand of unit 4. Some ripples have peaked crests, some rounded crests, with both representing wave ripples. Photo: J.M., August 2007.

surface of the diamict (figure 9) which are interpreted as shallow-water reworking after the deposition of the diamict. In this interpretation, a corollary is the temporary existence of a lake, impounded at the Humber Gap.

### 4.1.4. Unit 4: stratified sand

Unit 4, 2.5 m thick, is dominated by medium to fine (mean = 143 µm) sand which is moderately to poorly sorted (mean = 1.09) and reddish-yellow (5YR 6/6) grading to strong brown (7.5YR 5/6). It contains some comminuted fragments of coal, occasionally graded out in a distinct lens. The unit has faint subhorizontal stratification (Sr/Sl) with some channel or cross-bedding and symmetrical and asymmetrical ripples (up to 10–20 mm in height) whose surfaces are accentuated occasionally by clay drapes, for example, in the form of flaser bedding resulting from intermittent flows (figure 10*a*). Ripple crests are orientated approximately east–west. In one long, cleaned exposure, it was possible to see sigmoidal bed forms within the sand, indicating deposition in a littoral environment (figures 10 and 11). Symmetrical ripple forms have either peaked or rounded crests (figures 5 and 11*b*), both representing wave ripples, and the peaked ones characteristic of formation in very shallow, near-emergent conditions. Also within this unit were a number of thin silt–clay horizons (mean = 7 µm, sorting = 1.54, 75% silt, 20% clay), including a persistent subunit of dark grey (5YR 4/1) laminated clay–silt which was approximately 20 cm thick (figure 10*a*), and present in all exposures examined. This subunit contained alternating dark grey and reddish brown laminae, horizontally oriented and separated by sharp contacts. The subunit is interpreted as varves, and a 30 mm long pebble within it as a dropstone. Luminescence samples were collected from above and below this persistent clay–silt subunit.

Numerous structures within this unit suggest it underwent post-depositional deformation. Vertical sand pipes approximately 2 cm wide cut across part of the varved silt–clay subunit, causing it to bifurcate into two in places with sand between the two parts (figure 11*a*). These structures are interpreted as dykes of sand injected up from below forming sand sills within the silt–clay subunit. The upper part of this unit was strongly deformed with water-escape structures. Also towards the top, some irregular and rather chaotic vertical to subvertical structures are developed, including one prominent flat-bottomed 'sag' structure (approx. 80 cm in height) infilled with massive to faintly laminated sand containing silt–clay blocks at the toe, and with adjacent downturned strata and stepped normal faults in the host strata (centre of figure 11*a*). Other examples overlie sand dykes and sills, including one structure comprising grey silty-clay blocks trailing upwards from the top of the varved unit (figure 11*b*). Such structures clearly involved movement of some sediment downward and/or upwards, and they are interpreted as sediment-injection/dewatering structures, indicating that at least the upper part of this unit underwent soft-sediment deformation. Further evidence of soft-sediment deformation is provided by load casts and diapirs (figure 11*c*)—which formed by loading processes—and by dish structures (figure 11*d*)—which formed by upward water escape. The upper boundary to this unit was sharp and largely planar.

A glaciolacustrine origin for unit 4 is inferred from the occurrence of varves and a dropstone. The abundance of wave ripples indicates an open-water body. Peaked symmetrical ripples suggest that this at times experienced very shallow, near-emergent conditions, whereas round crested symmetrical

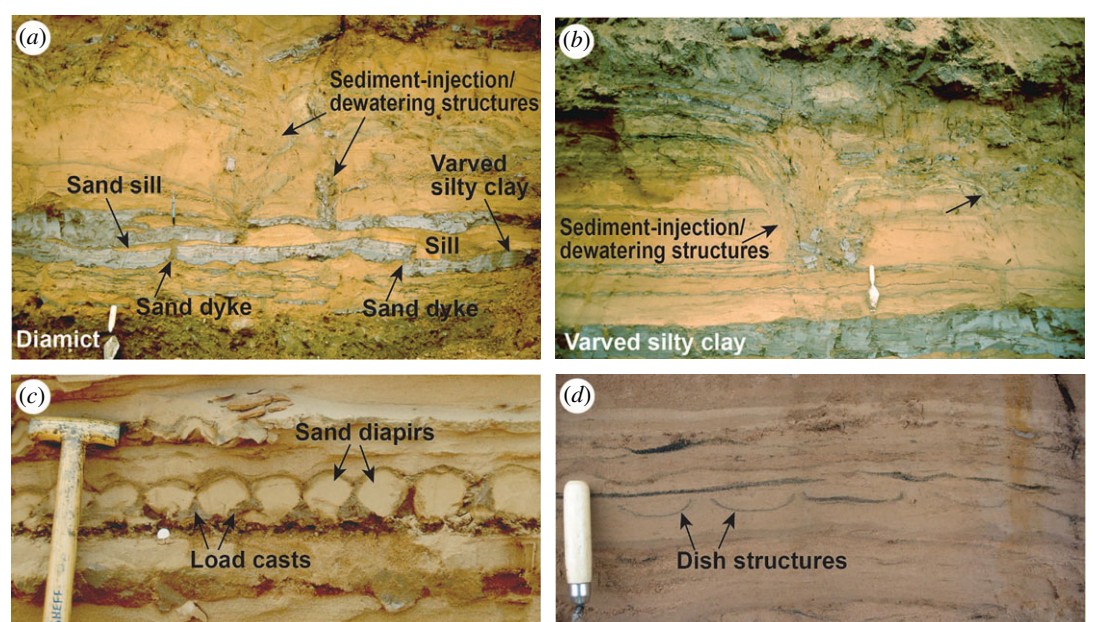

**Figure 11.** Soft-sediment deformation structures in sand and silty clay of unit 2. (*a*) Sand dykes and sills in varved silty clay, beneath prominent sediment-injection/dewatering structures in overlying sand. Trowel for scale. (*b*) Sediment-injection/dewatering structures. The structure on the left has a flat toe below scattered blocks of grey silty clay. Note smooth downturned strata on the left side of structure and normal step faults on the right side. Grey strata of silty clay extend across the top of the structure. Trowel for scale. (*c*) Load casts and sand diapirs. Coin and spade handle for scale. (*d*) Dish structures formed by water escape. All photographs J.M., October 2001.

ripples may have developed in deeper water [53] or formed by reworking of ripple crests during emergence [54]. Wave activity may have triggered dewatering of the loosely consolidated lake sediments, producing the extensive array of soft-sediment deformation structures. This unit is correlated with the Littoral Sands and Gravels of Gaunt *et al.* [23] associated with Lake Humber.

### 4.1.5. Unit 5: laminated clay–silt

Unit 5, as exposed, was approximately 65 cm thick but thinned towards the west. It consisted of uniform, horizontally laminated clayey-silt (80% silt, 20% clay), compact and dark grey (10YR 4/1) in colour, with occasional sandy partings. The mean grain size was 4.0–78 μm and sorting was between 1.28 and 1.41. Calcium carbonate concretions and occasional pebbles, interpreted as dropstones, were also observed. Lamination thicknesses varied from 1–2 mm up to 1 cm. This unit formed the present-day land surface over the western part of the quarry, where iron deposition on ped surfaces represents soil development, although any higher profile had been incorporated into modern ploughsoil. Unit 5 thickened eastwards towards the former course of the river Idle and is interpreted as being part of the Hemingbrough Formation, deposited at depth into proglacial Lake Humber. The unit was overlain by desiccated peat of Holocene age towards the east.

### 4.1.6. Organic lenses

Occasional lenses of dark brown organic silt (sampled as S1–2) occurred within a series of cut-and-fill channels sealed by a diamict (unit 3). A further organic deposit (sampled as S3) was recorded in a section which was interpreted in the field as having had any overlying Lake Humber clay–silts removed prior to excavation of sand and gravel deposits. Both organic deposits appear to relate to pools on an aggrading floodplain with flow in a northeasterly direction.

## 4.2. Plant macrofossils

The raw sediment from sample 2 (unit 2) was dominated by washed-together plant fragments (table 1), in particular water-worn twigs and it also contained numerous sclerotia of *Cenococcum geophilum*. Seeds and fruits were relatively well preserved; however, *Empetrum* was only represented by endocarps and no

**Table 1.** List of plant remains from Finningley (det. O.B.). The nomenclature follows http://www.theplantlist.org/. Sc, sclerotium; oo, oospore; le, leaf; ms, megaspore; s, seed, fruit or endocarp. *Arctous alpina*, *Arctostaphylos alpina*; *Silene suecica*, *Lychnis alpina*; *Stuckenia filiformis*, *Potamogeton filiformis*.

| taxon | type | sample 3 | sample 2 |
|---|---|---|---|
| fungi | | | |
| *Cenococcum geophilum* Fr. | Sc | — | 140 |
| algae | | | |
| *Nitella* sp. | oo | — | 1 |
| bryophytes | | | |
| *Sphagnum* sp. | le | — | 5 |
| vascular plants | | | |
| *Arctous alpina* (L.) Nied | s | 42 | — |
| *Betula nana* L. | s | 3 | — |
| *Batrachium* sp. | s | 4 | 2 |
| *Carex aquatilis* Wahlenb./*C. bigelowii* Torr. Ex Schwein. | s | 40 | — |
| *Carex* spp. | s | 5 | 2 |
| *Comarum palustre* L. | s | — | 1 |
| *Epilobium* sp. | s | — | 1 |
| *Empetrum nigrum* L. | s | — | 24 |
| *Luzula* cf. *spicata* (L.) DC. | s | — | 1 |
| *Potentilla anserina* L. | s | — | 2 |
| *Potentilla* cf. *crantzii* (Crantz) Beck ex Fritsch | s | 21 | — |
| *Ranunculus* sp. | s | — | 1 |
| *Rumex acetosa* L. | s | 1 | — |
| *Rumex acetosella* L. | s | 34 | — |
| *Salix polaris* Wahlenb. | l | 1 | — |
| *Selaginella selaginoides* (L.) P. Beauv ex Mart. and Schrank | ms | — | 9 |
| *Silene suecica* (Lodd.) Greuter and Burdet | s | 13 | 1 |
| *Stuckenia filiformis* (Pers.) Börner. | s | — | 2 |
| *Viola palustris* L. | s | 1 | — |

leaves were found. Only a single tiny leaf of *Salix polaris* was found. No caryopses of grasses were found and bryophytes were only represented by a few *Sphagnum* leaves. The lack of such remains could be due to poor preservation, but it could also be due to hydrodynamic sorting in flowing water. It is surprising that no moss remains were found, because mosses are important in Arctic plant communities. Remains of mosses usually preserve well, and have been reported from a number of Middle Devensian (MIS 3) sites in Britain and elsewhere in northwestern Europe (e.g. [55]). Again, this may be due to hydrodynamic sorting.

The most common fossil was *C. geophilum*, a fungus that lives in various soil types. Common presence of sclerotia of this species is usually taken as an indication of soil erosion [56], but the light sclerotia may also have been concentrated by flowing water. Remains of macrolimnophytes are rare in the fossil assemblage. *Ranunculus* sect. *Batrachium* sp. is represented by a few achenes, and this plant usually grows along the shores of ponds or small lakes. Submerged water plants comprise the charophyte *Nitella* sp. (1 oospore) and *Stuckenia filiformis* (2 endocarps). The submerged taxa indicate small lakes with clear, oligotrophic water. The rarity of water plant remains indicates that ponds or lakes were scarce in the area. *Carex* spp. may have grown in shallow water at lake or pond margins or in mires and *Sphagnum* in bogs. *Viola palustris* is also a mire plant, and *Selaginella selaginoides* is found in bogs or other areas with moist or wet soils.

Dwarf-shrub heath communities are indicated by macrofossils of several species of dwarf shrubs. Endocarps of *Arctous alpina* were common, which is surprising because there appears to be no previous

fossil records of this species from Britain [57,58]. *Arctous alpina* is a circumpolar low-arctic and boreal plant [59]. At the present day in the British Isles, it is restricted to montane moorland in Scotland. Fossil endocarps of *A. alpina* have been reported, for example, from Eemian deposits in Greenland [60] and Lateglacial deposits in southern Sweden and Denmark (e.g. [61,62]). The endocarps of *A. alpina* are similar to, but flatter than those of *Arctostaphylos uva-ursi*, which has been reported from Middle Devensian deposits in Britain [57]. Endocarps of *E. nigrum* were also common. These are referred to *E. nigrum* and they may come from *E. nigrum* subsp. *hermaphroditum* (Hagerup) Böcher. *Empetrum nigrum* endocarps are rarely reported from Middle Devensian deposits in Britain, but one endocarp was reported from Scotland by Bos *et al.* [58]. It has also, but rarely, been reported from Middle Devensian deposits elsewhere in northwest Europe. Houmark-Nielsen *et al.* [63] found a single endocarp of the species in a deposit in northwestern Denmark. *Empetrum nigrum* is a circumpolar plant, which has a wide geographical range in the Arctic, but is also widespread in the temperate zone.

*Betula nana* is represented by three nutlets. There are numerous fossil records of this species from Britain, following the first report by Heer [64], and its remains are common in Middle Devensian deposits [57]. *Betula nana* is also a circumpolar arctic and northern boreal plant, presently restricted in the UK to northern England and Scotland. The last woody plant recovered was *S. polaris*. It was represented by a tiny 1.4 mm long leaf. Although small, the margin was entire and showed no signs of a crenation or serration as seen in *Salix herbacea*. The sample also contained a few budscales and a small twig with two budscales of *Salix* sp. *Salix polaris* is one of the most cold-adapted of woody plants, and it has previously been reported from several Middle Devensian deposits in England [57,65]. It is currently found in northern Europe (including Svalbard), northern Asia and northwestern North America.

Herbaceous plants include *Luzula* cf. *spicata*, *Potentilla* cf. *crantzii*, *Rumex acetosa* and *Silene suecica*, which may have grown in heaths. *Epilobium* sp. grow in heaths or in plant communities on wet or moist soil. Their seeds have not previously been reported from Middle Devensian deposits in Britain according to Godwin [57]. *Rumex acetosella* is indicative of dry sandy or gravelly soils with patches of open vegetation and *Potentilla anserina* is often found near the shores of lakes, on bare ground or in grass-rich habitats; it can tolerate salt-rich soils.

In summary, the plant macrofossils suggest that the landscape was treeless tundra that supported widespread dwarf-shrub heaths. Some of the recorded plants indicate base-rich soils, whereas others indicate more acidic conditions. The flora indicate a continuous snow cover in winter. Moisture conditions varied from dry and freely draining to wetland, pond or riparian.

## 4.3. Vertebrate remains

In 2005, a workman from the Finningley Quarry took two lengths of tusk, presumed to be of mammoth and a scapula of a bovid into Doncaster Museum [66]. He refused to leave the material and their subsequent fate is unknown, but he did suggest that other bones had been found during quarrying. Howes (C. Howes, personal communication, 2018), who examined the material, has suggested that bovid was either *Bos primigenius*, or steppe bison, *Bison* cf. *priscus*. It is unfortunate that it is not possible to place the bones in the stratigraphy. However, the underlying ORG is completely decalcified and the overlying deposits of the proglacial lake belong to a period when carrying capacity was too low to support any substantial vertebrate fauna, so it is probable therefore that these relate to the MIS 3 deposits and some support for this is provided by the dung faunas from the insect samples.

## 4.4. The insect faunas

### 4.4.1. Coleoptera

Two samples produced insect remains. The fauna from sample 2 is more extensive than that from sample 3, and includes more species no longer found in Britain (cf. [67]). Several of the latter have a present distribution which is essentially Siberian. The fauna is dominated by a dung beetle, *Aphodius (Chilothorax) jacobsoni*, identified from heads, pronota and patterns of maculae on the elytra. In his review of subgenus *Chilothorax*, however, Frolov [68] indicated that *A. jacobsoni* consists of a complex of several closely related species, although none appears to occur west of eastern Kazakhstan, and southern Siberia and the core distribution is Mongolian [69]; there appear to be little habitat data available. Other fossil records are restricted to MIS 3 deposits at Queensford, on the Thames terraces in Oxfordshire, where it is the most common dung beetle [70] and Coope (in [71]) also included a possible identification from similar aged deposits at Sandy in Bedfordshire.

*Helophorus praenanus* is recorded from grassy pools in eastern Siberia, Mongolia and southwards into northern China [72,73]. The taxon was first identified as a fossil from Starunia, and Angus [72] also noted that it is a not infrequent component of insect assemblages from the colder parts of glacials. There are MIS 3 records from Baston Fen, south Lincolnshire [74], Earith, Cambridgeshire [75], Great Billing, Northamptonshire [76], Kempton Park, Surrey [77], Leeds, West Yorkshire [78], Lechlade, Queensford and Standlake, Oxfordshire [70] and Whitemoor Haye, Staffordshire [79]; records from Kirby-on-Bain in Lincolnshire [80] have recently been re-assigned to MIS 6 [25]. The remainder of the water beetle fauna also includes indicators of cold conditions, with the small dytiscids *Hydroporus lapponum* and *H. notabilis,* no longer found in the British Isles, being restricted largely to above the treeline in Scandinavia [81] and occurring in ponds and shallow lakes. Both occur in the Lateglacial at St Bees in Cumbria [82], but only the latter has been found on other MIS 3 sites, at Lechlade and Queensford [70] and Upton Warren, Worcestershire [83]. The ground beetle *Pelophila borealis* would occur on the sandy margins of temporary pools on the floodplain [84], where the small rove beetle *Bledius arcticus* would also have burrowed [85]. Both the more abundant *Notiophilus aquaticus* and *Amara quenseli* would have ranged more widely on sparsely vegetated sandy ground, where several of the rove beetles would have lived in the litter under low herbage. Several of the latter, including *Pycnoglypta lurida*, *Olophrum boreale*, *Eucnecosum* species, *Acidota quadrata*, *Boreaphilus henningianus* and *Holoboreaphilus nordenskioldi*, are essentially arctic and Siberian in distribution, frequent in litter beneath low willows; all are not infrequent MIS 3 fossils. *Holoboreaphilus nordenskioldi* is Holarctic, but only extends westwards as far as the Kanin Peninsula in Arctic Russia and is restricted to the tundra [86]. For the Nearctic, Morgan *et al.* [87] suggested an association with heavily disturbed ground over permafrost.

Using habitat traits from the classification in the BugsCEP database [88], we infer that the more extensive fauna from sample 2 indicates a varied landscape with more or less equal indications of water, meadowland, wetland/marshes and heathland/moorland. Both standing and running water are indicated, the former more robustly than the latter (figure 12). There are few indications of the specific nature of the flora from the beetles. The weevils *Otiorhynchus arcticus*, *O. nodosus*, *O. rugifrons* and *Tropiphorus obtusus* are essentially polyphagous on low vegetation, the larvae feeding on the roots. Species of *Notaris* are oligophagous on waterside and aquatic sedges, grasses and reeds [90,91], on which the small pollen beetle *Kateretes pedicularis* would also feed [92]. The remaining weevil, *Baris* cf. *artemisiae*, not currently found in Britain, breeds in wormwood, *Artemisia* species [93]. Species of *Phratora* feed on willows and poplars, while the ladybird *Ceratomegilla ulkei* is a predator on aphids on the taiga and tundra across the eastern Palaearctic and Nearctic from Kazakhstan to Hudson Bay in eastern Canada [94]. It is known in fossil form from MIS 3 deposits at Queensford [70], with earlier last glaciation records from Shropham in Norfolk [95] and Stanwick, Northamptonshire [96]. The remaining indicators of flora are the two byrrhids, *Simplocaria metallica* and *S. semistriata*, which feed on mosses [97,98]. The former is boreo-montane, not extending west of the Scandinavian mountains in Europe but known from Greenland [97].

The identification of the alpine chrysomelid *Oreina frigida* is tentative because of the fragmented nature of the fossil. Koch [99] noted it as breeding in purple coltsfoot, *Homogyne alpina*, and Pasteels *et al.* [100] found an association of the imagines with the umbellifer, *Meum athemanticum*.

The insect fauna from sample 3 was sparse, relatively poorly preserved and there were few species (table 2), of which only *Helophorus sibiricus* provides significant ecological information. No longer found in the British Isles, it is a water beetle associated with pools at the edge of snow melt and upland rivers [72,101], characteristic of the taiga but extending into the tundra zone and southwards into montane temperate forest in the east. It has a broad Holarctic distribution, ranging from the Scandinavian mountains eastwards across northern Russia and Siberia to Alaska and northern Canada, west of the Mackenzie Delta [102]. A frequent and widespread fossil in Middle Devensian/Weichselian (MIS 3) and Lateglacial deposits in the British Isles [88], it has the distinction of a fossil record extending back to the Miocene in southern Siberia [102]. The scarabaeids in the sample include *Aphodius borealis*, a Holarctic dung beetle, which shows a preference for elk (moose) dung in shaded localities [103], although it has also been recorded on dunes and in the droppings of other large herbivores [104]. *Aegialia sabuleti* is found in plant litter on sandy river and stream banks [105]. Pittino [106] has recently described a new species, *A. insularis* as endemic to the British Isles, but this cannot be separated from *A. sabuleti* on the fossil material. While *A. sabuleti* (s.l.) is a not infrequent Quaternary fossil [88], *Aphodius borealis* has only two other British fossil records, from MIS 3 deposits at Tattershall Castle, Lincolnshire [80], and from a Roman site in East Yorkshire (P.C.B., unpublished data, 2015, Barmby-on-the-Marsh, available in the BugsCEP database [88]).

Overall, the picture is of a pond on a braided floodplain on sparsely vegetated sandy tundra, visited occasionally by larger vertebrates.

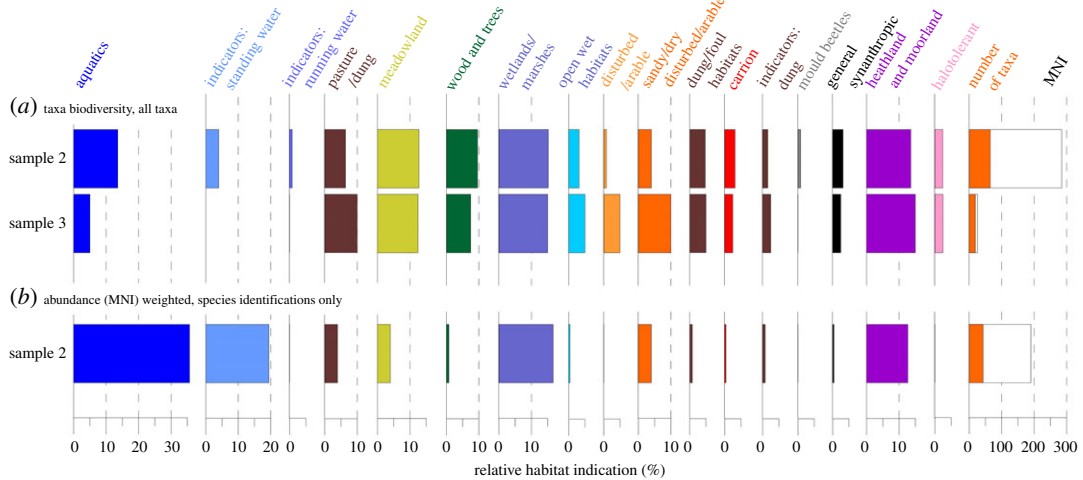

**Figure 12.** Beetle habitat trait-based environmental reconstruction using (*a*) taxon occurrence only and (*b*) species-level abundance-weighted data. (*a*) General reconstruction, giving equal weight to each taxon, based on the habitat preferences at all taxonomic resolutions; (*b*) uses only taxa identified to species and weights the reconstruction according to the number of individuals found, thus providing a more constrained reconstruction for the most reliable habitat indications while ignoring evidence from poorly resolved identifications. Neither reconstruction is 'correct', but together, they provide a tool for interpreting the environmental implications of the fauna. Note that a taxon may be included in more than one habitat class. MNI, minimum number of individuals. (See [89] for classification and methodology, [88] for classification data.)

### 4.4.2. Trait-based habitat reconstruction from the Coleoptera

A quantitative, habitat trait-based reconstruction of the environment represented by the fauna allows for a general landscape model to be constructed, and comparison of the landscapes represented by the Coleoptera in the two samples (figure 12).

Taxon occurrence data (i.e. assuming one individual per taxon) suggest a mixed landscape for sample 2 with more or less equal indications of water, meadowland, wetlands/marshes and heathland/moorland landscape components. Clear indications of standing water are provided by five species of water beetle (see above), but the running water indication is only provided by the Alder fly, *Sialis* sp., larvae (Megaloptera). Running water taxa are somewhat unreliable due to their propensity to be carried with the current and deposited in the sediments of pools and lakes. The presence of shaded environments is suggested by a number of taxa, mainly at the genus level (compare figure 12*a*,*b*), but only one taxon is an obligate xylophage or arboreal feeder. The single specimen of *Phratora* sp. would most likely have fed on the leaves of willow shrubs and the genus includes species common in arctic Scandinavia. The only potential woodland indication at the species level is the somewhat eurytopic *Simplocaria semistriatus*, a moss feeder as likely to be found under stones as in woodland.

Removing the poorly resolved identifications (i.e. not to species), which provide a less secure and more general reconstruction, and weighting the reconstruction by numbers of individuals (MNI) (figure 12*b*), focuses the picture to a pond in a less diverse, open landscape, with sparse vegetation of wetland and moorland character.

In the quantitative reconstruction, sample 3 appears somewhat similar to sample 2, with the exception of less prominent aquatic habitats and more disturbed ground (both wet and dry) in the former. The paucity of this fauna precludes the use of abundance weighting or only taxa identified to species, but the indications of disturbance and wet and dry habitats could fit with a braided river system. Evidence for grazing animals is potentially found in both samples, more so in sample 2, but this could be a reflection of the better quality of the data in the latter.

### 4.4.3. Trichoptera (caddisflies)

Sample 2 from Finningley contains preserved frontoclypeal, pro- and meso-notal sclerites from larvae of cased caddisflies (Order Trichoptera), and is made up of species from the families Phryganeidae, Apataniidae, Limnephilidae and Molannidae. As aquatic larvae, caddisflies can be found in habitats ranging from fast-flowing rivers to temporary field ponds. Assemblages of taxa from palaeo-deposits can offer a means of palaeoenvironmental reconstruction [107].

**Table 2.** Insect remains from Finningley (det. P C.B., M.G. and E.P.).

| taxon | sample 1 | sample 2 | sample 3 |
|---|---|---|---|
| Carabidae | | | |
| *Pelophila borealis* (Payk.) | | 1 | |
| *Notiophilus aquaticus* (L.) | | 7 | 1 |
| *Patrobus* sp. | | 1 | |
| *Bradycellus caucasicus* (Chaud.) | | 1 | |
| *Amara quenseli* (Schön.) | | 2 | |
| Haliplidae | | | |
| *Haliplus* sp. | | 1 | |
| Dytiscidae | | | |
| *Hydroporus lapponum* (Gyll.) | | 22 | |
| *H. notabilis* LeC. | | 1 | |
| *Hydroporus* sp. | | 1 | |
| *Oreodytes alpinus* (Payk.) | | 1 | |
| Hydroporinae indet. | | | 1 |
| *Agabus sturmii* (Gyll.) | | 1 | |
| *Agabus labiatus* (Brahm) | | 5 | |
| *Agabus* sp. | | 1 | |
| *Rhantus* sp. | | 1 | |
| *Colymbetes* sp. | | 1 | |
| Hydraenidae | | | |
| *Ochthebius* sp. | | 7 | |
| Hydrophilidae | | | |
| *Helophorus sibiricus* (Mots.) | | 16 | 1 |
| *H. grandis* Ill. | | 20 | |
| *H. aequalis* Thoms. | | 22 | |
| *H. praenanus* (Lom.) | | 1 | |
| *Helophorus* (small) spp. | | 7 | |
| *Cercyon melanocephalus* (L.) | | 1 | |
| *Cercyon* spp. | | 2 | |
| *Hydrobius fuscipes* agg. | | 1 | |
| Silphidae | | | |
| *Thanatophilus* sp. | | 1 | |
| Staphylinidae | | | |
| *Megarthrus prosseni* Schatz. | | 1 | |
| *Pycnoglypta lurida* (Gyll.) | | 14 | |
| *Olophrum fuscum* (Grav.) | | 10 | |
| *O. boreale* (Payk.) | | 2 | |
| *Olophrum* spp. | | 14 | |
| *Eucnecosum brachypterum* (Grav.) | | 13 | |
| *E. brunnescens* (Sahl.) | | 1 | |
| *Acidota quadrata* (Zett.) | | 2 | |
| *Lesteva longoelytrata* (Goeze) | | 7 | |

(*Continued.*)

| taxon | sample 1 | sample 2 | sample 3 |
|---|---|---|---|
| *Anthophagus* sp. | | | 1 |
| *Boreaphilus henningianus* Sahl. | | 2 | |
| *Holoboreaphilus nordenskioeldi* (Mäklin) | 3 | | |
| *Aploderus caelatus* (Grav.) | | 1 | |
| *Anotylus nitidulus* (Grav.) | | 1 | |
| *Bledius arcticus* Sahl. | | 1 | |
| *Bledius* spp. | | 2 | |
| *Stenus* sp. | | 6 | |
| *Quedius boops* (Grav.) (grp) | | 1 | |
| *Quedius* sp. | | 1 | |
| Mycetoporini indet. | | 2 | |
| *Tachyporus* sp. | | | 1 |
| *Tachinus elongatus* Gyll. | | 1 | |
| *Tachinus* sp. | | 2 | |
| Aleocharinae indet. | | 39 | 1 |
| Indet. | | | 1 |
| Byrrhidae | | | |
| *Simplocaria semistriata* (F.) | | 1 | |
| *S. metallica* (Sturm) | | 1 | |
| Brachypteridae | | | |
| *Kateretes pedicularius* (L.) | | 1 | |
| Cryptophagidae | | | |
| *Atomaria* spp. | | 2 | |
| Latridiidae | | | |
| *Corticaria/Corticarina* sp. | | 1 | |
| Coccindellidae | | | |
| *Scymnus femoralis* Gyll. | | 1 | |
| *Scymnus* (s.l.) spp. | | 2 | |
| *Ceratomegilla ulkei* Crotch | | 2 | |
| Scarabaeidae | | | |
| *Aegialia sabuleti* (Panz.) | | | 1 |
| *Aphodius merdarius* (F.) | | 1 | |
| *A. borealis* Gyll. | | | 2 |
| *A.* (*Chilothorax*) *jacobsoni* Kosh. | 7 | | |
| *Aphodius* spp | | 3 | 3 |
| Chrysomelidae | | | |
| Cf. *Oreina frigida* (Weise) | | 1 | |
| *Phratora* sp. | | | 1 |
| Chrysomelinae indet. | | 2 | |
| Curculionidae | | | |
| *Otiorhynchus arcticus* (*O. fabricius*) | | 2 | 1 |
| *O. nodosus* (Müll.) | | 2 | 1 |

(*Continued.*)

| taxon | sample 1 | sample 2 | sample 3 |
|---|---|---|---|
| *O. rugifrons* (Gyll.) | | | 1 |
| *Tropiphorus obtusus* (Bonsd.) | | | 1 |
| *Notaris bimaculatus* (F.) | | | 2 |
| *N. acridulus* (L.) | 1 | | |
| *N. aethiops* (F.) | | | 1 |
| *Baris* cf. *artemisiae* (Hbst) | | 1 | |
| Megaloptera | | | |
| Sialidae | | | |
| *Sialis* sp. (larvae) | | 3 | |
| Diptera | | | |
| Anthomyidae | | | |
| *Delia* cf. *fabricii* (Holm.) | | 1 | |
| *Botanophila* sp. | | 1 | |
| Calliphoridae | | | |
| Indet. | | 1 | |
| Indet. (puparium) | | 1 | |
| Trichoptera (table 3) | | 53 | |

The most abundant and diverse taxa in the Finningley sample are from the Limnephilidae (table 3), and these indicate a varied habitat with riverine and lake-edge species: *Ecclisopteryx dalecarlica* has larval cases made of mineral particles and is found in slow-flowing sections of river edges, together with *Anabolia nervosa*, *Limnephilus stigma* and *L. subcentralis*. *Grammotaulius* spp., *Limnephilus algosus* and *L. picturatus* are from more permanent shallow pools, ponds and lakes and have cases constructed of plant fragments. The cold-adapted family Apataniidae is also represented and *Apatania* spp. are found in stony and gravel-bed brooks, rivers and lakes; the larval case is made of mineral particles. *Molanna albicans* (Molannidae) is found in slow-flowing rivers and small upland lakes, building a case also of mineral grains, and *Agrypnia picta* (Phryganeidae) is a species of permanent ponds and mires, constructing a case of large fragments of plant material, e.g. *Carex* spp. [108,109].

Of special interest is the species *L. algosus* (McLachlan, 1868), a cased larva found among submerged vegetation in permanent shallow pools, ponds, lakes, slow-flowing brooks and rivers [109–111]. To date, there are no fossil records of this species in the UK, but reference is made to this species, described as Limnephilidae indet. in Greenwood *et al.* [107], Whittington *et al.* [112] and Schreve *et al.* [79], which can now be recorded as *L. algosus*. Figure 13 shows the frontoclypeal apotome from both modern and fossil material of this species; its colour pattern, shape, micro-sculpture and size are useful characters for identification. This species is recorded in sediments in the UK ranging from 41–43 cal yr BP at Whitemoor Haye, River Tame, Staffordshire [79], to 15 793–13 306 cal a BP. at Clettnadal, Shetland [112], and to 13 817–13 543 cal a BP (CALIB 3.0) from Hemington, River Trent, Nottinghamshire [107].

A map of present distribution (figure 14) shows *L. algosus* to be widely distributed in the western Palaearctic. In Europe, it is present in areas of high altitude and/or latitude and is described as a cold stenotherm (restricted to a narrow range of cold temperatures) of the alpine and subalpine zones [51]. It is found in Norway, Sweden, Finland, Russia (European Russia, Siberia and in the Russian Far East), Germany (Bavaria), Austria (Tyrol and Oberösterreich), Switzerland, Czech Republic (Bohemia) and Slovakia [113–115]. The circumpolar nature of this species distribution was highlighted by Nimmo [116], who described a new species of *Limnephilus* from the Northwest Territories, Canada; this he named *L. innuitorum*, after the Inuit people of the Canadian and Greenlandic Arctic. Later Nimmo *et al.* [117] also recorded *L. innuitorum* from Chukotka in the Russian Far East. Grigorenko [118] then recognized that both descriptions matched, making *innuitorum* a synonym of *algosus* [113].

Overall, the larval caddis assemblage suggests a cold, arctic-equivalent environment that is primarily one of slow-flowing streams and standing water (ponds and lakes) with occasional channels with moderate to fast flows.

**Table 3.** Trichoptera from Finningley (det. M.G.).

| Limnephilidae | | | |
|---|---|---|---|
| Apataniinae | *Apatania* sp. | 1 | case of mineral particles, in rivers, brooks and lakes shores |
| Drusinae | *Ecclisopteryx dalecarlica* Kolen | 1 | large streams and rivers; stony substrate. Case of sand grains |
| Limnephilinae | *Anabolia* cf *nervosa* (Curtis) | 4 | rivers, lakes and ponds on sandy-silty beds. Case of plant pieces or of sand grains |
| | *Grammotaulius* cf *signatipennis*McLach./*nigricornis* (Retzius) | 1 | among emergent vegetation in temporary pools and ponds, also rich fen. Case of plant fragments |
| | *Limnephilus algosus* (McLach.) | 17 | among submerged vegetation in permanent shallow ponds, pools and lakes, slow-flowing brooks and rivers. Case of leaf fragments, e.g. Carex sp. |
| | *L.stigma* Curtis | 1 | Among lush vegetation in ponds, lakes and slow-flowing rivers and marshes |
| | *L. subcentralis* Brauer | 8 | among marginal vegetation in pools, shallow ponds, lakes, rivers, streams with brackish waters. Case of plant fragments |
| | *L. picturatus* McLach. | 8 | |
| | *Limnephilus* sp2 in det | 15 | |
| | *Limnephilus* sp3 in det | 1 | |
| Molannidae | | | |
| | *Molanna albicans* (Zett.) | 1 | in small upland lakes (Wales); in lakes in central and northern Scotland and slow-flowing rivers in central Ireland. Case of mineral grains |

### 4.4.4. Diptera

The fly assemblage from the site consists entirely of puparial fragments and is badly preserved and fragmented. One of the specimens included anal spiracles which made identification possible, but in other cases, the small fragments became more fragmented while handling. Two of the taxa belonged to Anthomyiidae, a family of leaf and stem miners. *Delia* cf. *fabricii* is a species associated with smooth meadow grass, *Poa pratensis* [119]. The larvae pupate in the soil in the beginning of June and the adults emerge mid-summer [120]. *Botanophila* sp. belongs to another group of stem borers, although some species are associated with fungi. A fragment of a calliphorid puparium, probably associated with either carrion or dung, was also recovered but identification to species level was not possible. Although information from the dipterous assemblage is limited, the environmental information points to a grassland environment.

### 4.5. Chronology

The *Empetrum* endocarps (crowberry) sample from the organic lens within the channel (unit 2) provided a radiocarbon age of 37 057 ± 457 [14]C years BP (UBA-33853). This was calibrated to 40 314–39 552 cal. years BP (2 sigma), using the INTCAL13 dataset [121].

The remaining ages both on the channel and other units are based on the OSL results (figure 15 and table 4). Sample Fin 1 (Shfd02010) from unit 2, and above the radiocarbon sample, gave an age of 35.7 ± 2.0 ka corresponding closely to the radiocarbon date from this unit. Sample Fin 2 (Shfd02011) from the lower part of unit 4 beneath the clay gave an age of 23.4 ± 1.5 ka. Sample Fin 3 (Shfd02012) from above the clay in unit 4 gave an age of 19.2 ± 1.1 ka.

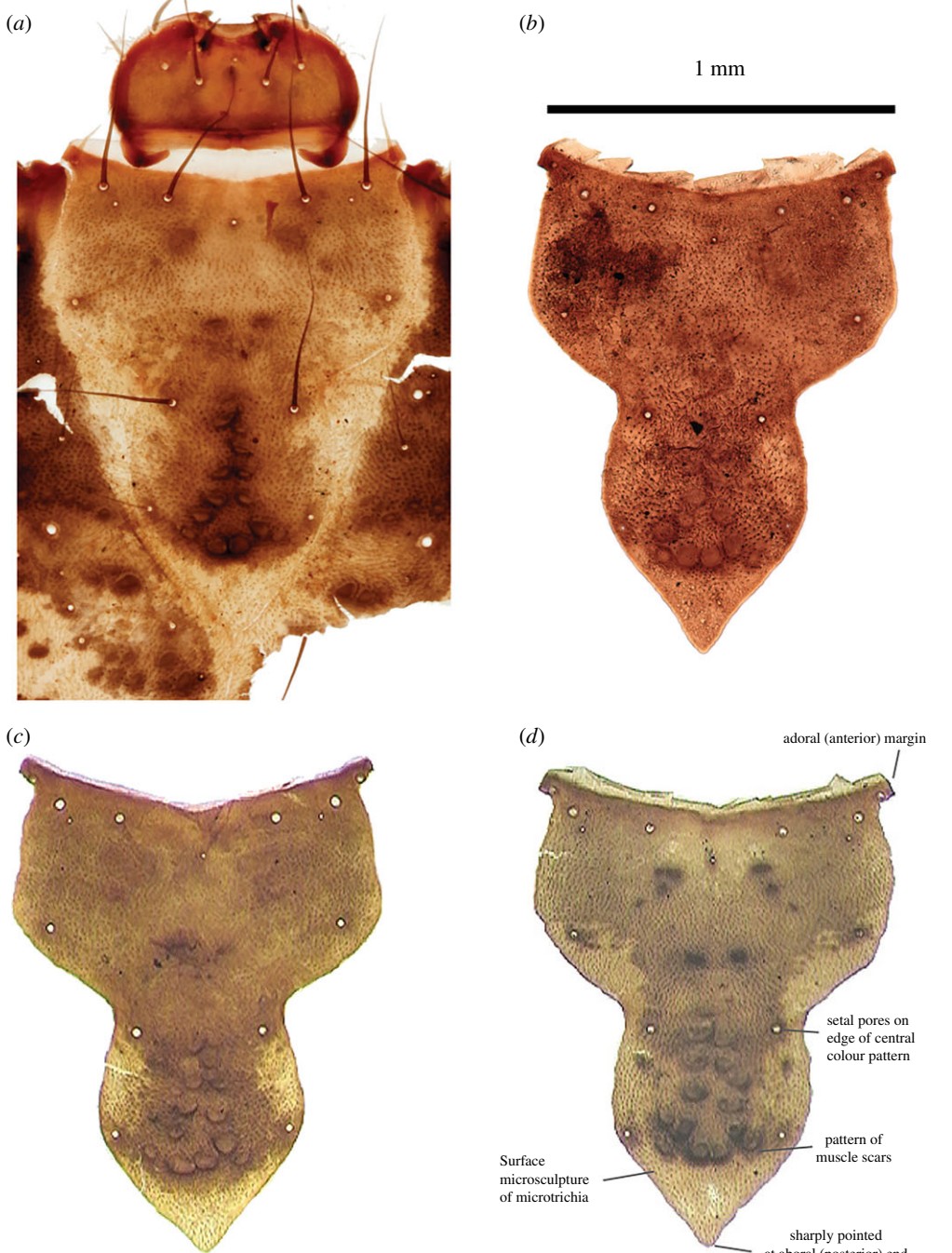

**Figure 13.** (a–d) *Limnephilus algosus*, showing the frontoclypeal apotome from both modern and fossil material of this species. Photo: Department of Geography, Loughborough University.

From these results, it is clear that unit 1 is older than 35.7 ± 2.0 ka and unit 2 dates to between 39.9 ± 0.4 and 35.7 ± 2.0 ka BP. This corresponds to MIS 3 [122] and to between Greenland Stadial 9 (GS-9) through to Greenland Stadial 7 (GS-7) encompassing the interstadial of GI-8 [6]. Unit 3 lies between 35.7 ± 2.0 and 23.4 ± 1.5 ka (MIS 3/2). Unit 4 dates to between 23.4 ± 1.5 ka and sometime after 19.2 ± 1.1 ka (MIS 2) and Greenland Stadial 2 (GS-2; [6]). Unit 5 is younger than 19.2 ± 1.1 ka.

## 4.6. Palaeoclimate

### 4.6.1. Periglacial structures and climate

Ice-wedge pseudomorphs within the sand and gravel of unit 1 indicate the former occurrence of permafrost, and their intraformational nature implies that permafrost was broadly contemporaneous with sediment

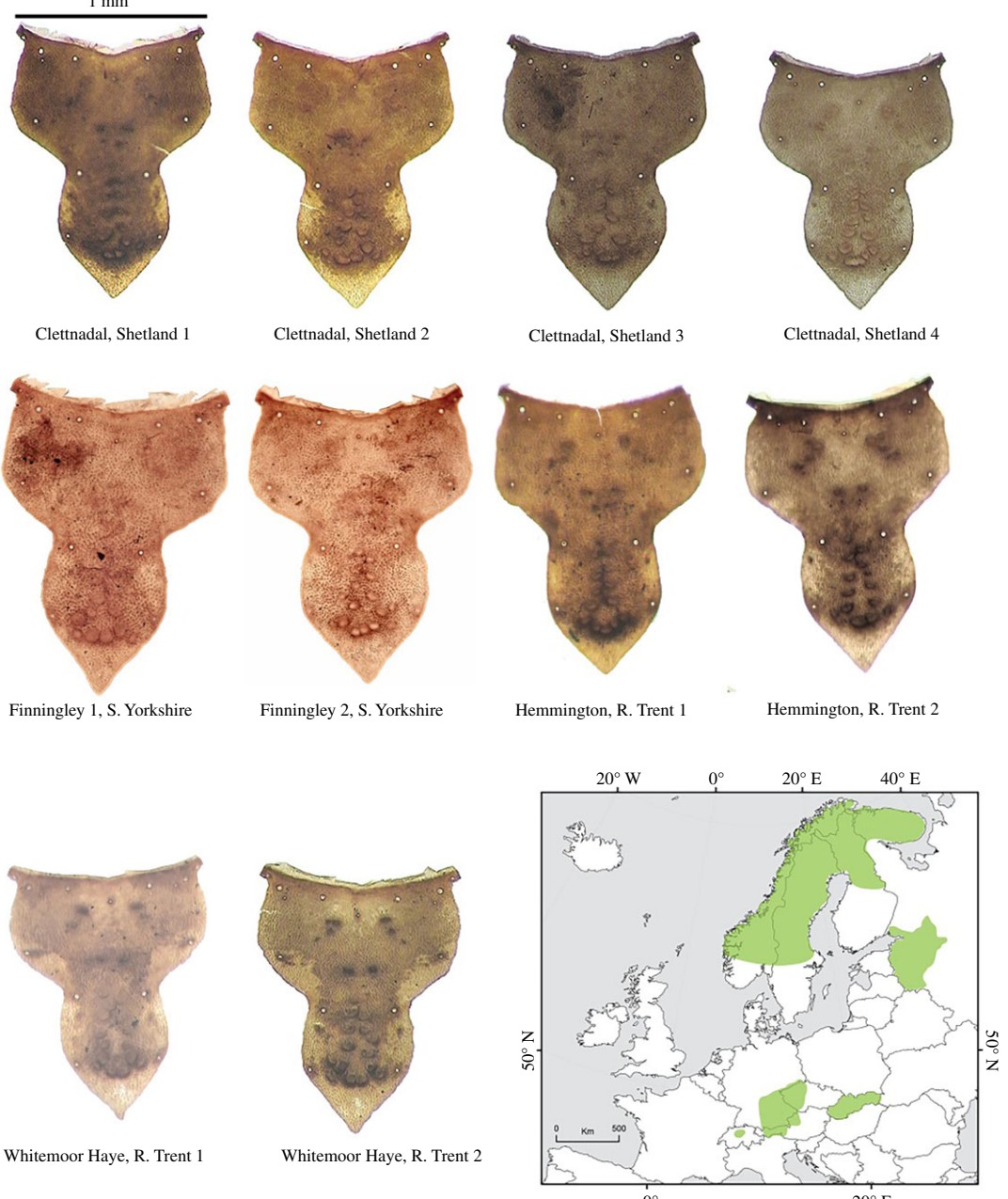

**Figure 14.** Fossil specimens of *L. algosus* and map of modern distribution. Photo: Department of Geography, Loughborough University.

deposition. Their occurrence within aggradations of braided river deposits is commonly taken to indicate that ice wedges developed in inactive parts of the floodplain; such wedges may have thawed due to lateral migration of river channels or water ponding or flowing in troughs overlying ice wedges [123]. The former presence of permafrost during deposition of unit 1 discounts an interglacial or interstadial environment for gravel deposition. Instead, the gravel was deposited under a cold-stage (permafrost) environment. This contrasts with the overlying deposits from which the organic remains were retrieved.

Involutions present within the gravel may relate to some form of periglacial disturbance (e.g. differential frost heave), but it is difficult to be more specific about genesis or climatic significance in this instance because involutions in gravel are often difficult to interpret.

### 4.6.2. Beetle-based climate reconstruction

A palaeo-temperature reconstruction was undertaken (Finningley paleo in figure 16; electronic supplementary material) using the mutual climatic range (MCR) method [126] as implemented in the BugsCEP software [89]. The MCR method reconstructs the thermal environment in which the largest

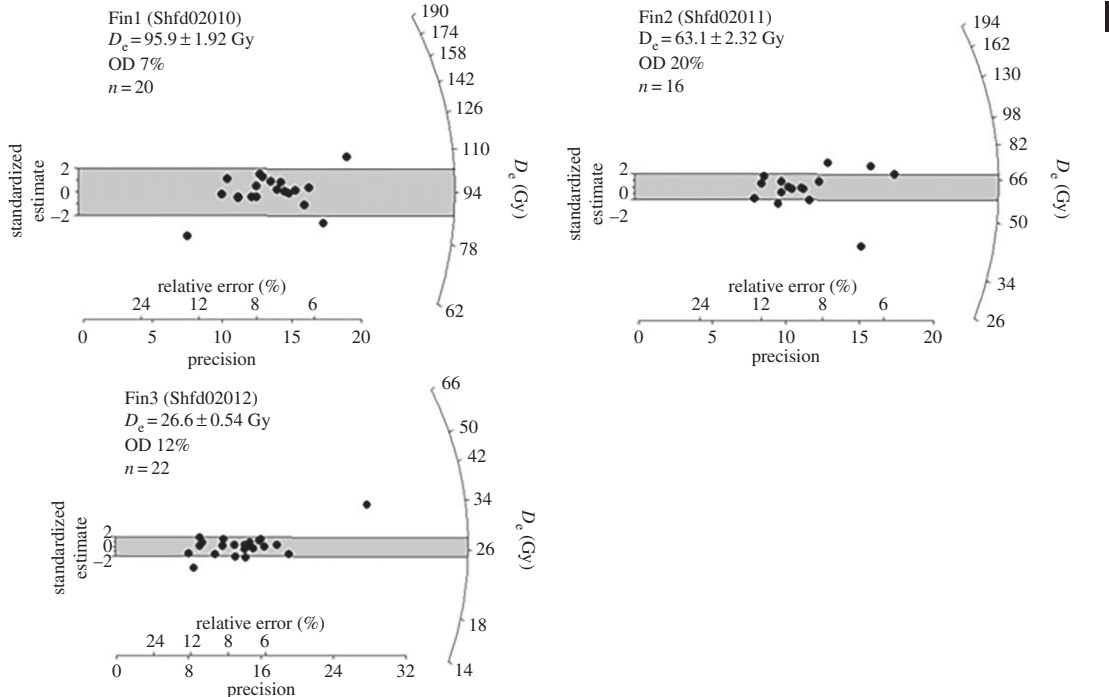

**Figure 15.** OSL Palaeodose (De) replicate measurements for samples Finn 1–3 (for location, see figure 6) showing a high degree of convergence and low OD.

**Table 4.** OSL data and ages. Ages are quoted in years from 2002 (when sampled) with 1 sigma uncertainties (for stratigraphic position, see figure 6).

| sample | unit | depth (m) | cosmic (Gy ka$^{-1}$) | dose rate (Gy ka$^{-1}$) | palaeodose (Gy) | age (ka) |
|---|---|---|---|---|---|---|
| Fin3 (Shfd02012) | 2 | 2.4 | 0.015 ± 0.0008 | 1.39 ± 0.08 | 26.6 ± 0.54 | 19.2 ± 1.1 |
| Fin2 (Shfd02011) | 2 | 3 | 0.014 ± 0.0007 | 2.70 ± 0.14 | 63.1 ± 2.3 | 23.4 ± 1.5 |
| Fin1 (Shfd02010) | 4 | 3.5 | 0.013 ± 0.0007 | 2.69 ± 0.14 | 95.9 ± 1.9 | 35.7 ± 2.0 |

proportion of species in a sample could survive, using their modern and historical ranges for reference. It allows for the calculation of TMax—mean temperature of the warmest month, TMin—mean temperature of the coldest month and TRange—difference between TMax and TMin and essentially an indication of continentality. A higher TRange suggests a more continental climate, with larger differences between summer and winter temperatures. Although previous publications have derived means and errors, or otherwise indicated mid-points for reconstructed ranges, these have been shown to be mathematically incorrect [89,127] and are thus not used here. Coleoptera over-winter as a range of different stages (egg, larval instars and/or adult, pupa), many Arctic taxa surviving a wide variety of winter conditions in diapause [86], and there are a number of winter active species reliant on snow cover [128]. However, wherever snow thickness is a few tens of centimetres or more, as is common where shrubs trap snow, Coleoptera (or anything else) over-wintering or active at or near the land surface (i.e. beneath the snow) may not provide a good indication of air temperature. Winter temperature reconstructions from any organism with restricted winter activity are thus inherently less constrained than summer temperatures, as is reflected in the reconstructions in figure 16, and calibration data for more extreme (hot or cold) temperatures in the BugsCEP software are limited [89].

Sample 3 provides an MCR reconstruction from only three species (*N. aquaticus*, *H. sibiricus* and *Ae. sabuleti*) of 8–14°C TMax and −22 to −2 TMin. This reconstruction is essentially constrained by the upper TMax and lower TRange limits of *H. sibiricus*, and the lower TMax and upper TRange limits of *A. sabuleti*; the thermal envelope for *N. aquaticus* is too poorly defined to be of use in defining the limits, but provides an additional species supporting the reconstructed temperatures. The reconstructed temperatures are compared to the modern climate at Finningley (1981–2010) (table 5) as calculated from

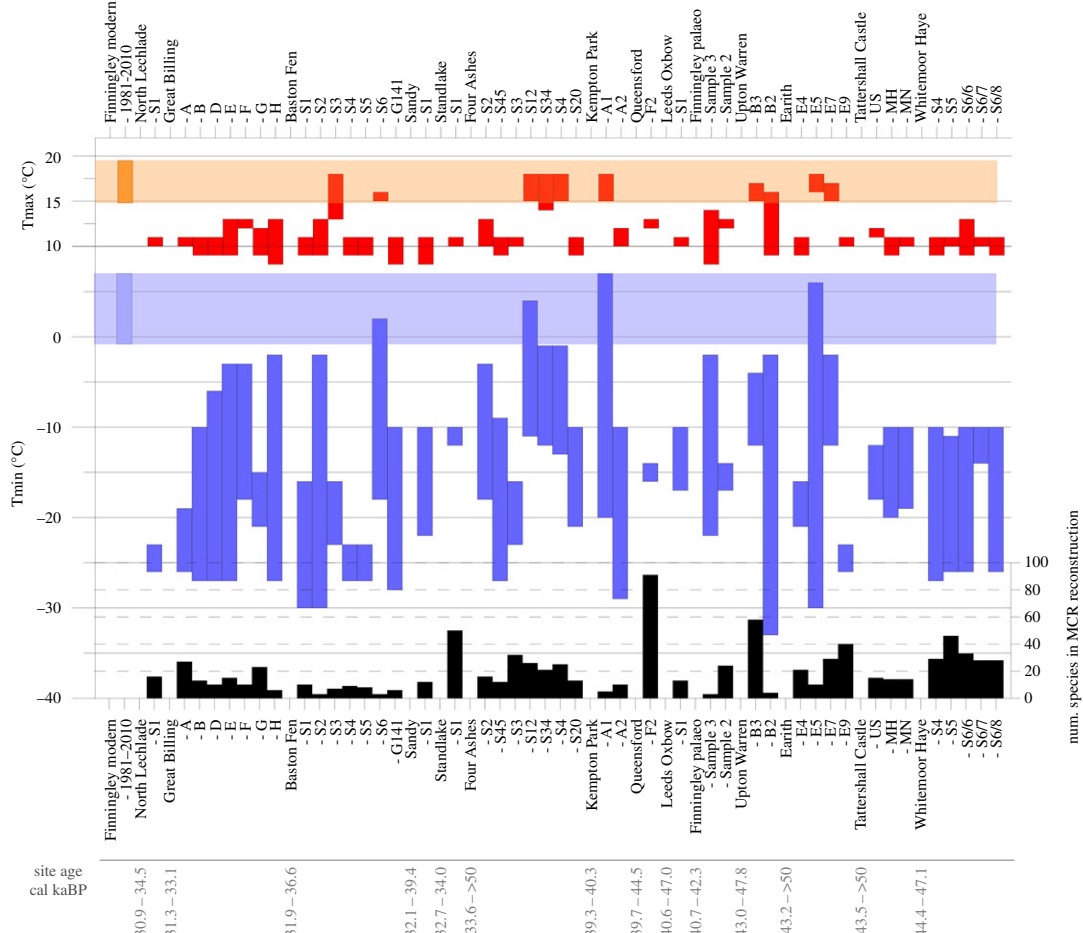

**Figure 16.** Beetle-based MCR reconstruction of the Finningley samples compared with other radiocarbon dated MIS 3 sites mentioned in the text. Modern Finningley climate is shown for comparison (leftmost bar and horizontal shaded areas). Samples with less than 3 MCR species have been omitted, but samples with poor maximum species overlaps (less than 80%) have been included to provide more samples for comparison. The lower bar chart shows the number of species used in each reconstruction. Sample names are as stored in the BugsCEP database [88]. Site ages, in calibrated ka BP, are given for either the range of dated insect samples or the overall site where these are not directly dated (see electronic supplementary material). The figure therefore does not represent a contiguous sequence. Dates calibrated with OxCal [124] using the IntCal 13 curve [125]. For sources and primary data, see electronic supplementary material, table S1.

**Table 5.** Beetle-based temperature reconstructions compared with modern climate at Finningley (1981−2010), in°C. NSPEC is the number of species in the sample with climate (MCR) data; overlap is the maximum percentage of these species with overlapping thermal envelopes, i.e. the area of climate space with maximum overlap. An ideal overlap is 100%, but large faunas or samples too coarse to resolve changing climates produce lower overlaps. Calculations using BugsCEP [88].

|  | TMaxLo | TMaxHi | TMinLo | TMinHi | TRangeLo | TRangeHi | NSPEC | overlap |
|---|---|---|---|---|---|---|---|---|
| sample 3 | 8 | 14 | −22 | −2 | 16 | 30 | 3 | 100 |
| sample 2 | 12 | 13 | −17 | −14 | 27 | 29 | 24 | 87.5 |
| modern | 14.8 | 19.5 | −0.8 | 7 | | | | |

the Climate, Hydrology and Ecology research Support System (CHESS) [129]. Sample 3 suggests that the climate at that time was considerably colder than the present day, and neither this nor the temperatures reconstructed from sample 2 overlap the equivalent values for the present, warmer climate.

Although sample 2 provides at a glance a similarly cold, but apparently more refined thermal reconstruction of 12−13°C, this represents an overlap of the thermal envelopes of only 21 (87.5%) of the 24 MCR species used in the reconstruction. Among the species most acutely responsible for the

incomplete overlap are water beetles of the *Helophorus* genus. Regarded as a pioneer species frequenting temporary pools, *H. grandis* is scarce north of southern Scandinavia [101], and appears to be restricted to relatively warm (TMax greater than 11°C), oceanic climates. *Helophorus praenanus*, on the other hand, appears to be able to cope with the variation between the extreme cold winters and mild summers of Eastern Siberia. The third species, which just fails to intersect the area of maximum overlap, is the rove beetle *Holoboreaphilus nordenskioeldi*, known only from areas with summers which get no warmer than 11°C. While there are undoubtedly improvements to be made in the primary MCR calibration data, especially for more continental species, the envelope for *Helophorus grandis* is reasonably well resolved for its colder summer limits. The thermal limits to the distribution of *H. praenanus* and *H. nordenskioeldi*, on the other hand, are poorly mapped and it could be that they overlap more than has been recorded. The same may apply to the eastern Palaearctic dung beetle, *A. jacobsoni* and other important species, such as the ladybird *C. ulkei*, yet to be included in the MCR calibration data, and thus not contributing to the reconstruction. Considering the extensive fossil record of *H. grandis* [88], and the difficulty of separating species in this group on the fossil parts, there is also the possibility that the fossil record includes both colder and warmer subspecies which have yet to be defined.

# 5. Discussion

## 5.1. Comparison of MIS 3 beetle-based climate reconstructions

Lemdahl & Coope [130] summarized the MCR results from insect faunas from Late Pleistocene sites in Europe, while Coope [29] in a more detailed examination of MIS 3 insect faunas concluded that, when correlated with the Greenland ice-core data, the preserved assemblages reflected the warmer episodes and that carrying capacity during the colder episodes was such that there were no associated organic sediments. It is debateable, however, whether the data are sufficiently robust to support this interpretation. Several of the older dates lay close to the limits of radiocarbon dating and should be treated as minima, others are bulk sample dates and the association between dated context and insect faunas in others is less than secure (see electronic supplementary material). The MIS 3 sample is further reduced by the number of species for which climate-space envelopes are available. Figure 16 provides beetle-based MCR summer (TMax) reconstructions for the majority of insect samples from MIS 3 sites in England (42 samples from radiocarbon dated sites with greater than 2 MCR species, see electronic supplementary material for details and figure S3 for a comparison of site dates with the Greenland ice-core record). The majority of summer temperatures lie between 9 and 11°C, which overlaps with the range of sample 3 from Finningley (8–14°C), but is slightly lower than that of sample 2 (12–13°C), the latter being based on more species and thus a more reliable reconstruction. The Finningley fauna most likely represents a cold stadial with both summers and winters slightly warmer than the climates indicated by the majority of the other MIS 3 samples. At least superficially, the climate is most similar to the temperature reconstruction based on the much larger, similarly dated fauna from Queensford near Dorchester on Thames (44 491–39 400 cal a BP).

This interpretation fits with the Finningley floral list, although this is more limited than from some other Middle Devensian sites in England. This is partly because it lacks southern, thermophilous elements, such as *Najas flexilis*, *Lycopus europaeus*, *Groenlandia densa* and *Scirpus lacustris*, which were found in the Middle Devensian flora at Earith locality 7 [65,131]. The occurrence of floral southern elements at some sites may reflect that the climate during MIS 3 was extremely unstable, alternating rapidly between cold stadials and warmer interstadials. Coope [75] showed that part of the plant remains from Earith came from a cold and continental episode, and another part came from a temperate and more oceanic episode. It is also possible that some remains of southern plants at British sites are reworked from layers deposited during warmer intervals. Due to the problems with dating close to the limits of radiocarbon, it is also possible that some deposits referred to the Middle Devensian in earlier publications are significantly older.

With the exception of a small number of samples showing very well constrained, and extremely cold (−27 to −23°C) winter temperature reconstructions, in particular North Lechlade, samples S4 and S5 from Baston Fen and E9 from Earith, the TMin values for Finningley agree with colder samples from other sites. Although 10 of the MIS 3 samples intersect the range of summer temperatures measured at Finningley between 1981 and 2010, only four of the corresponding winter reconstructions intersect the modern winter temperature range. Although the winter reconstructions are less well constrained, this could be tentatively interpreted as indicating that the interstadial climates represented in these

samples were significantly more continental than the present day, something which simple modern distribution data for several of the taxa would also suggest.

The MCR results for Baston Fen (insect sample AMS dates of 36 606–31 859 cal a BP) generally indicate summer and winter temperatures much colder than present day, and colder than or overlapping the Finningley reconstruction. Baston Fen's mostly high TRange values (electronic supplementary material) also suggest climates that are more continental throughout the site's range of dates. Baston S4 and S5, while showing cold summers, indicate significantly colder winters than most of the other MIS 3 samples, and at least 1°C and up to 20°C colder than Finningley. Two samples from this site (S3 and S6) suggest warmer summer conditions, equivalent to the lower range of present-day Finningley, but with overlapping winter temperatures implying a highly continental climate with cool summers and extremely cold winters for at least part of MIS 3.

Samples from Four Ashes (greater than 50 000–33 570 cal a BP) and Earith (greater than 50 000–43 168 cal a BP) provide reconstructions both at the colder end of Finningley's summers and summer temperatures within the lower span of modern variation. Potentially colder winters are also represented at these sites (Four Ashes S3 and S45, Earith E4 and E9). On the basis of the insects, these sites clearly cover both stadial and interstadial deposits, which are most likely to record colder and warmer conditions than Finningley, respectively. Interestingly, they also suggest periods of more oceanic climate, with less variation between summer and winter. The two small MCR faunas from Kempton Park (40 304–39 288 cal a BP) provide similar evidence, as do the two samples from Upton Warren (47 802–43 023 cal a BP, but see notes in electronic supplemental material).

MCR reconstructions from the beetles at Great Billing (33 069–31 348 cal a BP) are equally cold or colder than the Finningley fossils, with comparable summers and winters. Finningley overlaps with the reconstructed temperature regimes of the single sample MIS 3 sites at Leeds Oxbow (46 966–40 569 cal a BP), Sandy (39 439–32 296 cal a BP) and the abundant fauna at Queensford. The younger fauna from North Lechlade (34 586–30 886 cal a BP) suggests significantly colder winters (−27 to −23°C), and this contrasts with the slightly warmer winters (−12 to −10°C) indicated by the significant fauna from Standlake (33 975–32 673 cal a BP).

While Standlake and North Lechlade provide well-refined winter reconstructions above and below Finningley, the majority of samples from Tattershall Castle (greater than 50 000 to 43 499 cal a BP) and Whitemoor Haye (47 084 to 44 391 cal a BP) provide winter reconstructions which overlap the entire range of values. These sites all indicate similar summer temperatures, almost entirely lower than Finningley sample two (with the exception of Whitemoor Haye S6/7, which overlaps the range of values from the Finningley sample). Upton Warren was initially similarly dated to the latter two sites, but considered older by Coope ([132] but see [18,20]). It produces MCR reconstructions which tend towards present-day summer temperatures, and potentially slightly warmer winter temperatures than these sites. These values are similar to the possibly interstadial and warmer samples described for Baston Fen, Earith and Four Ashes, and several degrees warmer than Finningley.

In summary, the MCR reconstructions from Finningley are thought to record cold stadial conditions, but with milder summers and winters than those inferred from many other English beetle faunas dated to MIS 3. The overall MIS 3 beetle dataset from England indicates substantial variation in summer and winter climate during this interval (approx. 59–28 ka), consistent with independent evidence for substantial stadial–interstadial climate variability obtained from the Greenland ice-core record [6,133]. In view of the millennial timescales of such variability and the uncertainties of dating terrestrial sites at or near the limit of radiocarbon, it is clear the correlations even between terrestrial MIS 3 sites in England are challenging, and terrestial–ice-core correlations remain speculative. Coope's [134] comment that episodes of gravel deposition in many sequences may reflect even colder conditions when carrying capacity was reduced such that there was little contemporary biota remains valid.

## 5.2. Periglacial and proglacial lake environments

Evidence for periglaciation at Finningley is broadly consistent with that from other lowland regions of England and the near continent. The Finningley sequence records a change from permafrost conditions associated with deposition of the basal unit of sand and gravel to milder periglacial conditions during deposition of the overlying infilled channel.

Wedge structures similar to those in the basal unit have been reported from gravelly sequences of Pleistocene age in lowland England (see reviews in [135, pp. 53–63,123]). But the exact age of the gravel and wedges at Finningley is uncertain, precluding correlation with the British periglacial record. A minimum age is provided by the calibrated radiocarbon age of 39 933 ± 381 cal. years BP

from the overlying organic silt, but this study provides no maximum age. Thus, gravel deposition during permafrost conditions may have occurred during, for example, an MIS 3 stadial before approximately 40 ka or cold-climate conditions of MIS 4 or 6. Interestingly, there are similarities in terms of relatively small wedge size, intraformational nature and host stratified gravels with wedges and gravel of facies assemblage 1 at Baston, Lincolnshire, which have been dated by radiocarbon and OSL to the Middle Devensian [74]. If the wedges and gravel at Finningley are from a Middle Devensian stadial, one possibility is the Hasselo Stadial, when permafrost is known to have developed—inferred partly from ice-wedge pseudomorphs—in the Netherlands. The Hasselo Stadial has provided radiocarbon ages of 43 220–42 290 cal. BP [136]. These authors suggested that the subsequent Hengelo Interstadial (42 350–41 380 cal. BP) caused permafrost degradation and formation of involutions.

The sand and organic silt with a depositional age of approximately 40–35 ka infilling the river channel at Finningley are of similar age and environmental context to many Middle Devensian periglacial river deposits in lowland England (reviewed in [137, pp. 162–171, 135, pp. 58–61, 19, p. 452]). Contemporary rivers in lowland England were mainly of braided type and deposited gravel as a result of the supply of abundant coarse sediment from mass wasting on adjacent hillslopes [138]. A highly variable nival discharge regime favoured reworking of solifluction deposits during snowmelt floods, winnowing out the fine-grained sediment and leaving behind the gravel. Finer deposits exposed on floodplains and eroded from pre-existing deposits were vulnerable to reworking by wind and water, which deposited sediments such as coversand by wind action (e.g. [12]) and laminated silt and sand by aeolian rainout and sheetwash [139]. Palaeoecological evidence associated with the Middle Devensian river gravels in eastern England [74,140] indicates wetter conditions at the time of gravel deposition and hence more active river systems than those during the LGM. In summary, Middle Devensian periglacial environments, at least in southeast and eastern England, are thought to have been warmer and wetter than the cold and arid conditions of the Early and Late Devensian.

Both units 4 and 5 are interpreted as indicating the presence in the Finningley area of Lake Humber. The coarser sands and gravels of unit 4 are taken to indicate that in the period $23.4 \pm 1.5$ to $19.2 \pm 1.1$ ka, Finningley was close to the margins of this lake. Unit 5 is interpreted as indicating deeper water at Finningley occurred sometimes after $19.2 \pm 1.1$ ka. Such ages are not out of line with those presented from wave-rippled silty sand at Hemingbrough by Murton *et al.* [37], or the basal age Bateman *et al.* [9] reported from laminated clays from Lake Humber. Given that ice had to block the Humber Gap for Lake Humber to form, this also indicates the movement of the North Sea Ice lobe onshore at this time, something confirmed by recent glacial ages from Ferriby [39]. The low-lying nature of the Finningley site cannot add information as to the absolute lake levels as the sediments themselves occur at between 6 and 8 m OD. However, the switch from units 4 to 5 may indicate that an earlier, lower and more fluctuating Lake Humber preceded the more widespread and deeper Lake Humber associated with the laminated clays.

# 6. Conclusion

The new work shows that the Finningley sequence records a change from permafrost conditions to milder periglacial conditions during deposition of the infilled channel. Unit 1, at the base, is tentatively correlated with the ORG of Gaunt [22], and may represent more than one glacial–interglacial cycle. The organic silt in unit 2 dated to 40 314–39 552 cal a BP (MIS 3) contains Arctic flora and fauna indicative of deposition in a cold, continental arctic regime. The mean temperature of the warmest month is reconstructed—from beetle-based MCR data—as 8–14°C and, with less certainty, that of the coldest month as $-22$ to $-2$°C . The channel is sealed by a diamict (unit 3) derived from a low ridge of glaciofluvial deposits to the west. Unit 4, largely thin-bedded sands with some clay–silts, is the equivalent of Gaunt's [22] Littoral Sands and Gravels. Unit 5, beneath largely desiccated Holocene peat, consists of the laminated clay–silts of the Hemingbrough Formation, deposited at depth into the proglacial Lake Humber during the LGM.

Overall, during MIS 3 (59–28 ka), England appears to have experienced substantial variation in summer and winter climate. This is consistent with the Greenland ice-core record. Comparison of this new proxy data from the temporally well-constrained Finningley site with other MIS 3 records in England suggests that around 40 ka was a prominent cold stadial. However, difficulties of dating and correlation of most MIS 3 sites within England make it challenging to correlate with certainty to the Greenland ice-core record. At present, the stadial could be correlated with any of the Greenland stadials between 7 and 11.

Data accessibility. The datasets supporting this article have been uploaded as part of the electronic supplementary material. The new fossil insect data upon which this paper is based are provided as electronic supplementary material as well as being deposited in the freely available BugsCEP database (www.bugscep.com) and SEAD database (www.sead.se) on publication. DOI's and upload to the Neotoma repository (www.neotomadb.org) are available from 2020 for these resources. The compilation of MIS 3 dates is provided as electronic supplementary material, including as much metadata for each date as was obtained through our literature search. The BugsCEP database and software includes all of the MIS 3 beetle data that were used to undertake the climate reconstruction of comparative sites. BugsCEP's built in mutual climatic range (MCR) routines were used to calculate all beetle-based climate reconstructions presented. The software is downloadable as an MS Access 2000 database and all code is openly accessible in this file.

Authors' contributions. P.I.B. undertook the beetle-based environmental and climate reconstruction work, co-compiled the comparative sites data and contributed extensively to the text. All other authors contributed equally to the analysis and writing, and are listed in alphabetical order: M.D.B. undertook the OSL dating and interpretation, and contributed to the stratigraphic interpretation. P.C.B. identified the Coleoptera remains, coordinated the collaboration and writing, and contributed to site interpretation. O.B. identified and interpreted the plant macrofossil remains. B.M.C. initiated the fieldwork and interpreted the regional context of the sequence. C.F. recorded the sections and provided additional site data and photographs. M.G. identified and the interpreted the Trichoptera remains. J.M. carried out additional fieldwork on the site and contributed to site interpretation, particularly the fine-grained stratigraphy. D.M. carried out additional fieldwork on the site and contributed to site interpretation. E.P. sampled the organic sediments and identified the Dipterous remains.

Competing interests. We have no competing interests.

Funding. The lead author's research time was funded by Umeå University Faculty of Arts and Humanities.

Acknowledgements. Access to the site was kindly provided by the successive quarry managers of LaFarge/Tarmac. MSc students from Sheffield University helped clear sections. Laura Trinogga at the Doncaster Museum and Dmitri Logunov at the Manchester Museum are thanked for access to insect collections. Geoff Gaunt and members of the QRA fieldtrip 2001 provided stimulating discussions which helped shape this manuscript. Robert Angus kindly confirmed the identification of *Helophorus praenanus* and Tim Prosser provided the modern climate information. The authors wish to acknowledge the help given by Peter Neu (Germany), Peter Wiberg-Larsen (Denmark), Aki Rinni (Finland), Peter Barnard (UK) and Ian Wallace (UK) for Trichopteran reference specimens, distribution data and taxonomic expertise and Colin Howes for drawing our attention to the vertebrate remains. Ben Price (Natural History Museum, London) provided access to Imaging Facility and Mark Szegner Loughborough University) collate figures 13 and 14. We thank Dr Ian S. Evans and an anonymous reviewer for their careful reading of the manuscript and constructive remarks.

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
