## [Reviewer comments · Royal Society Open Science]

Review History

RSOS-190577.R0 (Original submission)

Review form: Reviewer 1

Is the manuscript scientifically sound in its present form?

Yes

Are the interpretations and conclusions justified by the results?

Yes

Is the language acceptable?

Yes

Is it clear how to access all supporting data?

Not Applicable

Do you have any ethical concerns with this paper?

No

Have you any concerns about statistical analyses in this paper?

No

Recommendation?

Accept with minor revision (please list in comments)

Comments to the Author(s)

This is basically a good paper that adds significantly to the corpus of data concerning the age and environment of deposition of the Older River Gravels, which are an important unit in the Middle to Late Pleistocene regional development of GB. My comments are all very minor and perhaps more food for thought. They are annotated on the attached manuscript (Appendix A).

Review form: Reviewer 2 (Ian Evans)

Is the manuscript scientifically sound in its present form?

Yes

Are the interpretations and conclusions justified by the results?

Yes

Is the language acceptable?

Yes

Is it clear how to access all supporting data?

Yes

Do you have any ethical concerns with this paper?

No

Have you any concerns about statistical analyses in this paper?

No

Recommendation?

Accept with minor revision (please list in comments)

Comments to the Author(s)

Comments by Dr Ian S. Evans, Durham University, on Buckland et al.,
Mid to Late Devensian landscape change in England.

A thorough analysis of stratigraphic and palaeontological/palaeoecological observations from this important site in eastern England is provided by a group of experienced scientists. Observations over a number of years as the quarry face progressed improve the reliability of interpretations. The numerous illustrations are well drawn and relevant: the photos are well selected.

The section 6.1 Discussion usefully compares the temperature ranges (summer and winter) implied at Finningley with those from other MIS 3 sites. Using the Greenland record, the authors correctly state that interstadials and stadials alternated within MIS 3, yet it is difficult to follow the implications of the various dates and temperatures in the text which (despite the conclusion)

reads as if there is some expectation of uniform conditions throughout MIS 3. The dates tabulated below Fig. 16 do not put things into chronological order. Section 6.1 would be aided by a summary temperature: chronology Figure plotting each observation as a box, with its range of calibrated dates and its range of T_{min} and (separately) T_{max}. (There are 13 date ranges mentioned in the text, i.e. 13 boxes.) The text could then be shortened and clarified.

I think the title could be improved: it does not cover England and there is little on Late Devensian. I suggest 'Mid Devensian climate and landscape in eastern England' would be more accurate as the first sentence of the title.

Readers are expected to be familiar with detailed terminology in several fields: sedimentology, botany and zoology. Possibly terms such as 'flaser' and 'stenotherm' might be defined.

Below I suggest a number of corrections and some cases where the expression can be polished. There is a tendency to use too many commas, making complicated sentences with too many qualifications: some could be split, e.g. lines 18-22 and 30-33 on page 6.

DETAILS:

Page 1: Line 39 'seek to' is anthropomorphic. 'data can ...'?

52 drop one of the 2 boths

55 perhaps 'Britain' (or even 'England') rather than UK?

Page 2: Line 1 'showed'

2 'the limit'

18 'setting, in'

34 'deep-water' rather than profundal

41 'from both'

43 delete comma

Page 3: 13 'were collected'

13-14 delete present commas: insert comma before 'and'.

14 'was revealed'

18 'was placed'

20 delete second 'then'

25 'that for'

29 Should Stratigraphy be section 5.1? – and later sections renumbered.

31 delete first comma

49 delete first comma

Page 4: 21 'which are interpreted'

Page 5: 2 'river Idle. It is interpreted as being'

11 'non-preservation' !

48 'whereas others indicate'

Page 6: 5.3.1 is italicised, unlike any later subsection headings...

2 ':' rather than ';

6 'appear'

16 One comma too many! Delete last, '[82] and occurring'.

26 'suggested'

41 Give name before [99] ?

Page 7: 28 remove comma. Split the following sentence?

30 'it is found'

39 delete last comma

39-40 "in papers published in" makes no sense, as the three following citations are normal papers, not collections.

40 replace ';' with ','

44 'BP at Clettnadal'

54 'arctic-equivalent'

Page 8: 13 'between ... and'

58 'glance a similarly'

Page 9: 11 6. Not 6.6.

16-17 past tense, especially for 2001 !

56 summer-winter-summer?? Rewrite sentence.

Page 10: 1 The sentence ending 'Queensford.' is a non-sentence. It should be combined with the following, or re-written.

8: Replace second ';' with 'and'. This is another clumsy sentence.

24: 'near continent'

39/40 ';' or '' ?

Page 11: top - excess '11'

11 'suggests that around'

14 'and 11'

Page 14: 26-27 Why is 'Published for the Quaternary Research Association' stated here? It is not stated elsewhere JQS is cited. Also, italicise journal title.

Decision letter (RSOS-190577.R0)

30-May-2019

Dear Dr Buckland

On behalf of the Editors, I am pleased to inform you that your Manuscript RSOS-190577 entitled "Mid to Late Devensian landscape change in England. New data from Finningley, South Yorkshire." has been accepted for publication in Royal Society Open Science subject to minor revision in accordance with the referee suggestions. Please find the referees' comments at the end of this email.

The reviewers and handling editors have recommended publication, but also suggest some minor revisions to your manuscript. Therefore, I invite you to respond to the comments and revise your manuscript.

- Ethics statement

- Data accessibility

<http://datadryad.org/submit?journalID=RSOS&manu=RSOS-190577>

- Competing interests

- Authors' contributions

- Acknowledgements

- Funding statement

Because the schedule for publication is very tight, it is a condition of publication that you submit the revised version of your manuscript before 08-Jun-2019. Please note that the revision deadline will expire at 00.00am on this date. If you do not think you will be able to meet this date please let me know immediately.

Kind regards,
Alice Power
Royal Society Open Science
openscience@royalsociety.org

on behalf of Dr Bethan Davies (Associate Editor) and Jon Blundy (Subject Editor)

Reviewer comments to Author:

Reviewer: 1

This is basically a good paper that adds significantly to the corpus of data concerning the age and environment of deposition of the Older River Gravels, which are an important unit in the Middle to Late Pleistocene regional development of GB. My comments are all very minor and perhaps more food for thought. They are annotated on the attached manuscript.

Reviewer: 2

Comments by Dr Ian S. Evans, Durham University, on Buckland et al.,
Mid to Late Devensian landscape change in England.

A thorough analysis of stratigraphic and palaeontological/palaeoecological observations from this important site in eastern England is provided by a group of experienced scientists. Observations over a number of years as the quarry face progressed improve the reliability of interpretations. The numerous illustrations are well drawn and relevant: the photos are well selected.

The section 6.1 Discussion usefully compares the temperature ranges (summer and winter) implied at Finningley with those from other MIS 3 sites. Using the Greenland record, the authors correctly state that interstadials and stadials alternated within MIS 3, yet it is difficult to follow the implications of the various dates and temperatures in the text which (despite the conclusion) reads as if there is some expectation of uniform conditions throughout MIS 3. The dates tabulated below Fig. 16 do not put things into chronological order. Section 6.1 would be aided by a summary temperature: chronology Figure plotting each observation as a box, with its range of calibrated dates and its range of T_{min} and (separately) T_{max}. (There are 13 date ranges mentioned in the text, i.e. 13 boxes.) The text could then be shortened and clarified.

I think the title could be improved: it does not cover England and there is little on Late Devensian. I suggest 'Mid Devensian climate and landscape in eastern England' would be more accurate as the first sentence of the title.

Readers are expected to be familiar with detailed terminology in several fields: sedimentology, botany and zoology. Possibly terms such as 'flaser' and 'stenotherm' might be defined. Below I suggest a number of corrections and some cases where the expression can be polished. There is a tendency to use too many commas, making complicated sentences with too many qualifications: some could be split, e.g. lines 18-22 and 30-33 on page 6.

DETAILS:

Page 1: Line 39 'seek to' is anthropomorphic. 'data can ...'?

52 drop one of the 2 boths

55 perhaps 'Britain' (or even 'England') rather than UK?

Page 2: Line 1 'showed'

2 'the limit'

18 'setting, in'

34 'deep-water' rather than profundal

41 'from both'

43 delete comma

Page 3: 13 'were collected'

13-14 delete present commas: insert comma before 'and'.

14 'was revealed'

18 'was placed'

20 delete second 'then'

25 'that for'

29 Should Stratigraphy be section 5.1? – and later sections renumbered.

31 delete first comma

49 delete first comma

Page 4: 21 'which are interpreted'

Page 5: 2 'river Idle. It is interpreted as being'

11 'non-preservation' !

48 'whereas others indicate'

Page 6: 5.3.1 is italicised, unlike any later subsection headings...

2 ':' rather than ','

6 'appear'

16 One comma too many! Delete last, '[82] and occurring'.

26 'suggested'

41 Give name before [99] ?

Page 7: 28 remove comma. Split the following sentence?

30 'it is found'

39 delete last comma

39-40 "in papers published in" makes no sense, as the three following citations are normal papers, not collections.

40 replace ';' with ','

44 'BP at Clettnadal'

54 'arctic-equivalent'

Page 8: 13 'between ... and'

58 'glance a similarly'

Page 9: 11 6. Not 6.6.

16-17 past tense, especially for 2001 !

56 summer-winter-summer?? Rewrite sentence.

Page 10: 1 The sentence ending 'Queensford.' is a non-sentence. It should be combined with the following, or re-written.

8: Replace second ';' with 'and'. This is another clumsy sentence.

24: 'near continent'

39/40 ';' or ',' ?

Page 11: top - excess '11'

11 'suggests that around'

14 'and 11'

Page 14: 26-27 Why is 'Published for the Quaternary Research Association' stated here? It is not stated elsewhere JQS is cited. Also, italicise journal title.

Author's Response to Decision Letter for (RSOS-190577.R0)

See Appendix B.

Decision letter (RSOS-190577.R1)

11-Jun-2019

Dear Dr Buckland,

I am pleased to inform you that your manuscript entitled "Mid to Late Devensian landscape change in England. New data from Finningley, South Yorkshire." is now accepted for publication in Royal Society Open Science.

on behalf of Dr Bethan Davies (Associate Editor) and Jon Blundy (Subject Editor)
openscience@royalsociety.org

Follow Royal Society Publishing on Twitter: [@RSocPublishing](https://twitter.com/RSocPublishing)

Appendix A**ROYAL SOCIETY
OPEN SCIENCE****Mid to Late Devensian landscape change in England. New
data from Finningley, South Yorkshire.**

Journal:	Royal Society Open Science
Manuscript ID	RSOS-190577
Article Type:	Research
Date Submitted by the Author:	02-Apr-2019
Complete List of Authors:	Buckland, Philip; Umeå Universitet Institutionen för idé- och samhällsstudier, Environmental Archaeology Lab Bateman, Mark; University of Sheffield, Geography Bennike, Ole; Geological Survey of Denmark and Greenland, GEUS Buckland, Paul; Consultant Palaeoecologist Chase, Brian; University of Cape Town, Environmental & Geographical Science Frederick, Charles; Consulting Geoarchaeologist Greenwood, Malcolm; Loughborough University, Geography Murton, Julian; University of Sussex, Geography Murton, Della; Cambridge University, Zoology Panagiotakopulu, Eva; University of Edinburgh, Institute of Geography
Subject:	Geology < EARTH SCIENCES, Climatology < EARTH SCIENCES, Palaeontology < EARTH SCIENCES
Keywords:	Mid-Devensian, Vale of York, Plant Macrofossils, Trichoptera, Coleoptera, Periglaciation
Subject Category:	Earth science

Mid to Late Devensian landscape change in England. New data from Finningley, South Yorkshire.

Philip Buckland^{1*†}, Mark D Bateman², Ole Bennike³, Paul C Buckland⁴, Brian Chase⁵, Charles Frederick⁶, Malcolm Greenwood⁷, Julian Murton⁸, Della Murton⁹, Eva Panagiotakopulu¹⁰

1 Environmental Archaeology Lab., Umeå University, Umeå, SE-901 87, Sweden

2 Department of Geography, University of Sheffield, Sheffield, S10 2TN, UK

3 GEUS Geological Survey of Denmark and Greenland, ØsterVoldgade 10, Copenhagen, 1350, Denmark

4 20 Den Bank Close, Sheffield, S10 5PA, UK

5 Environmental & Geographical Science Building, South Lane, Upper Campus, University of Cape Town,

Private Bag X3, Rondebosch 7701, South Africa

6 Consulting Geoarchaeologist, 2901 FM 1496, Dublin, TX-76446, Texas, USA

7 Dept. of Geography, Loughborough University, Leics., LE11 3TU, UK

8 Dept. of Geography, University of Sussex, Brighton, BN1 9RH, UK

9 Dept. Of Zoology, University of Cambridge, Downing Street, Cambridge, CB2 3EJ, UK

10 Institute of Geography, University of Edinburgh, Drummond Street, Edinburgh,

EH8 9XP, UK

Keywords: Mid-Devensian; Vale of York; Plant Macrofossils; Trichoptera; Coleoptera; Periglaciation.

1. Summary

Whilst there is extensive evidence for the Late Devensian, less is known about Early and Middle Devensian (~110 to 30 ka) climates and environments in the UK. The Greenland ice-core record suggests the UK should have endured multiple changes but the terrestrial palaeo-record lacks sufficient detail for confirmation. Data from Finningley, South Yorkshire, seek to redress this. A channel with organic silts, dated 40,314 - 39,552 cal a BP, contained plant macrofossil and insect remains showing tundra with dwarf-shrub heath and bare ground. Soil moisture conditions varied from free draining to riparian, with ponds and wetter vegetated areas. The climate was probably low-arctic with snow cover during the winter. Mutual Climatic Range, based on Coleoptera, shows mean monthly winter temperatures of -22 to -2°C and summer ones of 8 to 14 °C. Periglacial structures within the basal gravel deposits indicate cold-climate conditions, including permafrost.

A compilation of MCR reconstructions for other Middle Devensian British sites shows MIS 3 between 59-28 ka to include substantial variation in climate consistent with the Greenland ice-core record. Exact correlation is hampered by temporal resolution but the Finningley site stadial at ~40 ka may correlate with one of the Greenland Stadials 7 to 11.

2. Introduction

During the Late Quaternary the UK was subjected to multiple cold stages, leading to both periglaciation and glaciation, both of which sculpted the landscape. Recent research has started to address systematically the pattern and chronology of the British and Irish Ice Sheet (BIIS), particularly during the Last Glacial Maximum (LGM, ~21 ka) and Lateglacial (e.g. [1,2,3]). During the LGM (although not necessarily synchronously) the BIIS extended to the Scilly Isles [4] in the south-west of the UK and to Norfolk in the south-east (e.g. [5]). Less is known about the climate and environments in the UK Early and Middle Devensian (~110 to 30 ka). Rasmussen *et al.* [6], based on Greenland ice-core records, show 25 stadials and 24 interstadials between 110 ka and the start of the Holocene (~11.7 ka) but the extent and duration of Devensian glacial ice in the UK prior to the LGM is less certain. Whilst Straw [7,8] has argued for two distinct Devensian advances

*Author for correspondence (Philip.buckland@umu.se).

†Present address: Environmental Archaeology Lab., Umeå University, Umeå, SE-901 87, Sweden

of ice as far south as Lincolnshire and Yorkshire (figure 1), one LGM and another at some time pre-LGM, evidence for this was not found by [9] in Yorkshire or Lincolnshire, where the two Marsh tills of Straw were reinterpreted as two oscillations within the LGM ice advance. Carr *et al.* [10] show extensive ice across northern England and Scotland during the Ferder episode ~70 ka in Marine Isotope Stage 4 (MIS), although this limit is not well constrained by evidence. Clark *et al.* [2] show that by 27 ka (MIS 2) once again much of northern England and Scotland were covered by ice, suggesting that initiation must have occurred sometime before that in MIS 3. Roberts *et al.* [5] showed ice advanced southward down the North Sea to Dogger Bank, withdrew by ~23.1 ka before advancing to its maximal position on the Norfolk coast ~22.8–21.5 ka. During the Devensian, periglacialiation took place beyond the ice limits, with regional patterned ground and involutions dated to 60–55 ka (MIS 3), 35–31 ka (MIS 3), 22–20 ka (MIS 2) and 12–11 ka in eastern England [11]. The Devensian cold periods were interspersed with warmer interstadials [12, table 3.4], most notably the Chelford Interstadial (MIS 5c, ~108–92 ka [13,14,15]), the Brimpton (MIS 5a, ~86–72 ka [16,17]) and the Upton Warren (early during MIS 3 [18]; possibly ~42.5–38.5 ¹⁴C ka, according to Catt *et al.* [19]). However, the chronologies, environments and climatic conditions associated with these interstadials remain not well understood. For example, aminostratigraphy suggests that the Upton Warren Interstadial significantly pre-dates MIS 3 [20]. In summary, whilst the Greenland ice-core record would suggest the UK landscape should have endured multiple climatic and environmental changes, at present studies of the terrestrial palaeo-record lack sufficient detail to confirm this.

This study reports observations from one of the UK's main onshore sedimentary basins which contains Pleistocene deposits laid down in a proglacial setting in order to elucidate further the Middle to Late Devensian terrestrial palaeo-record and landscape change in eastern England. The study takes a detailed multi-proxy approach to both sediments and preserved fossils found at a site close to the Late Devensian ice limit at Finningley, South Yorkshire. Results are evaluated in the context of other dated Middle Devensian sites for which palaeoclimatic or palaeoenvironmental information exists.

3. Regional Setting (figure 1)

East of Doncaster the Sherwood Sandstone Group dips gently beneath the Quaternary deposits of the Humberhead Levels. The detailed mapping by Gaunt [21,22,23] provide the basis for a regional Quaternary stratigraphy (figure 2), with the oldest deposits being ascribed to the Anglian (MIS 12) represented by heavily cryoturbated sands and gravels found on the north–south ridge between Bawtry and Rossington (cf. [24,25]). The overlying Older River Gravel (ORG; [26,22]) forms a gently undulating surface sloping northwards from Austerfield through to Hatfield, where it meets a similar fan of gravels relating to the Don drainage [22, figure 41]. A flora and insect fauna from within the ORG at Austerfield indicates a MIS 5e age [27], although Coope [28] raises the possibility of a MIS 9 age. It seems probable that the ORG may represent deposits from several glacial–interglacial cycles.

More recent general reviews of the regional Quaternary stratigraphy are provided in Bateman *et al.* [29] and Gaunt *et al.* [30] but eastwards and northwards these deposits are overlain by those of the ‘25 ft Drift,’ littoral sands and profundal clay–silts of proglacial Lake Humber. (Hemingbrough Glaciolacustrine Formation; [31]). Lake Humber is thought to have formed in the Vale of York when North Sea ice blocking the Humber Gap at Ferriby. Gaunt [21, 22] interpreted the regional stratigraphy in terms of a two-stage model, with shorelines at approximately 33m and 7m (25 ft Drift). Whilst the existence of a high level lake has been challenged [32, 33, 34], its transient nature, leading to limited deep-water deposition, has been stressed by Bateman *et al.* [35] and its relationship with a short-lived extension of a western tongue of the Vale of York glacier has recently been considered by Friend *et al.* [36]. Recent optically stimulated luminescence (OSL) ages both from lacustrine and regional glacial sediments [35, 37, 38, 39, 40] indicate a LGM age for both phases, although the sequence appears more complex than Gaunt [22] originally suggested [41, 42].

The quarry at Finningley, lies east of the Finningley to Austerfield road at Crow Wood (figure 3 A & B; Lat. 53° 28' 20"N; Long 00 59' 02" W). It was worked as a series of west–east faces (figure 3 and figure 4). The relationship between these small outcrops, the ORG and 25 ft Drift was progressively revealed in 2001–7 as quarrying proceeded southwards (figure 4).

4. Methods

4.1. Section logging, sediment characterisation and dating

A number of vertical sections were cleaned back, recorded and scaled photomontages prepared (figure 5 and figure 6; Supplementary material figs 1 and 2). Included on these were sediment texture and colour (using Munsell colours), primary and secondary sedimentary structures, bed contacts, sediment body geometry as well as observations on clast form and lithology.

To support this work, key units were sampled for particle-size analysis using a CILAS 940 laser diffraction particle analyser. For each measurement, samples retaining their field moisture were dispersed in de-ionised water and underwent ultrasound prior to measurement. Results were used to determine mean particle size and sorting as per Gale and Hoare [43].

Samples for OSL dating were collected by hammering lightproof PVC tubes into freshly cleaned exposures. In the laboratory, the samples were prepared to clean and extract quartz as per Bateman and Catt [44], using the grain-size

fractions 90–180 μm . Samples were measured in a Risø DA-12 luminescence reader. Equivalent doses (D_e) were determined using the SAR protocol [45] with an experimentally determined preheat of 240°C for 10 s. Results show rapid decay of OSL dominated by a fast component and good growth with laboratory dose. Replicate Palaeodose (D_e) values for sample Shfd02010 had a very low over-dispersion (7%) and so the D_e value used for age calculation was based on the Common Age Model [46]. Dosimetry was calculated via ICP with elemental concentrations converted to dose using data from Guerin *et al.* [47], taking into account attenuation factors relating to sediment grain sizes used, density and palaeomoisture. The latter was set at 23±5% based on estimations of sediment saturation potential, and assumptions of groundwater levels based on the region's palaeoenvironmental history as is presently understood. The Prescott and Hutton [48] algorithm was used to calculate the cosmogenic derived dose rate.

4.2. Organic lens

Within a series of cut-and-fill channels sealed by a diamict (unit 3, see below) occurred occasional lenses of dark brown organic silt. Three 3 litre samples were processed for plant and animal macrofossils. Two (S1–2), collected in 2001, from a channel (figure 7), close to the western edge of the pit and another (S3) revealed in 2006 further to the northeast. The latter was recorded in a section from which any overlying Lake Humber clay–silts had been removed prior to excavation of sand and gravel deposits; both appear to relate to pools on an aggrading floodplain with flow in a northeasterly direction.

Samples of the organic silts were placed directly into polythene bags and sealed. Each was disaggregated in water over a 300 μm sieve and the residue retained on the mesh placed in a bowl and paraffin (kerosene) added. This adsorbs onto the surface of insect remains, and when water is added the light oil and insects float and can be decanted off. The float was then washed in detergent to remove the paraffin, cleaned with methanol and then sorted under a low-power binocular microscope (cf. [49]). Preservation in Sample 1 was insufficient to warrant further work and Samples 2 and 3 were utilised for plant macrofossil and insect identification. As the amount of plant material was slight, only a few tens of grams, the material was recombined and the easily identified seeds of crowberry, *Empetrum nigrum/hermaphroditum*, were separated for material for accelerator mass spectrometry (AMS) dating. Taxonomy for plant macrofossils follows The Plant List [50], for Coleoptera Böhme [51] and Trichoptera Graf *et al.* [52].

5. Results Stratigraphy

At the west end of the pit (~4 m OD), a north–south section, parallel to and close to the Finningley to Austerfield road, exposed over 3m of sands and gravels, shallowly cross-bedded to the north (1 on figure 3 B & figure 5). The series of west-east sections exposed as quarrying progressed southwards (2 & 3 on figure 3 B & figure 4) provided more complete sequences. The stratigraphic sequence (figure 6) is numbered and described from the quarry floor upwards:

Unit 1 sand and gravel

Unit 1, as exposed, comprised at least 2 m of reddish yellow (7.5YR 6/6) sands and gravels (Gm) directly overlying the sub-horizontal eroded upper surface of the Sherwood Sandstone (figure 6). This unit is massive, although occasionally stratified to crudely stratified with localised cross-beds. Clasts comprising rounded to sub-rounded quartzite pebbles and a few cobbles with some imbrication were observed where not disturbed (see below). The unit appears completely decalcified and the upper part is penetrated to 1.5 m by involutions, festoons and wedge features. Ice-wedge pseudomorphs up to 28 cm wide were infilled with sandy gravel or open-work gravel with vertically aligned elongate pebbles (figure 8). The downturned beds of the host material suggest that this fill was secondary, once ice had melted. Given that the wedge tops are found within unit 1, these ice-wedge pseudomorphs are considered intraformational. This unit is interpreted as being the ORG of Gaunt [22].

Unit 2 Infilled channel

Unit 2 has been eroded into the underlying sands and gravels of unit 1, forming an infilled channel (figure 7). Within this channel was a series of cut-and-fill structures, sometimes bedded lenses of sandy gravel, sands (Gms, brown, 7.5YR 4/5) and clay–silts (Fm, dark greyish brown, 10YR 4/2). Of particular note was a 25 cm thick lens of thin-bedded slightly contorted, organic silt with sandy partings (figure 7) which was sampled for plant macrofossils, insects and radiocarbon dating (samples 1–2).

Given that the channel infill contains gravels through to fine-grained material, unit 2 is interpreted as indicating a range of fluvial flow regimes with the clay and organic lenses representing low flow, possibly in overbank or channel marginal settings. Whilst the contortion of the organic and clay lenses could be due to cryoturbation, it probably results from post-depositional de-watering. The unit indicates multiple cycles of cutting and infilling of small channels which switched position through time, typical of a braided-stream environment with peaked flows associated with a periglacial environment.

Unit 3 diamict

Unit 3 comprises a massive 1–1.5 m thick matrix-supported poorly sorted silty-clay diamict (Dmm, on average 50% gravel, 25% sand, 25% silt–clay). This showed considerable lateral variability in colour and texture. Where the deposit sits on higher ground, it is thinner and more heavily involuted, appearing more red (strong brown 7.5YR 4/6). Where the diamict sits on sandier facies of the underlying sands and gravels it too is sandier (strong brown 7.5YR 5/6), and where sitting on gravels it is more clast-rich. In places the uppermost part of this unit appears laminated and contains occasional

dropstones. Sections showed some discontinuous traces of bedding and deformed U-shaped lenses of dark greyish brown (10YR4/2) silty sand or greyish-brown (10YR 5/2) silty-clay lenses aligned parallel or sub-parallel to the lower contact. The clasts, up to cobble size, are dominated by sub-rounded to rounded quartzites ultimately derived from the Sherwood Sandstone. The lower boundary of the unit, where it overlies unit 1, appears sharp, erosive and irregular, infilling low points in the underlying unit. Flame structures of sand and gravel extend up to 50 cm into the diamict in variable directions (figure 9). Where unit 3 overlies unit 2 folds and sheared bodies of the silty sand within unit 2 were observed. The upper boundary is also irregular with apparent downward intrusions of sand (figure 9).

The incorporation of rounded quartzite clasts is interpreted as indicating reworking of unit 1 and therefore a local rather than regional derivation for this diamict. Given the apparent limits of the LGM ice in the Humberhead Levels to the north (Friend *et al.* 2016) [36], a glacial origin is improbable, leaving either solifluction or a sub-aqueous mass flow similar to that inferred in Klassen & Murton [53] as possibilities. The absence of erratics, beyond those derived from either the ORG or underlying Fluvio-Glacial Sands and Gravels (*sensu* Gaunt [22]), can be accounted for by solifluction of locally derived sediments and fits with the periglacial environment at the time of deposition as judged from the associated involutions. The fines contained within the diamict (25% silt–clay), would have made it frost-susceptible and these along with any permafrost could have created saturated conditions at the base of the active layer. However, whilst the diamict thins considerably as it rises to the west up an interfluvium, suggesting an association with topography, present-day relief to allow either form of flow is limited. Gaunt [22] maps similar deposits as ‘Head’ (= solifluction deposit) in the shallow valley to the north of the quarry and there are more extensive deposits, which were perhaps continuous with these before the area was levelled for the expansion of Finningley (now Doncaster Sheffield Airport) airfield during the Second World and Cold War. It has been observed that sub-aqueous flows are often overlain by a drape of silty clay resulting from suspension settling from the water column. This is similar to the deposits that are found above or involuted into the top of the diamict. This clay drape, where it has not been modified by subsequent cryoturbation, has ripples on the surface of the diamict (figure 9), which is interpreted as shallow-water reworking after the deposition of the diamict. In this interpretation, a corollary is the temporary existence of a lake, impounded at the Humber Gap.

Unit 4 stratified sand

Unit 4, 2.5 m thick, is dominated by medium to fine (mean = 143 μm) sand which is moderately to poorly sorted (mean = 1.09) and reddish-yellow (5YR 6/6) grading to strong brown (7.5YR 5/6). It contains some comminuted fragments of coal, occasionally graded out in a distinct lens. The unit has faint sub-horizontal stratification (Sr/SI) with some channel or cross bedding and symmetrical and asymmetrical ripples (up to 10–20 mm in height) whose surfaces are accentuated occasionally by clay drapes, for example in the form of flaser bedding (figure 10 A). Ripple crests are orientated approximately east–west. In one long, cleaned exposure it was possible to see sigmoidal bed forms within the sand, indicating deposition in a littoral environment (figure 10 and figure 11). Symmetrical ripple forms have either peaked or rounded crests (figure 5 and figure 11 B), both representing wave ripples, and the peaked ones characteristic of formation in very shallow, near-emergent conditions. Also within this unit were a number of thin silt–clay horizons (mean = 7 μm , sorting = 1.54, 75% silt, 20% clay), including a persistent subunit of dark-grey (5YR 4/1) laminated clay–silt which was approximately 20 cm thick (figure 10 A), and present in all exposures examined. This subunit contained alternating dark grey and reddish brown laminae, horizontally oriented and separated by sharp contacts. The subunit is interpreted as varves, and a 30 mm long pebble within it as a dropstone. Luminescence samples were collected from above and below this persistent clay–silt subunit.

Numerous structures within this unit suggest it underwent post-depositional deformation. Vertical sand pipes ~2 cm wide cut across part of the varved silt–clay subunit, causing it to bifurcate into two in places with sand between the two parts (figure 11A). These structures are interpreted as dykes of sand injected up from below forming sand sills within the silt–clay subunit. The upper part of this unit was strongly deformed with water-escape structures. Also towards the top, some irregular and rather chaotic vertical to subvertical structures are developed, including one prominent flat-bottomed ‘sag’ structure (approx. 80 cm in height) infilled with massive to faintly laminated sand containing silt–clay blocks at the toe, and with adjacent downturned strata and stepped normal faults in the host strata (centre of figure 11A). Other examples overlie sand dykes and sills, including one structure comprising grey silty-clay blocks trailing upwards from the top of the varved unit (figure 11B). Such structures clearly involved movement of some sediment downward and/or upwards, and they are interpreted as sediment-injection / dewatering structures, indicating that at least the upper part of this unit underwent soft-sediment deformation. Further evidence of soft-sediment deformation is provided by load casts and diapirs (figure 11C)—which formed by loading processes—and by dish structures (figure 11D)—which formed by upward water escape. The upper boundary to this unit was sharp and largely planar.

A glaciolacustrine origin for unit 4 is inferred from the occurrence of varves and a dropstone. The abundance of wave ripples indicates an open-water body. Peaked symmetrical ripples suggest that this at times experienced very shallow, near-emergent conditions, whereas round crested symmetrical ripples may have developed in deeper water [54] or formed by reworking of ripple crests during emergence [55]. Wave activity may have triggered dewatering of the loosely consolidated lake sediments, producing the extensive array of soft-sediment deformation structures. This unit is correlated with the Littoral Sands and Gravels of Gaunt *et al.* [23] associated with Lake Humber.

Unit 5 laminated clay–silt

Unit 5, as exposed, was ~65 cm thick but thinned towards the west. This consisted of uniform, horizontally laminated clayey-silt (80% silt, 20% clay), compact, and dark-grey (10YR 4/1) in colour, with occasional sandy partings. The mean grain size was 4.0–78 μm and sorting was between 1.28 and 1.41. Calcium carbonate concretions and occasional pebbles, interpreted as dropstones, were also observed. Lamination thicknesses varied from 1–2 mm up to 1 cm. This unit formed the present-day land-surface over the western part of the quarry, where iron deposition on ped surfaces represents soil

development, although any higher profile had been incorporated into modern ploughsoil. Eastwards unit 5 was overlain by desiccated peat of Holocene age, and thickened eastwards towards the former course of the river Idle and is interpreted being part of the Hemingbrough Formation, deposited at depth into proglacial Lake Humber.

5.1. Plant Macrofossils

The raw sediment from Sample 2 was dominated by washed-together plant fragments (Table 2), in particular water-worn twigs and it also contained numerous sclerotia of *Cenococcum geophilum*. Seeds and fruits were relatively well preserved, however, *Empetrum* was only represented by endocarps and no leaves were found. Only a single tiny leaf of *Salix polaris* was found. No caryopses of grasses were found and bryophytes were only represented by a few *Sphagnum* leaves. The lack of such remains could be due to preservation, but it could also be due to hydrodynamic sorting in flowing water. It is surprising that no moss remains were found, because mosses are important in Arctic plant communities. Remains of mosses usually preserve well, and have been reported from a number of Middle Devensian (MIS3) sites in Britain and elsewhere in north-western Europe (e.g. [56]). Again, this may be due to hydrodynamic sorting. The most common fossil was *Cenococcum geophilum*; a fungus that lives in various soil types. Common presence of sclerotia of this species is usually taken as an indication of soil erosion [57], but the light sclerotia may also have been concentrated by flowing water. Remains of macrolimnophytes are rare in the fossil assemblage. *Ranunculus* sect. *Batrachium* sp. is represented by a few achenes, and this plant usually grows along the shores of ponds or small lakes. Submerged water plants comprise the charophyte *Nitella* sp. (1 oospore) and *Stuckenia filiformis* (2 endocarps). The submerged taxa indicate small lakes with clear, oligotrophic water. The rarity of water plant remains indicates that ponds or lakes were scarce in the area. *Carex* spp. may have grown in shallow water at lake or pond margins or in mires and *Sphagnum* in bogs. *Viola palustris* is also a mire plant, and *Selaginella selaginoides* is found in bogs or other areas with moist or wet soils.

Dwarf shrub heath communities are indicated by macrofossils of several species of dwarf shrubs. Endocarps of *Arctous alpina* were common, which is surprising because there appears to be no previous fossil records of this species from Britain [58,59]. *A. alpina* is a circumpolar low-arctic and boreal plant [60]. At the present day in the British Isles, it is restricted to montane moorland in Scotland. Fossil endocarps of *A. alpina* have been reported for example from Eemian deposits in Greenland [61] and Lateglacial deposits in southern Sweden and Denmark (e.g. [62,63]). The endocarps of *A. alpina* are similar to, but flatter than those of *Arctostaphylos uva-ursi*, which has been reported from Middle Devensian deposits in Britain [58]. Endocarps of *Empetrum nigrum* were also common. These are referred to *E. nigrum* and they may come from *E. nigrum* subsp. *hermaphroditum* (Hagerup) Böcher. *E. nigrum* endocarps are rarely reported from Middle Devensian deposits in Britain, but one endocarp was reported from Scotland by Bos *et al.* [59]. It has also, but rarely, been reported from Middle Devensian deposits elsewhere in north-west Europe. Houmark-Nielsen *et al.* [64] found a single endocarp of the species in a deposit in north-western Denmark. *E. nigrum* is a circumpolar plant, which has a wide geographical range in the Arctic, but is also widespread in the temperate zone.

Betula nana is represented by three nutlets. There are numerous fossil records of this species from Britain, following the first report by Heer [65], and its remains are common in Middle Devensian deposits [58]. *B. nana* is also a circumpolar arctic and northern boreal plant, presently restricted in the UK to northern England and Scotland. The last woody plant recovered was *Salix polaris*. It was represented by a tiny 1.4 mm long leaf. Although small, the margin was entire and showed no signs of a crenation or serration as seen in *S. herbacea*. The sample also contained a few bud scales and a small twig with two bud scales of *Salix* sp. *S. polaris* is one of the most cold-adapted of woody plants, and it has previously been reported from several Middle Devensian deposits in England [66,58]. It is currently found in northern Europe (including Svalbard), northern Asia and north western North America.

Herbaceous plants include *Luzula* cf. *spicata*, *Potentilla* cf. *crantzii*, *Rumex acetosa* and *Silene suecica*, which may have grown in heaths. *Epilobium* sp. grow in heaths or in plant communities on wet or moist soil. Their seeds have not previously been reported from Middle Devensian deposits in Britain according to Godwin [58]. *Rumex acetosella* is indicative of dry sandy or gravelly soils with patches of open vegetation and *Potentilla anserina* is often found near the shores of lakes, on bare ground or in grass-rich habitats; it can tolerate salt-rich soils.

In summary, the plant macrofossils suggest that the landscape was treeless tundra that supported widespread dwarf-shrub heaths. Some of the recorded plants indicate base-rich soils, whereas indicate more acidic conditions. The flora indicate a continuous snow cover in winter. Moisture conditions varied from dry and freely draining to wetland, pond or riparian.

5.2. Vertebrate remains

In 2005, a workman from the Fittingley Quarry took two lengths of tusk, presumed to be of mammoth, and a scapula of a bovid into Doncaster Museum [67]. He refused to leave the material and their subsequent fate is unknown, but he did suggest that other bones had been found during quarrying. Howes (pers. comm.), who examined the material, has suggested that bovid was either *Bos primigenius*, or steppe bison, *Bison* cf. *priscus*. It is unfortunate that it is not possible to place the bones in the stratigraphy. However, the underlying ORG is completely decalcified and the overlying deposits of the proglacial lake belong to a period when carrying capacity was too low to support any substantial vertebrate fauna, so it is probable therefore that these relate to the MIS 3 deposits and some support for this is provided by the dung faunas from the insect samples.

5.3. The Insect faunas

5.3.1. Coleoptera

Two samples produced insect remains. The fauna from Sample 2 is more extensive than that from Sample 3, and includes more species no longer found in Britain (cf. [68]); several have a present distribution which is essentially Siberian. The fauna is dominated by a dung beetle, *Aphodius (Chilothorax) jacobsoni*, identified on heads, pronota and patterns of maculae on the elytra. In his review of subgenus *Chilothorax*, however, Frolov [69] indicates that *A. jacobsoni* consists of a complex of several closely related species, although none appears to occur west of eastern Kazakhstan, and southern Siberia and the core distribution is Mongolian [70]; there appears to be little habitat data available. Other fossil records are restricted to MIS 3 deposits at Queensford, on the Thames terraces in Oxfordshire, where it is the most common dung beetle [71] and Coope (in [72]) also includes a possible identification from similar aged deposits at Sandy in Bedfordshire.

Helophorus praeanus is recorded from grassy pools in eastern Siberia, Mongolia and southwards into northern China [73,74]. The taxon was first identified as a fossil from Starunia, and Angus [73] also notes that it is a not infrequent component of insect assemblages from the colder parts of glacials. There are MIS 3 records from Baston Fen, south Lincolnshire [75], Earith, Cambridgeshire [76], Great Billing, Northamptonshire [77], Kempton Park, Surrey [78], Leeds, West Yorkshire [79], Lechlade, Queensford and Standlake, Oxfordshire [71] and Whitemoor Haye, Staffordshire [80]; records from Kirby-on-Bain in Lincolnshire [81] have recently been re-assigned to MIS 6 [24]. The remainder of the water beetle fauna also includes indicators of cold conditions, with the small dytiscids *Hydroporus lapponum* and *H. notabilis*, no longer found in the British Isles, being restricted largely to above the treeline in Scandinavia [82], occurring in ponds and shallow lakes. Both occur in the Lateglacial at St Bees in Cumbria [83], but only the latter has been found on other MIS 3 sites, at Lechlade and Queensford [71] and Upton Warren, Worcestershire [84]. Whilst the ground beetle *Pelophila borealis* would occur on the sandy margins of temporary pools on the floodplain [85], where the small rove beetle *Bledius arcticus* would have burrowed [86], both the more abundant *Notiophilus aquaticus* and *Amara quenseli* would have ranged more widely on sparsely vegetated sandy ground, where several of the rove beetles would have lived in the litter under low herbage. Several of the latter, including *Pycnoglypta lurida*, *Olophrum boreale*, *Eucnecosum* species, *Acidota quadrata*, *Boreaphilus henningianus* and *Holoboreaphilus nordenskioldi* are essentially arctic and Siberian in distribution, frequent in litter beneath low willows; all are not infrequent MIS 3 fossils. *H. nordenskioldi* is Holarctic, but only extends westwards as far as the Kanin Peninsula in Arctic Russia and is restricted to the tundra [87]. For the Nearctic, Morgan *et al.* [88] suggest an association with heavily disturbed ground over permafrost.

Utilising habitat traits from the classification in the BugsCEP database [89], the more extensive fauna from Sample 2 indicates a varied landscape with more or less equal indications of water, meadowland, wetland/marshes and heathland/moorland. Both standing and running water are indicated, the former more robustly than the latter (figure 12). There are few indications of the specific nature of the flora from the beetles. The weevils *Otiorynchus arcticus*, *O. nodosus*, *O. rugifrons* and *Tropiphorus obtusus* are essentially polyphagous on low vegetation, the larvae feeding on the roots, whilst species of *Notaris* are oligophagous on waterside and aquatic sedges, grasses and reeds [90,91], on which the small pollen beetle *Kateretes pedicularis* would also feed [92]. The remaining weevil, *Baris cf. artemisiae*, not currently found in Britain, breeds in wormwood, *Artemisia* species [93]. Species of *Phratora* feed on willows and poplars, whilst the ladybird *Ceratomegilla ulkei* is a predator on aphids on the taiga and tundra across the eastern Palaearctic and Nearctic from Kazakhstan to Hudson Bay in eastern Canada [94]. It is known in fossil form from MIS 3 deposits at Queensford [71], with earlier last glaciation records from Shropham in Norfolk [95] and Stanwick, Northamptonshire [96]. The remaining indicators of flora are the two byrrhids, *Simplocaria metallica* and *S. semistriata*, which feed on mosses [97,98]. The former is boreo-montane, not extending west of the Scandinavian mountains in Europe but known from Greenland [97].

The identification of the alpine chrysomelid *Oreina frigida* is tentative because of the fragmented nature of the fossil. [99] notes it as breeding in purple coltsfoot, *Homogyne alpina*, and Pasteels *et al.* [100] found an association of the imagines with the umbellifer, *Meum athemanticum*.

The insect fauna from Sample 3 was sparse, relatively poorly preserved and there were few species (Table 3), of which only *Helophorus sibiricus* provides significant ecological information. No longer found in the British Isles, it is a water beetle associated with pools at the edge of snow melt and upland rivers [73,101], characteristic of the taiga but extending into the tundra zone and southwards into montane temperate forest in the east. It has a broad Holarctic distribution, ranging from the Scandinavian mountains eastwards across northern Russia and Siberia to Alaska and northern Canada, west of the Mackenzie Delta [102]. A frequent and widespread fossil in Middle Devensian/Weichselian (MIS 3) and Lateglacial deposits in the British Isles [89], it has the distinction of a fossil record extending back to the Miocene in southern Siberia [102]. The scarabaeids in the sample include *Aphodius borealis*, a Holarctic dung beetle, which shows a preference for elk (moose) dung in shaded localities [103], although it has also been recorded on dunes and in the droppings of other large herbivores [104]. *Aegialia sabuleti* is found in plant litter on sandy river and stream banks [105]. Pittino [106] has recently described a new species, *A. insularis* as endemic to the British Isles, but this cannot be separated from *A. sabuleti* on the fossil material. Whilst *A. sabuleti* (s.l.) is a not infrequent Quaternary fossil [89], *Aphodius borealis* has only two other British fossil records, from MIS 3 deposits at Tattershall Castle, Lincolnshire [81], and from a Roman site in East Yorkshire (Buckland, unpubl.).

Overall, the picture is of a pond on a braided floodplain on sparsely vegetated sandy tundra, visited occasionally by larger vertebrates.

5.3.2. Trait-based habitat reconstruction from the Coleoptera

A quantitative, habitat trait-based reconstruction of the environment represented by the fauna allows for a general landscape model to be constructed, and comparison of the landscapes represented by the Coleoptera in the two samples (figure 12).

Taxon occurrence data (i.e. assuming one individual per taxon) suggest a mixed landscape for Sample 2 with more or less equal indications of water, meadowland, wetlands/marshes and heathland/moorland landscape components. Clear indications of standing water are provided by five species of water beetle (see above), but the running water indication is only provided by the Alder fly, *Sialis* sp., larvae (Megaloptera). Running water taxa are somewhat unreliable due to their propensity to be carried with the current and deposited in the sediments of pools and lakes. The presence of shaded environments is suggested by a number of taxa, mainly at the genus level (compare Fig 12 A and B), but only one taxon is an obligate xylophage or arboreal feeder. The single specimen of *Phratora* sp. would most likely have fed on the leaves of willow shrubs and the genus includes species common in arctic Scandinavia. The only potential woodland indication at the species level is the somewhat eurytopic *Simplocaria semistriatus*, a moss feeder as likely to be found under stones as in woodland.

Removing the poorly resolved identifications (i.e. not to species), which provide a less secure and more general reconstruction, and weighting the reconstruction by numbers of individuals (MNI) (Fig 12 B), focusses the picture to a pond in a less diverse, open landscape, with sparse vegetation of wetland and moorland character.

In the quantitative reconstruction, Sample 3 appears somewhat similar to Sample 2, with the exception of less prominent aquatic habitats and more disturbed ground (both wet and dry) in the former. The paucity of this fauna precludes the use of abundance weighting or only taxa identified to species, but the indications of disturbance and wet and dry habitats could fit with a braided river system. Evidence for grazing animals is potentially found in both samples, more so in Sample 2, but this could be a reflection of the better quality of the data in the latter.

5.3.3. Trichoptera (Caddis flies)

Sample 2 from Finningley contains preserved frontoclypeal, pro- and meso-notal sclerites from larvae of cased caddisflies (Order Trichoptera), and is made up of species from the families Phryganeidae, Apataniidae, Limnephilidae and Molannidae. As aquatic larvae, caddis flies can be found in habitats ranging from fast-flowing rivers to temporary field ponds. Assemblages of taxa from palaeo-deposits can offer a means of palaeoenvironmental reconstruction [107].

The most abundant and diverse taxa in the Finningley sample are from the Limnephilidae, (Table 4) and these indicate a varied habitat with riverine and lake-edge species: *Ecclisopteryx dalecarlica* has larval cases made of mineral particles; together with *Anabolia cf. nervosa*, *Limnephilus stigma* and *L. subcentralis*, is found in slow-flowing sections of river edges, and also from more permanent shallow pools, ponds and lakes. *Grammotaulius* spp., *Limnephilus algosus* and *L. picturatus* have cases constructed of plant fragments. The cold-adapted family Apataniidae is also represented and *Apatania* spp. are found in stony and gravel-bed brooks, rivers and lakes; the larval case is made of mineral particles. *Molanna albicans* (Molannidae) is found in slow-flowing rivers and small upland lakes, building a case also of mineral grains, and *Agrypnia picta* (Phryganeidae) is a species of permanent ponds and mires, constructing a case of large fragments of plant material, e.g. *Carex* spp. [108,109].

Of special interest is the species *Limnephilus algosus* (McLachlan, 1868), a cased larva found amongst submerged vegetation in permanent shallow pools, ponds, lakes, slow-flowing brooks and rivers [110,111,109]. To date there are no fossil records of this species in the UK but reference is made to this species, described as *Limnephilidae* indet., in papers published in Greenwood *et al.* [107]; Whittington *et al.* [112] and Schreve *et al.* [80], which can now be recorded as *Limnephilus algosus*. Figure 13 shows the frontoclypeal apotome from both modern and fossil material of this species; its colour pattern, shape, micro-sculpture and size are useful characters for identification. This species is recorded in sediments in the UK ranging from 41–43 cal yr BP at Whitemoor Haye, River Tame, Staffordshire [80], to 15,793–13,306 cal a BP., Clettnadal, Shetland [112], and to 13,817–13,543 cal a BP (CALIB 3.0) from Hemington, River Trent, Nottinghamshire [107].

A map of present distribution (figure 14) shows *Limnephilus algosus* to be widely distributed in the western Palaearctic. In Europe it is present in areas of high altitude and/or latitude and is described as a cold stenotherm of the alpine and sub alpine zones [52]. It is found in Norway, Sweden, Finland, Russia (European Russia, Siberia and in the Russian Far East), Germany (Bavaria), Austria (Tyrol and Oberösterreich), Switzerland, Czech Republic (Bohemia) and Slovakia [113,114,115]. The circumpolar nature of this species distribution was highlighted by Nimmo [116] in a publication that described a new species of *Limnephilus* from the Northwest Territories, Canada; this he named *L. innuitorum*, after the Inuit people of the Canadian and Greenlandic Arctic. Later Nimmo *et al.* [117] also recorded *L. innuitorum* from Chukotka in the Russian Far East. Grigorenko [118] then recognised that both descriptions matched, making *innuitorum* a synonym of *algosus* (Ivanov 2011) [113].

Overall, the larval caddis assemblage suggests a cold, arctic equivalent environment that is primarily one of slow-flowing streams and standing water (ponds and lakes) with occasional channels with moderate to fast flows.

5.3.4. Diptera

The fly assemblage from the site consists entirely of puparial fragments and is badly preserved and fragmented. One of the specimens included anal spiracles which made identification possible, but in other cases the small fragments became more fragmented whilst handling. Two of the taxa belonged to Anthomyiidae, a family of leaf and stem miners. *Delia cf. fabricii* is a species associated with smooth meadow grass, *Poa pratensis* [119]. The larvae pupate in the soil in the beginning of June and the adults emerge mid-summer [120]. *Botanophila* sp. belongs to another group of stem borers,

although some species are associated with fungi. A fragment of a calliphorid puparium, probably associated with either carrion or dung, was also recovered but identification to species level was not possible. Although information from the dipterous assemblage is limited, the environmental information points to a grassland environment.

5.4. Chronology

The *Empetrum* endocarps (crowberry) sample from the organic lens within the channel (unit 2) provided a radiocarbon age of $37,057 \pm 457$ ^{14}C years BP (UBA-33853). This corresponds to 40,314 - 39,552 cal. years BP, using the INTCAL13 data set [121].

The remaining ages both on the channel and other units are based on the OSL results (figure 15 & Table 1). Sample Fin 1 (Shfd02010) from unit 2, and above the radiocarbon sample, gave an age of 35.7 ± 2.0 ka corresponding closely to the radiocarbon date from this unit. Sample Fin 2 (Shfd02011) from the lower part of unit 4 beneath the clay gave an age of 23.4 ± 1.5 ka. Sample Fin 3 (Shfd02012) from above the clay in unit 4 gave an age of 19.2 ± 1.1 ka.

From these results it is clear that unit 1 is older than 35.7 ± 2.0 ka and unit 2 dates to between 39.9 ± 0.4 to 35.7 ± 2.0 ka BP. This corresponds to MIS 3 [122] and to between Greenland Stadial 9 (GS-9) through to Greenland Stadial 7 (GS-7) encompassing the interstadials of GI-8 [6]. Unit 3 lies between 35.7 ± 2.0 ka and 23.4 ± 1.5 ka (MIS 3/2). Unit 4 dates to between 23.4 ± 1.5 ka and sometime after 19.2 ± 1.1 ka (MIS 2) and Greenland Stadial 2 (GS-2; [6]). Unit 5 is younger than 19.2 ± 1.1 ka.

5.5. Palaeoclimate

5.5.1. Periglacial structures and climate

Ice-wedge pseudomorphs within the sand and gravel of unit 1 indicate the former occurrence of permafrost, and their intraformational nature implies that permafrost was broadly contemporaneous with sediment deposition. Their occurrence within aggradations of braided river deposits is commonly taken to indicate that ice wedges developed in inactive parts of the floodplain; such wedges may have thawed due to lateral migration of river channels or water ponding or flowing in troughs overlying ice wedges [123]. The former presence of permafrost during deposition of unit 1 discounts an interglacial or interstadial environment for gravel deposition. Instead, the gravel was deposited under a cold-stage (permafrost) environment.

Involutions present within the gravel may relate to some form of periglacial disturbance (e.g. differential frost heave), but it is difficult to be more specific about genesis or climatic significance in this instance because involutions in gravel are often difficult to interpret.

5.5.2. Beetle-based climate reconstruction

A palaeo-temperature reconstruction was undertaken (Finningley paleo in figure 16 and supplementary data) using the Mutual Climatic Range (MCR) method [124] as implemented in the BugsCEP software [125]. The MCR method reconstructs the thermal environment in which the largest proportion of species in a sample could survive, using their modern and historical ranges for reference. It allows for the calculation of TMax – mean temperature of the warmest month, TMin – mean temperature of the coldest month, and TRange – difference between TMax and TMin and essentially an indication of continentality. A higher TRange suggests a more continental climate, with larger differences between summer and winter temperatures. Although previous publications have derived means and errors, or otherwise indicated mid-points for reconstructed ranges, these have been shown to be mathematically incorrect [125,126] and are thus not used here. Coleoptera over-winter as a range of different stages (egg, larval instars and/or adult, pupa), many Arctic taxa surviving a wide variety of winter conditions in diapause [87], and there are a number of winter active species reliant on snow cover [127]. However, wherever snow thickness is a few tens of centimetres or more, as is common where shrubs trap snow, Coleoptera (or anything else) over-wintering or active at or near the land surface (i.e. beneath the snow) will not provide a good indication of air temperature. Winter temperature reconstructions from any organism with restricted winter activity are thus inherently less constrained than summer temperatures, as is reflected in the reconstructions in figure 16, and calibration data for more extreme (hot or cold) temperatures in the BugsCEP software are limited [125].

Sample 3 provides an MCR reconstruction from only three species (*Notiophilus aquaticus*, *Helophorus sibiricus* and *Aegialia sabuleti*) of $8\text{--}14^\circ\text{C}$ TMax and -22 to -2 TMin. This reconstruction is essentially constrained by the upper TMax and lower TRange limits of *H. sibiricus*, and the lower TMax and upper TRange limits of *A. sabuleti*; the thermal envelope for *N. aquaticus* is too poorly defined to be of use in defining the limits, but provides an additional species supporting the reconstructed temperatures. The reconstructed temperatures are compared to the modern climate at Finningley (1981–2010) (Table 5) as calculated from the Climate, Hydrology and Ecology research Support System (CHESS) [128]. Sample 3 suggests that the climate at that time was considerably colder than the present day, and neither this nor the temperatures reconstructed from Sample 2 overlap the equivalent values for the present, warmer climate. Although Sample 2 provides at a glance similarly cold, but apparently more refined thermal reconstruction of $12\text{--}13^\circ\text{C}$, this represents an overlap of the thermal envelopes of only 21 (87.5 %) of the 24 MCR species used in the reconstruction. Among the species most acutely responsible for the incomplete overlap are water beetles of the *Helophorus* genus. Regarded as a pioneer species frequenting temporary pools, *H. grandis* is scarce north of southern Scandinavia [101], and appears to be restricted to relatively warm (TMax $>11^\circ\text{C}$), oceanic climates. *H. praenanus*, on the other hand, appears to

be able to cope with the variation between the extreme cold winters and mild summers of Eastern Siberia. The third species which just fails to intersect the area of maximum overlap, is the rove beetle *Holoboreaphilus nordenskiöldi*, known only from areas with summers which get no warmer than 11°C. Whilst there are undoubtedly improvements to be made in the primary MCR calibration data, especially for more continental species, the envelope for *Helophorus grandis* is reasonably well resolved for its colder summer limits. The thermal limits to the distribution of *H. praeanus* and *Holoboreaphilus nordenskiöldi*, on the other hand, are poorly mapped and it could be that they overlap more than has been recorded. The same may apply to the eastern Palearctic dung beetle, *Aphodius jacobsoni* and other important species, such as the ladybird *Ceratomegilla ulkei*, yet to be included in the MCR calibration data, and thus not contributing to the reconstruction. Considering the extensive fossil record of *H. grandis* [89], and the difficulty of separating species in this group on the fossil parts, there is also the possibility that the fossil record includes both colder and warmer subspecies which have yet to be defined.

6.6. Discussion

6.1. Comparison of MIS 3 beetle-based climate reconstructions

Lemdahl and Coope [129] summarise the MCR results from insect faunas from Late Pleistocene sites in Europe, whilst Coope (2001) [28] in a more detailed examination of MIS 3 insect faunas concludes that, when correlated with the Greenland ice-core data, the preserved assemblages reflected the warmer episodes and that carrying capacity during the colder episodes was such that there were no associated organic sediments. It is debateable, however, whether the data are sufficiently robust to support this interpretation. Several of the older dates lay close to the limits of radiocarbon dating and should be treated as minima, others are bulk sample dates, and the association between dated context and insect faunas in others is less than secure (see supplemental data). The MIS 3 sample is further reduced by the number of species for which climate-space envelopes are available. Figure 16 provides beetle-based MCR summer (TMax) reconstructions for the majority of insect samples from MIS 3 sites in England (42 samples from radiocarbon dated sites with >2 MCR species, see supplementary data for detail). The majority of summer temperatures lie between 9 and 11 °C, which overlaps with the range of Sample 3 from Finningley (8–14°C), but is slightly lower than that of Sample 2 (12–13°C), the latter being based on more species and thus a more reliable reconstruction. The Finningley fauna most likely represents a cold stadial with both summers and winters slightly warmer than the climates indicated by the majority of the other MIS 3 samples. At least superficially, the climate is most similar to the temperature reconstruction based on the much larger, similarly dated fauna from Queensford near Dorchester on Thames: (44,491–39,400 cal a BP).

This interpretation fits with the Finningley floral list, although this is more limited than from some other Middle Devensian sites in England. This is partly because it lacks southern, thermophilous elements, such as *Najas flexilis*, *Lycopus europaeus*, *Groenlandia densa* and *Scirpus lacustris*, which were found in the Middle Devensian flora at Earith locality 7 [130,66]. The occurrence of floral southern elements at some sites may reflect that the climate during MIS 3 was extremely unstable, alternating rapidly between cold stadials and warmer interstadials. Coope [76] showed that part of the plant remains from Earith came from a cold and continental episode, and another part came from a temperate and more oceanic episode. It is also possible that some remains of southern plants at British sites are reworked from layers deposited during warmer intervals. Due to the problems with dating close to the limits of radiocarbon, it is also possible that some deposits referred to the Middle Devensian in earlier publications are significantly older.

With the exception of a small number of samples showing very well constrained, and extremely cold (-27 to -23°C) winter temperature reconstructions, in particular North Lechlade, samples S4 and S5 from Baston Fen and E9 from Earith, the TMin values for Finningley agree with colder samples from other sites. Although ten of the MIS 3 samples intersect the range of summer temperatures measured at Finningley between 1981 and 2010, only four of the corresponding winter reconstructions intersect the modern winter temperature range. Although the winter reconstructions are less well constrained, this could be tentatively interpreted as indicating that the interstadial climates represented in these samples were significantly more continental than the present day, something which simple modern distribution data for several of the taxa would also suggest.

The MCR results for Baston Fen (insect sample AMS dates of 36,606–31,859 cal a BP) generally indicate summer and winter temperatures much colder than present day, and colder than or overlapping the Finningley reconstruction. Baston Fen's mostly high TRange values (supplementary data) also suggest climates that are more continental throughout the site's range of dates. Baston S4 and S5, whilst showing cold summers, indicate significantly colder winters than most of the other MIS 3 samples, and at least 1°C and up to 20°C colder than Finningley. Two samples from this site (S3 and S6) suggest warmer summer conditions, equivalent to the lower range of present day Finningley, but with overlapping winter temperatures implying a highly continental climate with cool summers and extremely cold winters for at least part of MIS 3.

Samples from Four Ashes (>50,000 to 33,570 cal a BP) and Earith (> 50,000 to 43,168 cal a BP) provide reconstructions both at the colder end of Finningley's summers, including potentially colder winters (Four Ashes S3 and S45, Earith E4 and E9), but also summer temperatures within the lower span of modern variation. On the basis of the insects, these sites clearly cover both stadial and interstadial deposits, which are most likely to record colder and warmer conditions than Finningley, respectively. Interestingly, they also suggest periods of more oceanic climate, with less variation between summer and winter. The two small MCR faunas from Kempton Park (40,304 to 39,288 cal a BP) provide similar evidence, as do the two samples from Upton Warren (47,802 to 43,023 cal a BP, but see notes in supplemental data). MCR reconstructions from the beetles at Great Billing (33,069–31,348 cal a BP) are equally cold or colder than the Finningley fossils, with comparable summers and winters. Whilst Finningley overlaps with the reconstructed temperature

regimes of the single sample MIS 3 sites at Leeds Oxbow (46,966–40,569 cal a BP), Sandy (39,439–32,296 cal a BP) and the abundant fauna at Queensford. The younger fauna from North Lechlade (34,586–30,886 cal a BP) suggests significantly colder winters (-27 to -23°C), and this contrasts with the slightly warmer winters (-12 to -10°C) indicated by the significant fauna from Standlake (33,975–32,673 cal a BP).

Whilst Standlake and North Lechlade provide well-refined winter reconstructions above and below Finningley, the majority of samples from Tattershall Castle (>50,000 to 43,499 cal a BP) and Whitemoor Haye (47,084 to 44,391 cal a BP) provide winter reconstructions which overlap the entire range of values. These sites all indicate similar summer temperatures, almost entirely lower than Finningley sample two (with the exception of Whitemoor Haye S6/7, which overlaps the range of values from the Finningley sample). Upton Warren, although initially similarly dated to the latter two sites but considered older by Coope ([131]; but see [18]; [20]), produces MCR reconstructions which tend towards present-day summer temperatures, and potentially slightly warmer winter temperatures than these sites. These values are similar to the possibly interstadial and warmer samples described for Baston Fen, Earith and Four Ashes, and several degrees warmer than Finningley.

In summary, the MCR reconstructions from Finningley are thought to record cold stadial conditions, but with milder summers and winters than those inferred from many other English beetle faunas dated to MIS 3. The overall MIS 3 beetle dataset from England indicates substantial variation in summer and winter climate during this interval (~59–28 ka), consistent with independent evidence for substantial stadial–interstadial climate variability obtained from the Greenland ice-core record [132,6]. In view of the millennial timescales of such variability and the uncertainties of dating terrestrial sites at or near the limit of radiocarbon, it is clear the correlations even between terrestrial MIS 3 sites in England are challenging, and terrestrial-ice-core correlations remain speculative. Coope's [133] comment that episodes of gravel deposition in many sequences may reflect even colder conditions when carrying capacity was reduced such that there was little contemporary biota remains valid.

6.2. Periglacial and proglacial lake environments

Evidence for periglaciation at Finningley is broadly consistent with that from other lowland regions of England and the near-continent. The Finningley sequence records a change from permafrost conditions associated with deposition of the basal unit of sand and gravel to milder periglacial conditions during deposition of the overlying infilled channel.

Wedge structures similar to those in the basal unit have been reported from gravelly sequences of Pleistocene age in lowland England (see reviews in [134, p.53–63]; [123]). But the exact age of the gravel and wedges at Finningley is uncertain, precluding correlation with the British periglacial record. A minimum age is provided by the calibrated radiocarbon age of 39,933±381 cal. years BP from the overlying organic silt, but this study provides no maximum age. Thus, gravel deposition during permafrost conditions may have occurred during, for example, a MIS 3 stadial before ~40 ka or cold-climate conditions of MIS 4 or 6. Interestingly, there are similarities in terms of relatively small wedge size, intraformational nature and host stratified gravels with wedges and gravel of facies assemblage 1 at Baston, Lincolnshire, which have been dated by radiocarbon and OSL to the Middle Devensian [75]. If the wedges and gravel at Finningley are from a Middle Devensian stadial, one possibility is the Hasselo Stadial, when permafrost is known to have developed—inferred partly from ice-wedge pseudomorphs—in the Netherlands. The Hasselo Stadial has provided radiocarbon ages of 43,220–42,290 cal. BP [135]. These authors suggested that the subsequent Hengelo Interstadial (42,350–41,380 cal. BP) caused permafrost degradation and formation of involutions.

The sand and organic silt with a depositional age of ~40–35 ka infilling the river channel at Finningley are of similar age and environmental context to many Middle Devensian periglacial river deposits in lowland England (reviewed in [136, p. 162–171]; [134, p.58–61]; [19, p.452]). Contemporary rivers in lowland England were mainly of braided type and deposited gravel as a result of the supply of abundant coarse sediment from mass wasting on adjacent hillslopes [137]. A highly variable nival discharge regime favoured reworking of solifluction deposits during snowmelt floods, winnowing out the fine-grained sediment and leaving behind the gravel. Finer deposits exposed on floodplains and eroded from pre-existing deposits were vulnerable to reworking by wind and water, which deposited sediments such as coversand by wind action (e.g. [11]) and laminated silt and sand by aeolian rainout and sheetwash [138]. Palaeoecological evidence associated with the Middle Devensian river gravels in eastern England [139,75] indicates wetter conditions at the time of gravel deposition and hence more active river systems than those during the LGM. In summary, Middle Devensian periglacial environments, at least in southeast and eastern England, are thought to have been warmer and wetter than the cold and arid conditions of the Early and Late Devensian.

Both units 4 and 5 are interpreted as indicating the presence in the Finningley area of Lake Humber. The coarser sands and gravels of unit 4 are taken to indicate that in the period 23.4±1.5 ka to 19.2±1.1 ka Finningley was close to the margins of this lake. Unit 5 is interpreted as indicating deeper water at Finningley occurred sometime after 19.2 ±1.1 ka. Such ages are not out of line with those presented from wave-rippled silty sand at Hemingbrough by Murton *et al.* [37], or the basal age Bateman *et al.* [39] reported from laminated clays from Lake Humber. Given that ice had to block the Humber Gap for Lake Humber to form, this also indicates the movement of the North Sea Ice lobe on-shore at this time, something confirmed by recent glacial ages from Ferriby [40]. The low-lying nature of the Finningley site cannot add information as to the absolute lake levels as the sediments themselves occur at between 6 and 8 m OD. However, the switch from units 4 to 5 may indicate that an earlier, lower and more fluctuating Lake Humber preceded the more widespread and deeper Lake Humber associated with the laminated clays.

7. Conclusion

11

The new work shows that the Finningley sequence records a change from permafrost conditions to milder periglacial conditions during deposition of the infilled channel. Unit 1, at the base, is tentatively correlated with the ORG of Gaunt [22], and may represent more than one glacial–interglacial cycle. The organic silt in Unit 2 dated to 40,314 - 39,552 cal a BP (MIS 3) contains Arctic flora and fauna indicative of deposition in a cold, continental arctic regime. The mean temperature of the warmest month is reconstructed—from beetle-based MCR data—as 8 to 14°C and, with less certainty, that of the coldest month as -22 to -2°C. The channel is sealed by a diamict (Unit 3) derived from a low ridge of glaciofluvial deposits to the west. Unit 4, largely thin bedded sands with some clay–silts, is the equivalent of Gaunt’s [22] Littoral Sands and Gravels. Unit 5, beneath largely desiccated Holocene peat, consists of the laminated clay–silts of the Hemingbrough Formation, deposited at depth into the proglacial Lake Humber during the LGM.

Overall MIS 3 in England between 59–28 ka appears to have had substantial variation in summer and winter climate. This is consistent with the Greenland ice-core record. Comparison of this new proxy data from the temporally well constrained Finningley site with other MIS 3 records in England suggests around 40 ka was a prominent cold stadial. However difficulties of dating and correlation of most MIS 3 sites within England makes it challenging to correlate with certainty to the Greenland ice-core record. At present the stadial could be correlated with any of the Greenland Stadials between 7 to 11.

Acknowledgments

Access to the site was kindly provided by the successive quarry managers of LaFarge/Tarmac. MSc students from Sheffield University helped clear sections. Laura Trinogga at the Doncaster Museum and Dmitri Logunov at the Manchester Museum are thanked for access to insect collections. Geoff Gaunt and members of the QRA fieldtrip 2001 provided stimulating discussions which helped shape this manuscript. Robert Angus kindly confirmed the identification of *Helophorus praenanus* and Tim Prosser provided the modern climate information. The authors wish to acknowledge the help given by Peter Neu (Germany), Peter Wiberg-Larsen (Denmark), Aki Rinni (Finland), Peter Barnard, (UK), and Ian Wallace (UK) for Trichopteran reference specimens, distribution data and taxonomic expertise and Colin Howes for drawing our attention to the vertebrate remains. Ben Price (Natural History Museum, London) provided access to Imaging Facility and Mark Szegner Loughborough University) collated figure 13 and figure 14.

Ethical Statement

Does not apply to this manuscript.

Funding Statement

The lead author’s research time was funded by Umeå University Faculty of Arts and Humanities.

Data Accessibility

The datasets supporting this article have been uploaded as part of the Supplementary Material.

The new fossil insect data upon which this paper is based are provided as supplementary material as well as being deposited in the freely available BugsCEP database (www.bugscep.com) and SEAD database (www.sead.se) on publication. DOI’s are not currently available through these resources.

The compilation of MIS 3 dates is provided as supplementary material, including as much metadata for each date as was obtained through our literature search.

The BugsCEP database and software includes all of the MIS 3 beetle data that was used to undertake the climate reconstruction of comparative sites. BugsCEP’s built in Mutual Climatic Range (MCR) routines were used to calculate all beetle-based climate reconstructions presented. The software is downloadable as an MS Access 2000 database and all code are openly accessible in this file.

Competing Interests

We have no competing interests.

Authors' Contributions

Philip Buckland undertook the beetle based environmental and climate reconstruction work, co-compiled the comparative sites data, and contributed extensively to the text. All other authors contributed equally to the analysis and writing, and are listed in alphabetical order:

Mark D Bateman undertook the OSL dating and interpretation, and contributed to the stratigraphic interpretation.

Paul Buckland identified the Coleoptera remains, coordinated the collaboration and writing, and contributed to site interpretation.

Ole Bennike identified and interpreted the plant macrofossil remains.

Brian Chase initiated the fieldwork and interpreted the regional context of the sequence.

Charles Frederick recorded the sections and provided additional site data and photographs.

Malcolm Greenwood identified and interpreted the Trichoptera remains.

Julian Murton carried out additional fieldwork on the site and contributed to site interpretation, particularly the fine-grained stratigraphy.

Della Murton carried out additional fieldwork on the site and contributed to site interpretation.

Eva Panagiotakopulu sampled the organic sediments and identified the Dipterous remains.

References

1. Clark CD, Evans DJA, Khatwa A, *et al.* 2004. Map and GIS database of glacial landforms and features related to the last British ice sheet. *Boreas* **33**: 359–375.
2. Clark CD, Hughes ALC, Greenwood SL, *et al.* 2012. Pattern and timing of retreat of the last British-Irish Ice Sheet. *Quaternary Science Reviews* **44**: 112–146.
3. Hughes ALC, Clark CD, Jordan CJ. 2014. Flow-pattern evolution of the last British Ice Sheet. *Quaternary Science Reviews* **89**: 148–168.
4. Smedley RK, Scourse JD, Small D, *et al.* 2017. New age constraints for the limit of the British-Irish Ice Sheet on the Isles of Scilly. *Journal of Quaternary Science* **32**: 48–62.
5. Roberts DH, Evans DJA, Callard SL, *et al.* 2018. Ice marginal dynamics of the last British-Irish Ice Sheet in the southern North Sea: Ice limits, timing and the influence of the Dogger Bank. *Quaternary Science Reviews* **198**: 181–207.
6. Rasmussen SO, Bigler M, Blockley SP, *et al.* 2014. A stratigraphic framework for abrupt climatic changes during the Last Glacial period based on three synchronized Greenland ice-core records: refining and extending the INTIMATE event stratigraphy. *Quaternary Science Reviews* **106**: 14–28.
7. Straw A. 1980. An early Devensian glaciation in eastern England reiterated. *Quaternary Newsletter* **31**: 18–23.
8. Straw A. 2016. Devensian glaciers and proglacial lakes in Lincolnshire and southern Yorkshire. *Mercian Geologist* **19**: 39–46.
9. Evans DJA, Roberts DH, Bateman MD, *et al.* 2018. Sedimentation during MIS 3 at the eastern margins of the Glacial Lake Humber basin, England. *Journal of Quaternary Science* **33**: 871–891.
10. Carr SJ, Holmes RVD, van der Meer JJM, *et al.* 2006. The Last Glacial Maximum in the North Sea Basin: micromorphological evidence of extensive glaciation. *Journal of Quaternary Science* **21**: 131–153.
11. Bateman MD, Hitchens S, Murton JB, *et al.* 2014. The evolution of periglacial patterned ground in East Anglia, UK. *Journal of Quaternary Science* **29**: 301–317.
12. Anderson DE, Goudie AS, Parker AG 2007. *Global Environments through the Quaternary*. Oxford University Press: Oxford.
13. Worsley P, Robinson JE, Coope GR, *et al.* 1983. A Pleistocene succession from beneath Chelford Sands at Oakwood Quarry, Chelford, Cheshire. *Geological Journal* **18**: 307–324.
14. Rendell H, Worsley P, Green F, *et al.* 1991. Thermoluminescence dating of the Chelford Interstadial. *Earth and Planetary Science Letters* **103**: 182–189.
15. Worsley P. 2015. Late Pleistocene geology of the Chelford area of Cheshire. *Mercian Geologist* **18**: 202–212.
16. Bryant, I.D., Holyoak, D.T. and Moseley, K.A., 1983. Late Pleistocene deposits at Brimpton, Berkshire, England. *Proc Geol Assoc.* **94**(4): 321–343.
17. Worsley P, Robinson JE, Coope GR, *et al.* 1983. A Pleistocene succession from beneath Chelford Sands at Oakwood Quarry, Chelford, Cheshire. *Geological Journal* **18**: 307–324.
18. Coope GR, Gibbard, PL, Hall AR, *et al.* 1997. Climatic and environmental reconstruction based on fossil assemblages from Middle Devensian (Weichselian) deposits of the River Thames at South Kensington, Central London, UK. *Quaternary Science Reviews* **16**: 1163–1195.
19. Catt JA, Gibbard PL, Lowe JJ, *et al.* 2006. Quaternary: ice sheets and their legacy. In *The Geology of England and Wales* (2nd ed). Brenchley PJ, Rawson PF (eds). Geological Society: London, 429–467.
20. Penkman KEH, Preece RC, Bridgland DR, *et al.* 2013. An aminostratigraphy for the British Quaternary based on *Bithynia* opercula. *Quaternary Science Reviews* **61**: 111–134.
21. Gaunt GD. 1976a. *The Quaternary geology of the southern part of the Vale of York*. Earth Sciences. Unpubl. PhD, University of Leeds.
22. Gaunt GD. 1994. *Geology of the country around Goole, Doncaster and the Isle of Axholme. Memoir of the British Geological Survey Sheets 79 and 88 (England and Wales)*. HMSO: London.
23. Gaunt GD, Fletcher TP, Wood CJ 1992. *Geology of the country around Kingston upon Hull and Brigg. Memoir of the Geological Survey of England and Wales, Sheets 80 and 89*. HMSO: London.
24. Bridgland DR, Howard AJ, White MJ, *et al.* (eds). 2014. *Quaternary of the Trent*. Oxbow Books: Oxford.
25. Bridgland, D.R., Howard, A.J., White, M.J., White, T.S. and Westaway, R., 2015. New insight into the Quaternary evolution of the River Trent, UK. *Proc Geol Assoc.* **126**(4–5), pp.466–479.
26. Gaunt GD. 1976b. The Devensian maximum ice limit in the Vale of York. *Proceedings of the Yorkshire Geological Society* **40**: 631–637.
27. Gaunt GD, Coope GR, Osborne PJ, *et al.* 1972. *An interglacial deposit near Austerfield, southern Yorkshire*. HMSO: London.
28. Coope GR. 2001. Biostratigraphical distinction of interglacial coleopteran assemblages from southern Britain attributed to Oxygen Isotope Stages 5e and 7. *Quaternary Science Reviews* **20**: 1717–1722.
29. Bateman MD, Buckland PC, Frederick CD, *et al.*, (eds) 2001. *The Quaternary of East Yorkshire and North Lincolnshire. Field Guide*. Quaternary Research Association: London.
30. Gaunt GD, Buckland PC, Bateman MD. 2006. The geological background to the development and demise of a wetland - the Quaternary history of the Humberhead Levels. *Yorkshire Naturalist' Union Bulletin* **45**: Suppl.: 6–46.
31. Thomas GSP. 1999. Northern England. In *A revised correlation of Quaternary deposits in the British Isles*. Bowen DQ. (ed) Geological Society of London Special Report **23**: 91–98.
32. Ford JR, Cooper AH, Price SJ, *et al.* 2008. *Geology of the Selby District - a brief explanation of the geological map. 1:50,000 sheet 71 Selby (England and Wales)*. British Geological Survey: Nottingham.
33. Murton DK, Murton JB. 2012. Middle and Late Pleistocene glacial lakes of lowland Britain and the southern North Sea Basin. *Quaternary International* **260**: 115–142.
34. Murton DK. 2018. A re-evaluation of Late Devensian glacial Lake Humber levels in the Vale of York, UK. *Proceedings of the Geologists' Association* **129**: 561–576.
35. Bateman MD, Buckland PC, Chase B, *et al.* 2008. The Late-Devensian pro-glacial Lake Humber: new evidence from littoral deposits at Ferrybridge, Yorkshire, England. *Boreas* **37**: 195–210.
36. Friend RJ, Buckland PC, Bateman MD, *et al.* 2016. The 'Lindholme Advance' and the extent of the Last Glacial Maximum in the Vale of York. *Mercian Geologist* **19**: 18–25.
37. Murton DK, Pawley SM, Murton JB. 2009. Sedimentology and luminescence ages of Glacial Lake Humber deposits in the central Vale of York. *Proceedings of the Geologists' Association* **120**: 209–222.
38. Bateman MD, Buckland PC, Whyte MA, *et al.* 2011. Re-evaluation of the Last Glacial Maximum typesite at Dimlington, UK. *Boreas* **40**: 573–584.
39. Bateman MD, Evans DJA, Buckland PC, *et al.* 2015. Last glacial dynamics of the Vale of York and North Sea lobes of the British and Irish Ice Sheet. *Proceedings of the Geologists' Association* **126**: 712–730.
40. Bateman MD, Evans DJA, Roberts DH, *et al.* 2018. The timing and consequences of the blockage of the Humber Gap by the last British-Irish Ice Sheet. *Boreas* **47**: 41–61.
41. Fairburn WA. 2014. *A re-interpretation of the physiographic evolution of the southern end of the Vale of York from the mid-Pleistocene to Early Holocene*. Unpubl. PhD thesis. University of Sheffield.
42. Fairburn WA, Bateman MD. 2016. A new multi-stage recession model for Proglacial Lake Humber during the retreat of the last British-Irish Ice Sheet. *Boreas* **45**: 133–151.
43. Gale S, Hoare P, 1991. Petrographic Methods for the Study of Unlithified Rocks. New York: Belhaven Press.
44. Bateman MD, Catt JA. 1996. An absolute chronology for the raised beach and associated deposits at Sowerby, East Yorkshire, England. *Journal of Quaternary Science* **11**: 389–395.
45. Murray AS, Wintle AG. 2000. Luminescence dating of quartz using an improved single-aliquot regenerative-dose protocol. *Radiation Measurements* **32**: 57–73.
46. Galbraith RF, Green PF. 1990. Estimating the Component Ages in a Finite Mixture. *Nuclear Tracks and Radiation Measurements* **17**: 197–206.
47. Guérin G, Mercier N, Adamiec G 2011. Dose-rate conversion factors: update. *Ancient TL*, **29**(1): 5–8.
48. Prescott JR, Hutton JT 1994. Cosmic ray contributions to dose rates for luminescence and ESR dating: large depths and long-term time variations. *Radiation measurements*, **23**(2–3): 497–500.
49. Coope GR, Osborne PJ. 1968. Report on the Coleopterous Fauna of the Roman Well at Barnsley Park, Gloucestershire. *Transactions of the Bristol and Gloucestershire Archaeological Society* **86**: 84–87.
50. The Plant List 2013. Version 1.1. <http://www.theplantlist.org/> (accessed 3.18).
51. Böhme J. 2005. Die Käfer Mitteleuropas. Bd. K-Katalog: Faunistische Übersicht. Elsevier GmbH, Spektrum Akademischer Verlag, Heroldsberg, Krefeld.
52. Graf W, Murphy J, Dahl J, *et al.* 2008. *Distribution and Ecological Preferences of European Freshwater Organisms. Volume*

- 1: *Trichoptera*. Pensoft Publishers: Sofia-Moscow.
53. Klassen RA, Murton JB. 1996. Quaternary geology of the Buchans area, Newfoundland: Implications for mineral exploration. *Canadian Journal of Earth Sciences* **33**(2): 363–377.
54. Collinson JD, Moutney JP, Thompson DB. 2006. *Sedimentary Structures*, 3rd edition. Terra Publishing: Harpenden.
55. Reineck HE, Singh IB. 1975. *Depositional Sedimentary Environments with Reference to Terrigenous Clastics*. Springer: Berlin.
56. Dickson JH. 1973. *Bryophytes of the Pleistocene. The British Record and its Chronological and Ecological Implications*. Cambridge University Press: Cambridge.
57. Liedberg-Jönsson B. 1988. *The Late Weichselian macrofossil flora in western Skåne, southern Sweden*. LUNDQUA Thesis **24**. University of Lund.
58. Godwin H. 1975. *The history of the British flora: a factual basis for phytogeography*. Cambridge University Press: Cambridge.
59. Bos JAA, Dickson JH, Coope GR, et al. 2004. Flora, fauna and climate of Scotland during the Weichselian Middle Pleniglacial – palynological, macrofossil and coleopteran investigations. *Palaeogeography, Palaeoclimatology, Palaeoecology* **204**: 65–100.
60. Hultén E, Fries M. 1986. *Atlas of North European vascular plants I–III*. Koeltz Scientific Books: Königstein.
61. Bennike O, Böcher J. 1994. Land biotas of the last interglacial/glacial cycle on Jameson Land, East Greenland. *Boreas* **23**(4): 479–87.
62. Berglund, BE, Digerfeldt, G. 1970. A palaeoecological study of the Late-Glacial lake at Torreberga, Scania, South Sweden. *Oikos*: 98–128.
63. Bennike, O, Jensen, JB, Lemke, W, Kuijpers, A, Lomholt, S. 2004. Late- and postglacial history of the Great Belt, Denmark. *Boreas*, **33**(1): 18–33.
64. Houmark-Nielsen M, Bennike O, Björck, S. 1996. Terrestrial biotas and environmental changes during the late Middle Weichselian in north Jylland, Denmark. *Bulletin of the Geological Society of Denmark* **43**: 169–176.
65. Heer O. 1862. On the fossil flora of Bovey Tracey. *Philosophical Transactions of the Royal Society of London* **152**: 1039–1086.
66. Bell, FG. 1970. Late Pleistocene floras from Earith, Huntingdonshire. *Phil. Trans. R. Soc. Lond. B*, **258**(826): 347–378.
67. Lomax DR, Sherburn J. 2013. Rediscovering Doncaster's geology and palaeontology: hidden treasures of the borough. *Deposits* **34**: 7–12.
68. Buckland PI, Buckland PC. 2018. Species found as fossils in Quaternary sediments. In *Checklist of Beetles of the British Isles*, (3rd ed). Duff, AG. Pemberley Books: Iver: 171–174.
69. Frolov AV. 2001. A review of the subgenus *Chilothorax* of the genus *Aphodius* (Coleoptera, Scarabaeidae) occurring in Mongolia, with description of new species. *Vestnik zoologii*, **35**: 39–45.
70. Akhmetova LA, Frolov AV. 2014. A review of the scarab beetle tribe Aphodiini (Coleoptera, Scarabaeidae) of the fauna of Russia. *Entomological Review* **94**: 846–879.
71. Briggs DJ, Coope GR, Gilbertson DD. 1985. *The Chronology and Environmental Framework of Early Man in the Upper Thames Valley*. British Archaeological Reports, 137: Oxford.
72. Gao C, Coope GR, Keen DH, et al. 1998. Middle Devensian deposits of the Ivel Valley at Sandy, Bedfordshire, England. *Proceedings of the Geologists' Association*, **109**: 127–137.
73. Angus RB. 1973. Pleistocene *Helophorus* (Coleoptera, Hydrophilidae) from Borislav and Starunia in the Western Ukraine, with a reinterpretation of M. Lomnicki's species, description of a new Siberian species, and comparison with British Weichselian Faunas. *Philosophical Transactions of the Royal Society of London B: Biological Sciences* **265**: 299–326.
74. Angus RB, Jia F-L, Chen Z-N. 2014. A review of the *Helophorus frater-praenanus* group of species, with description of a new species and additional faunal records of *Helophorus FABRICIUS* from China and Bhutan (Coleoptera: Helophoridae). *Koleopterologische Rundschau* **84**: 209–219.
75. Briant RM, Coope GR, Preece RC, et al. 2004. Fluvial system response to Late Devensian (Weichselian) aridity, Baston, Lincolnshire, England. *Journal of Quaternary Science* **19**: 479–495.
76. Coope GR. 2000. Middle Devensian (Weichselian) coleopteran assemblages from Earith, Cambridgeshire (UK) and their bearing on the interpretation of 'Full glacial' floras and faunas. *Journal of Quaternary Science* **15**: 779–788.
77. Morgan A. 1969. A Pleistocene fauna and flora from Great Billing, Northamptonshire, England. *Opuscula Entomologica* **34**: 109–129.
78. Gibbard PL, Coope GR, Hall AR, et al. 1982. Middle Devensian deposits beneath the 'Upper Floodplain' terrace of the River Thames at Kempton Park, Sunbury, England. *Proceedings of the Geologists' Association* **93**: 275–289.
79. Gaunt GD, Coope GR, Franks JW. 1970. Quaternary deposits at Oxbow opencast coal site in the Aire Valley, Yorkshire. *Proceedings of the Yorkshire Geological Society* **38**: 175–200.
80. Schreve D, Howard A, Currant A, et al. 2013. A Middle Devensian woolly rhinoceros (*Coelodonta antiquitatis*) from Whitemoor Haye Quarry, Staffordshire (UK): palaeoenvironmental context and significance. *Journal of Quaternary Science*, **28**: 118–130.
81. Girling MA. 1980. *Two Late Pleistocene insect faunas from Lincolnshire*. Unpubl. Ph.D. thesis, University of Birmingham.
82. Nilsson AN, Holmen M. 1995. *The aquatic Adephaga (Coleoptera) of Fennoscandia and Denmark. II. Dytiscidae*. Fauna Entomologica Scandinavica, **32**. E.J.Brill: Leiden.
83. Coope GR, Joachim MJ. 1980. Lateglacial environmental changes interpreted from fossil Coleoptera from St. Bees, Cumbria, N.W. England. In *Studies in the Lateglacial of North-West Europe*. Lowe JJ, Gray JM, Robinson JE (eds) Pergamon: Oxford; 55–68.
84. Coope GR, Shotton FW, Strachan I. 1961. A Late Pleistocene fauna and flora from Upton Warren, Worcestershire. *Philosophical Transactions of the Royal Society of London* **B244**: 379–421.
85. Lindroth CH. 1985. *The Carabidae (Coleoptera) of Fennoscandia and Denmark*. Fauna Entomologica Scandinavica, **15**, 1. E.J.Brill: Leiden.
86. Lindroth CH. 1935. The Boreo-British Coleoptera. *Zoogeographica* **2**: 579–634.
87. Morgan AV, Morgan A, Miller RF. 1984. Range extension and fossil occurrences of *Holoboreaphilus nordenskiöldi* Mäklin (Coleoptera: Staphylinidae) in North America. *Canadian Journal of Zoology* **62**: 463–467.
88. Chernov YI, Makarova OL, Penev LD, et al. 2014. The beetles (Insecta, Coleoptera) in the Arctic fauna. Communication 1. Faunal composition. *Entomological Review* **94**: 438–478.
89. Buckland PI, Buckland PC. 2006. *BugsCEP Coleopteran Ecology Package*. IGBP PAGES/World Data Center for Paleoclimatology Data Contribution Series # 2006-116. NOAA/NCDC Paleoclimatology Program, Boulder CO, USA. URL: <http://www.ncdc.noaa.gov/paleo/insect.html> or <http://www.bugscep.com>. (Versions: BugsCEP v7.56; Bugsddata v7.11; BugsMCR v2.0; BugStats v1.2) [Downloaded 10 4 2017]
90. Morris MG. 1997. *Broad-Nosed Weevils. Coleoptera : Curculionidae (Entiminae)* Handbooks for the Identification of British Insects, **5**, 17a. Royal Entomological Society: London.
91. Morris MG. 2002. *True Weevils (Part 1) Coleoptera: Curculionidae (Subfamilies Raymondionymidae to Smicronychinae)*. Handbooks for the identification of British insects, **3**, 17b. Royal Entomological Society & Field Studies Council: London.
92. Kirk-Spriggs AH. 1996. *Pollen beetles. Coleoptera: Kateretidae and Nitidulidae: Meligethinae*. Handbooks for the identification of British insects **5**, 6a. Royal Entomological Society: London.
93. Rheinheimer J, Hassler M. 2010. *Die Rüsselkäfer Baden-Württembergs*. LUBW Landesanstalt für Umwelt, Messungen und Naturschutz Baden-Württemberg: Karlsruhe.
94. Gordon RD. 1985. The Coccinellidae (Coleoptera) of America north of Mexico. *Journal of the New York Entomological Society* **93**: 1–932.
95. Coope GR. 1995. The Effects of Quaternary Climatic Changes on Insect Populations: Lessons from the Past. In: *Insects in a changing environment*, R. Harrington & N. E. Stork (eds). Academic Press: London; 29–48.
96. Briant RM, Gibbard PL, Boreham S, et al. 2008. Limits to resolving catastrophic events in the Quaternary fluvial record: a case study from the Nene valley, Northamptonshire, UK. In *Landscape evolution: denudation, climate and tectonics over different time and space scales*, Gallagher K, Jones SJ & Wainwright J (eds). Geological Society Special Publ. 296, 79–104. London.
97. Böcher J. 1988. The Coleoptera of Greenland. *Meddelelser om Grønland Bioscience* **26**.
98. Strand A. 1946. Nord Norges Coleoptera. *Tromsø Museums Arshefter, Naturhistorisk Avd.* **34**, 67(1).
99. Koch K. 1992. *Die Käfer Mitteleuropas. Ökologie* 3. Goecke & Evers: Krefeld.
100. Pasteels JM, Dobler S, Rowell-Rahier M, et al. 1995. Distribution of autogenous and host-derived chemical defenses in *Oreina* leaf beetles (Coleoptera: Chrysomelidae). *Journal of Chemical Entomology* **21**: 1163–1179.
101. Hansen M. 1987. *The Hydrophiloida (Coleoptera) of Fennoscandia and Denmark*. Scandinavian Science Press: Leiden.
102. Fikáček M, Prokin A, Angus RB. 2011. A long-living species of the hydrophiloid beetles: *Helophorus sibiricus* from the early Miocene deposits of Kartashevo (Siberia, Russia). *Zookeys* **130**: 239–254.
103. Rice ME. 2010. Niche Preference of a Coprophagous Scarab beetle (Coleoptera: Scarabaeidae) for summer moose dung in Denali National Park, Alaska. *Coleopterists Bulletin* **64**: 148–150.
104. Roslin, T, Forshage, M, Ødegaard, F, Ekblad, C, Liljeberg, G. 2014. *Nordens dyngbaggar*. Hyönteistarvike Tibiale Oy.
105. Hyman PS. 1992. *A review of the scarce and threatened Coleoptera of Great*

- 1
2
3
4
5
6
7
8
9
10
11
12
13
14
15
16
17
18
19
20
21
22
23
24
25
26
27
28
29
30
31
32
33
34
35
36
37
38
39
40
41
42
43
44
45
46
47
48
49
- Britain, Part 1.* (Revised & updated by M.S.Parsons). UK Joint Nature Conservation Committee: Peterborough.
106. Pittino R. 2006. A revision of the genus *Psammodorus* Thompson 1859 in Europe, with description of two new species (Coleoptera, Scarabaeoidea: Aegialiidae). *Geomale Italiana d'Entomologia* **11**: 325–342.
107. Greenwood MT, Agnew MD, Wood PJ 2003. The use of caddisfly fauna (Insecta: Trichoptera) to characterize the Late-glacial River Trent, England. *Journal of Quaternary Science* **18**: 645–661.
108. Wallace, I, 2003. The beginner's guide to caddis (order Trichoptera). *Bulletin of the Amateur Entomologists' Society* **62**: 15-26.
109. Rinni A, Wiberg-Larsen P. 2017 *Trichoptera Larvae of Finland: A Key to the Caddis Larvae of Finland and Nearby Countries.* Trificon Books: Helsinki.
110. Ekrem T, Roth S, Andersen T, *et al.* 2012. Insects inhabiting freshwater and humid habitats in Finnmark, northern Norway. *Norwegian Journal of Entomology* **59**: 91–107.
111. Andersen T, Hagenlund LK. 2012. Caddisflies (Trichoptera) from Finnmark, northern Norway. *Norwegian Journal of Entomology* **59**: 133–154.
112. Whittington, G., Buckland, P., Edwards, K.J., Greenwood, M., Hall, A.M. and Robinson, M., 2003. Multiproxy Devensian Late-glacial and Holocene environmental records at an Atlantic coastal site in Shetland. *Journal of Quaternary Science: Published for the Quaternary Research Association* **18**(2): 151-168.
113. Ivanov VD. 2011. Caddisflies of Russia: Fauna and biodiversity. *Zoosymposia* **5**: 171–209.
114. Schmidt-Kloiber A, Hering D. 2015. www.freshwaterecology.info – An online tool that unifies, standardises and codifies more than 20,000 European freshwater organisms and their ecological preferences. *Ecological Indicators*, **53**, 271-282.
115. Neu, PJ, Malicky, H, Graf, W. Schmidt-Kloiber, A. 2018. *Distribution Atlas of European Trichoptera.* ConchBooks.
116. Nimmo AP. 1991. Seven new species of *Limnephilus* from western North America with description of female *L. pallens* (Banks) (Trichoptera, Limnephilidae, Limnephilinae, Limnephilini). *Proceedings of the Entomological Society of Washington* **93**: 499–508.
117. Nimmo AP, Arefina TI, Levanidova IM. 1997. Fam. Limnephilidae. In *Key to the insects of Russian Far East*. V, pt1. Kononenko VS (ed). Dal'nauka (Vladivostok): 93–126 (in Russian).
118. Grigorenko VN. 2002. Some taxonomical notes on the limnephiline caddisflies (Trichoptera, Limnephilidae, Limnephilini). In *Proceedings of Xth International Symposium on Trichoptera. Potsdam, Germany. July 30th–August 5th 2000.* Mey W (ed). Goecke & Evers: Keltern; (*Deutsches Entomologisches Institut, Nova Supplementa Entomologica* **15**): 107–109.
119. Skidmore P. 1996. *A Dipterological perspective on the Holocene history of the North Atlantic area.* Unpubl. PhD thesis, University of Sheffield.
120. Johansen TJ. 1990. Infestation by *Delia fabricii* Holmgren (Diptera: Anthomyiidae) in Smooth Meadow-grass (*Poa pratensis* L.) grown for seed production in Northern Norway. *Acta Agriculturae Scandinavica* **40**: 45–51.
121. Stuiiver M, Reimer PJ, Reimer RW. 2018. CALIB 7.1 [WWW program]. <http://calib.org> [accessed 2018-11-22].
122. Martinson DG, Pisias NG, Hayes, *et al.* 1987. Age dating and the orbital Theory of the ice ages: development of a high-resolution 0 to 300,000-year chronostratigraphy. *Quaternary Research* **27**: 1–29.
123. Murton JB, Ballantyne CK. 2017. Periglacial and permafrost ground models for Great Britain. In *Engineering Geology and Geomorphology of Glaciated and Periglaciated Terrains – Engineering Group Working Party Report.* Griffiths JS, Martin CJ (eds). Geological Society, London, Engineering Group Special Publications, **28**: 501–597.
124. Atkinson TC, Briffa KR, Coope GR, *et al.* 1986. Climatic calibration of coleopteran data. In *Handbook of Holocene Palaeoecology and Palaeohydrology.* Berglund BE (ed.), John Wiley & Sons: London; 851–858.
125. Buckland PI. 2007. *The Development and Implementation of Software for Palaeoenvironmental and Palaeoclimatological Research: the Bugs Coleopteran Ecology Package (BugsCEP)* (PhD thesis). Archaeology and Environment, 23. Umeå University, Sweden.
126. Elias SA. 2010. *Advances in Quaternary Entomology.* Developments in Quaternary Sciences, 12. Elsevier: Amsterdam.
127. Jaskuła R, Soszyńska-Maj A. 2011. What do we know about winter active ground beetles (Coleoptera, Carabidae) in Central and Northern Europe? *ZooKeys* **100**: 517-532.
128. Robinson EL, Blyth E, Clark DB, *et al.* 2017. Climate hydrology and ecology research support system meteorology dataset for Great Britain (1961–2015) [CHESS-met] v1.2. NERC Environmental
- Information Data Centre. <https://doi.org/10.5285/b745e7b1-626c-4ccc-ac27-56582e77b900>
129. Lemdahl G, Coope GR. 2013. Beetle records: Late Pleistocene of Europe. In *Encyclopedia of Quaternary Science* (2nd ed.) Elias, SA (ed.). Elsevier: Amsterdam; 200–206.
130. Bell, FG, 1969. The occurrence of southern, steppe and halophyte elements in Weichselian (last-glacial) floras from southern Britain. *New Phytologist*, **68**(4): 913-922.
131. Coope GR. 1977. Fossil coleopteran assemblages as sensitive indicators of climatic changes during the Devensian (Last) cold stage. *Philosophical Transactions of the Royal Society of London* **B280**: 313–337.
132. Wolff EW, Chappellaz J, Blunier T, *et al.* 2010. Millennial-scale variability during the last glacial: The ice-core record. *Quaternary Science Reviews* **29**: 2828–2838.
133. Coope GR. 2002. Changes in the thermal climate in Northwestern Europe during Marine Oxygen Isotope Stage 3, estimated from fossil insect assemblages. *Quaternary Research* **57**: 401–408.
134. Ballantyne, CK, Harris C. 1994. *The Periglaciation of Great Britain.* Cambridge University Press: Cambridge.
135. Vandenberghe J, van der Plicht J. 2016. The age of the Hengelo interstadial revisited. *Quaternary Geochronology* **32**: 21–28.
136. Jones RL, Keen DH. 1993. *Pleistocene Environments in the British Isles.* Chapman and Hall: London.
137. Van Huissteden J, Gibbard PL, Briant RM. 2001. Periglacial fluvial systems in northwest Europe during marine isotope stages 4 and 3. *Quaternary International* **79**: 75–88.
138. Rose J, Lee JA, Kemp RA, *et al.* 2000. Palaeoclimate, sedimentation and soil development during the last Glacial Stage (Devensian), Heathrow Airport, London, UK. *Quaternary Science Reviews* **19**: 827–847.
139. West RG, Andrew R, Catt JA, *et al.* 1999. Late and Middle Pleistocene deposits at Somersham, Cambridgeshire, UK: a model for reconstructing fluvial/estuarine depositional environments. *Quaternary Science Reviews* **18**: 1247–1314.
140. Bronk Ramsey C. 2009. Bayesian analysis of radiocarbon dates. *Radiocarbon* **51**: 337-360.
141. Reimer PJ, Bard E, Bayliss A, *et al.* 2013. IntCal13 and Marine13 Radiocarbon Age Calibration Curves 0–50,000 Years cal BP. *Radiocarbon* **55**: 1869–1887.

Tables

Table 1. OSL data and ages. Ages are quoted in years from 2002 (when sampled) with 1 sigma uncertainties (for stratigraphic position, see figure 6).

Sample	Unit	Depth (m)	Cosmic (Gy/ka)	Dose Rate (Gy/ka)	Palaeodose (Gy)	Age (ka)
Fin3 (Shfd02012)	2	2.4	0.015±0.0008	1.39±0.08	26.6±0.54	19.2±1.1

15

1	Fin2	2	3	0.014±0.0007	2.70±0.14	63.1±2.3	23.4±1.5
2	(Shfd02011)						
3							
4	Fin1	4	3.5	0.013±0.0007	2.69±0.14	95.9±1.9	35.7±2.0
5	(Shfd02010)						
6							
7							

Table 2. List of plant remains from Finningley (det. O. Bennike)

Taxon	Type	Sample 2
FUNGI		
Cenococcum geophilum Fr.	sc –	140
ALGAE		
Nitella sp.	oo –	1
BRYOPHYTES		
Sphagnum sp.	le –	5
VASCULAR PLANTS		
Arctous alpina (L.) Nied	s 42	–
Betula nana L.	s 3	–
Batrachium sp.	s 4	2
Carex aquatilis Wahlenb./ C. bigelowii Torr. Ex Schwein.	s 40	–
Carex spp.	s 5	2
Comarum palustre L.	s –	1
Epilobium sp.	s –	1
Empetrum nigrum L.	s –	24
Luzula cf. spicata (L.) DC.	s –	1
Potentilla anserina L.	s –	2
Potentilla cf. crantzii (Crantz) Beck ex Fritsch	s 21	–
Ranunculus sp.	s –	1
Rumex acetosa L.	s 1	–
Rumex acetosella L.	s 34	–
Salix polaris Wahlenb.	l 1	–
Selaginella selaginoides (L.) P. Beauv ex Mart. & Schrank	ms –	9
Silene suecica (Lodd.) Greuter & Burdet	s 13	1
Stuckenia filiformis (Pers.) Börner.	s –	2
Viola palustris L.	s 1	–

The nomenclature follows <http://www.theplantlist.org/>

Sc: sclerotium, oo: oospore, le: leaf, ms: megaspore, s: seed, fruit or endocarp

1 *Arctous alpina* = *Arctostaphylos alpina*, *Silene suecica* = *Lychnis alpina*, *Stuckenia filiformis* =
2
3 *Potamogeton filiformis*
4
5
6
7
8
9
10
11
12
13
14
15
16
17
18
19
20
21
22
23
24
25
26
27
28
29
30
31
32
33
34
35
36
37
38
39
40
41
42
43
44
45
46
47
48
49
50
51
52
53
54
55
56
57
58
59
60

Table 3. Insect remains from Finningley (det. P. C. Buckland, M. Greenwood and E.

Panagiotakopulu)

	Taxon	Sample 1	Sample 2	Sample 3
7	Carabidae			
9	Pelophila borealis (Payk.)		1	
11	Notiophilus aquaticus (L.)		7	1
13	Patrobus sp.		1	
14	Bradycellus caucasicus (Chaud.)		1	
16	Amara quenseli (Schön.)		2	
18	Haliplidae			
19	Halipus sp.		1	
21	Dytiscidae			
23	Hydroporus lapponum (Gyll.)		22	
25	H. notabilis LeC.		1	
26	Hydroporus sp.		1	
28	Oreodytes alpinus (Payk.)		1	
30	Hydroporinae indet.			1
31	Agabus sturmii (Gyll.)		1	
33	Agabus labiatus (Brahm)		5	
35	Agabus sp.		1	
37	Rhantus sp.		1	
38	Colymbetes sp.		1	
40	Hydraenidae			
42	Ochthebius sp.		7	
44	Hydrophilidae			
45	Helophorus sibiricus (Mots.)		16	1
47	H. grandis Ill.		20	
48	H. aequalis Thoms.		22	
50	H. praenanus (Lom.)		1	
52	Helophorus (small) spp.		7	
54	Cercyon melanocephalus (L.)		1	
55	Cercyon spp.		2	
57	Hydrobius fuscipes agg.		1	
59	Silphidae			
60	Thanatophilus sp.		1	

Staphylinidae

1	Megarthus prosseni Schatz.	1	
2			
3	Pycnoglypta lurida (Gyll.)	14	
4			
5	Olophrum fuscum (Grav.)	10	
6			
7	O. boreale (Payk.)	2	
8			
9	Olophrum spp.	14	
10	Eucnecosum brachypterum (Grav.)	13	
11			
12	E. brunnescens (Sahl.)	1	
13			
14	Acidota quadrata (Zett.)	2	
15			
16	Lesteva longolytrata (Goeze)	7	
17			
18	Anthophagus sp.		1
19			
20	Boreaphilus henningianus Sahl.	2	
21			
22	Holoboreaphilus nordenskiöldi (Mäklin)	3	
23			
24	Aploderus caelatus (Grav.)	1	
25			
26	Anotylus nitidulus (Grav.)	1	
27			
28	Bledius arcticus Sahl.	1	
29			
30	Bledius spp.	2	
31			
32	Stenus sp.	6	
33			
34	Quedius boops (Grav.) (grp)	1	
35			
36	Quedius sp.	1	
37			
38	Mycetoporini indet.	2	
39			
40	Tachyporus sp.		1
41			
42	Tachinus elongatus Gyll.	1	
43			
44	Tachinus sp.	2	
45			
46	Aleocharinae indet.	39	1
47			
48	Indet.		1
49			
50	Byrrhidae		
51			
52	Simplocaria semistriata (F.)	1	
53			
54	S. metallica (Sturm)	1	
55			
56	Brachypteridae		
57			
58	Kateretes pedicularius (L.)	1	
59			
60	Cryptophagidae		
	Atomaria spp.	2	
	Latridiidae		
	Corticaria/Corticarina sp.	1	
	Coccinellidae		

1	Scymnus femoralis Gyll.	1	
2	Scymnus (s.l.) spp.	2	
3			
4	Ceratomegilla ulkei Crotch	2	
5			
6	Scarabaeidae		
7			
8	Aegialia sabuleti (Panz.)		1
9	Aphodius merdarius (F.)	1	
10			
11	A. borealis Gyll.		2
12			
13	A. (Chilothorax) jacobsoni Kosh.	7	
14	Aphodius spp	3	3
15			
16	Chrysomelidae		
17			
18	Cf. Oreina frigida (Weise)	1	
19			
20	Phratora sp.		1
21	Chrysomelinae indet.	2	
22			
23	Curculionidae		
24			
25	Otiorhynchus arcticus (O. Fabricius)	2	1
26			
27	O. nodosus (Müll.)	2	1
28			
29	O. rugifrons (Gyll.)		1
30			
31	Tropiphorus obtusus (Bonsd.)		1
32			
33	Notaris bimaculatus (F.)		2
34			
35	N. acridulus (L.)	1	
36			
37	N. aethiops (F.)		1
38			
39	Baris cf. artemisiae (Hbst)	1	
40			
41	Megaloptera		
42			
43	Sialidae		
44			
45	Sialis sp. (larvae)	3	
46			
47	Diptera		
48			
49	Anthomyidae		
50			
51	Delia cf. fabricii (Holm.)	1	
52			
53	Botanophila sp.	1	
54			
55	Calliphoridae		
56			
57	Indet.	1	
58			
59	Indet. (puparium)	1	
60			
	Trichoptera (see Table 4).	53	

Table 4. Trichoptera from Finningley (det. M. Greenwood)

1	Limnephilidae		
2			
3	Apataniinae	Apatania sp	1
4			Case of mineral
5			particles, in rivers,
6			brooks and lakes shores.
7			
8	Drusinae	Ecclisopteryx dalecarlica Kolen	1
9			Large streams and
10			rivers; stoney substrate.
11			Case of sand grains.
12			
13	Limnephilinae	Anabolia cf nervosa (Curtis)	4
14			Rivers, lakes and ponds
15			on sandy-silty beds.
16			Case of plant pieces or
17			of sand grains.
18			
19			
20		Grammotaulius cf	1
21			Amongst emergent
22		signatipennis McLach./ nigricornis	vegetation in temporary
23		(Retzius)	pools and ponds, also
24			rich fen. Case of plant
25			fragments.
26			
27			
28		Limnephilus algosus (McLach.)	17
29			Amongst submerged
30			vegetation in permanent
31			shallow ponds, pools
32			and lakes, slow-flowing
33			brooks and rivers. Case
34			of leaf fragments e.g.
35			Carex sp.
36			
37			
38		L.stigma Curtis	1
39			Amongst lush vegetation
40			in ponds, lakes and
41			slow-flowing rivers and
42			marshes.
43			
44		L. subcentralis Brauer	8
45			Amongst marginal
46			vegetation in pools,
47			shallow ponds, lakes,
48			rivers, streams with
49			brackish waters. Case of
50			plant fragments.
51			
52			
53		L. picturatus McLach.	8
54			
55		Limnephilus sp2 in det	15
56			
57			
58			
59			
60			

21

1
2
3
4
5
6
7
8
9
10
11
12
13
14
15
16
17
18
19
20
21
22
23
24
25
26
27
28
29
30
31
32
33
34
35
36
37
38
39
40
41
42
43
44
45
46
47
48
49
50
51
52
53
54
55
56
57
58
59
60

Molannidae

Limnephilus sp3 in det

1

Molanna albicans (Zett.)

1

In small upland lakes (Wales); in lakes in central and northern Scotland and slow-flowing rivers in central Ireland. Case of mineral grains.

Table 5. Beetle-based temperature reconstructions compared with modern climate at Finningley (1981–2010), in °C. NSPEC is the number of species in the sample with climate (MCR) data; overlap is the maximum percentage of these species with overlapping thermal envelopes, i.e. the area of climate space with maximum overlap. An ideal overlap is 100%, but large faunas or samples too coarse to resolve changing climates produce lower overlaps. Calculations using BugsCEP [89].

	TMaxLo	TMaxHi	TMinLo	TMinHi	TRangeLo	TRangeHi	NSPEC	Overlap
Sample 3	8	14	-22	-2	16	30	3	100
Sample 2	12	13	-17	-14	27	29	24	87.5
Modern	14.8	19.5	-0.8	7				

Figures

For final submissions, figures should be uploaded as separate files.

Figure and table captions

Figure 1. Map showing location of Finningley and Last Glacial Maximum features in the East Midlands and East and South Yorkshire (redrawn and modified from [39])

Figure 2. Schematic geological cross-section of Quaternary deposits in the southern part of the Vale of York projected onto a west–east line near Rossington (redrawn and modified from [30]).

Figure 3A. Map of Finningley in relation to the projected shoreline of ‘low-level’ Lake Humber (from [22]) and the Lindholme lobe of the Vale of York glacier (modified from [36]).

Figure 3B. Satellite image of the Finningley Quarry in December 2002 showing the location of the sections (Map data: Google Earth: Image @ The GeoInformation Group. Image NASA).

Figure 4. Section at the Finningley Quarry, looking south-east, September 2001. The base of the pit is in the Older River Gravel (Unit 1), the working levels in Littoral Sands and Gravels (Unit 4) and the upper level of the pit in the clay-silts of the Hemingbrough Formation (Unit 5). Photo: P. C. Buckland.

Figure 5. Vertical face through units 3–5 in the south section at the Finningley Quarry. From base upwards is diamict (unit 3, dark grey), stratified sand (unit 4), and laminated clay–silt (unit 5, grey). Symmetrical ripple forms with rounded crests and overlain with silt–clay drapes in unit 4 are interpreted as wave ripples. 2-m high pole for scale. Photo: C. Frederick.

Figure 6. Schematic stratigraphy of the Finningley Quarry. Unit 1 = Older River Gravel (s.l.) of Gaunt (1994), 2 = channel deposits with lens of organic silts, 3 = diamict, 4 = Littoral Sands and Gravels (idem) and 5 = Lake Humber clay–silts (Hemingbrough Formation).

Figure 7. Vertical section through unit 2 at the Finningley Quarry: channel deposits with the contorted organic silt sampled for plant and invertebrate remains towards base. Photo: P. C. Buckland.

Figure 8. Ice-wedge pseudomorph within sand and gravel (unit 1). The wedge is >1.45 m high and has a maximum true width (orthogonal to the axial plane of the wedge) of 28 cm. Note smooth downturning of host strata adjacent to the wedge and vertically aligned elongate pebbles within sandy gravel infill of the wedge. The top of the wedge is overlain by a continuous layer of grey sand and gravel at the top of the trowel, indicating that the wedge is intraformational within unit 1. October 2001. Photo: J. Murton.

Figure 9. Diamict of unit 3. Involved lower contact of diamict shows flame structures (indicated by arrows), inclined in various directions, of underlying sand and gravel extending up for as much as 0.5 m into the diamict. Diamict is texturally heterogeneous, with pockets of sand and dark grey silt–clay, and some clayey bands. Trowel for scale. August 2007. Photo: J. Murton.

Figure 10. (A) Sand and varved silty clay of unit 4 overlying diamict of unit 3. Note mud drape (dark grey) directly overlying diamict and flaser bedding within the overlying sand. Above this is a prominent layer 16–20 cm thick of varved silty clay. Ripple form sets are visible in the top of the overlying sand. October 2001. (B) Symmetrical ripple forms (wavy bedding) in sand of unit 4. Some ripples have peaked crests, some rounded crests, with both representing wave ripples. Photo: J. Murton, August 2007.

Figure 11 Soft-sediment deformation structures in sand and silty clay of unit 2. (A) Sand dykes and sills in varved silty clay, beneath prominent sediment-injection / dewatering structures in overlying sand. Trowel for scale. (B) sediment-injection / dewatering structures. The structure on the left has a flat toe below scattered blocks of grey silty clay. Note smooth downturned strata on left side of structure and normal step faults on right side. Grey strata of silty clay extend across the top of the structure. Trowel for scale. (C) Load casts and sand diapirs. Coin and spade handle for scale. (D) Dish structures formed by water escape. All photographs J. Murton, October 2001.

Figure 12. Beetle habitat trait-based environmental reconstruction using A) taxon occurrence only and B) species level abundance weighted data. A) provides a general reconstruction, giving equal weight to each taxon, based on the habitat preferences at all taxonomic resolutions; B) uses only taxa identified to species and weights the reconstruction according to the number of individuals found, thus providing a more constrained reconstruction for the most reliable habitat indications whilst ignoring evidence from poorly resolved identifications. Neither reconstruction is 'correct', but together they provide a tool for interpreting the environmental implications of the fauna. Note that a taxon may be included in more than one habitat class. MNI = Minimum Number of Individuals. (See [125] for classification and methodology, [89] for classification data).

Figure 13. *Limnephilus algeus*, showing the frontoclypeal apotome from both modern and fossil material of this species. Photo: Dept. of Geography, Loughborough University

Figure 14: Fossil specimens of *Limnephilus algeus* and map of modern distribution. Photo: Dept. of Geography, Loughborough University.

Figure 15. OSL Palaeodose (De) replicate measurements for samples Finn 1-3 (for location see figure 6) showing a high degree of convergence and low overdispersion (OD).

Figure 16. Beetle-based MCR reconstruction of the Finningley samples compared with other radiocarbon dated MIS 3 sites mentioned in the text. Modern Finningley climate is shown for comparison (leftmost bar and horizontal shaded areas). Samples with <3 MCR species have been omitted but samples with poor maximum species overlaps (<80%) have been included to provide more samples for comparison. The lower bar chart shows the number of species used in each reconstruction. Sample names are as stored in the BugsCEP database (Buckland & Buckland 2006) [89]. Site ages, in calibrated ka BP, are given for either the range of dated insect samples or the overall site where these are not directly dated (see supplemental data). The figure therefore does not represent a contiguous sequence. Dates calibrated with OxCal [140] using the IntCal 13 curve [141]. For sources and primary data, see Supplementary Table 1.

Table 1. OSL data and ages. Ages are quoted in years from 2002 (when sampled) with 1 sigma uncertainties (for stratigraphic position, see figure 6).

Table 2. List of plant remains from Finningley (det. O. Bennike).

Table 3. Insect remains from Finningley (det. P. C. Buckland, M. Greenwood and E. Panagiotakopulu).

Table 4: Trichoptera from Finningley (det. M. Greenwood).

Table 5. Beetle-based temperature reconstructions compared with modern climate at Finningley (1981–2010), in °C.

NSPEC is the number of species in the sample with climate (MCR) data; overlap is the maximum percentage of these species with overlapping thermal envelopes, i.e. the area of climate space with maximum overlap. An ideal overlap is 100%, but large faunas or samples too coarse to resolve changing climates produce lower overlaps. Calculations using BugsCEP [89].

Supplemental Figures

Supplementary Figure 1. Vertical face through units 3–5 in an east-west section at the Finningley Quarry. From base upwards is diamict (unit 3, dark grey), stratified sand (unit 4, orange–brown), and laminated clay–silt (unit 5, grey). The dark grey layer ~20 cm thick within the sand of unit 3 is a subunit of laminated clay–silt. Above this subunit, the sand is cross-bedded, with foresets showing a gentle apparent dip to the east. (ranging pole marked in 0.5m lengths. Photo: C. Frederick 2001).

Supplementary Figure 2. Finningley Quarry: W Section.

Vertical face through units 1 and 3–5 in south-north section at Finningley Quarry. From base upwards is sand and gravel (unit 1, orange–brown), diamict (unit 3, dark grey), stratified sand (unit 4, light grey), and laminated clay–silt (unit 5, dark grey). Height of face is ~3–4 m.

1
2
3
4
5
6
7
8
9
10
11
12
13
14
15
16
17
18
19
20
21
22
23
24
25
26
27
28
29
30
31
32
33
34
35
36
37
38
39
40
41
42
43
44
45
46
47
48
49
50
51
52
53
54
55
56
57
58
59
60

Figure 1. Map showing location of Finningley and Last Glacial Maximum features in the East Midlands and East and South Yorkshire (redrawn and modified from [39])

Figure 2. Schematic geological cross-section of Quaternary deposits in the southern part of the Vale of York projected onto a west-east line near Rossington (redrawn and modified from [30]).

160x113mm (600 x 600 DPI)

Figure 3A. Map of Finningley in relation to the projected shoreline of 'low-level' Lake Humber (from [22]) and the Lindholme lobe of the Vale of York glacier (modified from [36]).

Figure 3B. Satellite image of the Finningley Quarry in December 2002 showing the location of the sections (Map data: Google Earth: Image @ The GeoInformation Group. Image NASA).

248x137mm (300 x 300 DPI)

1
2
3
4
5
6
7
8
9
10
11
12
13
14
15
16
17
18
19
20
21
22
23
24
25
26
27
28
29
30
31
32
33
34
35
36
37
38
39
40
41
42
43
44
45
46
47
48
49
50
51
52
53
54
55
56
57
58
59
60

Figure 4. Section at the Finningley Quarry, looking south-east, September 2001. The base of the pit is in the Older River Gravel (Unit 1), the working levels in Littoral Sands and Gravels (Unit 4) and the upper level of the pit in the clay-silts of the Hemingbrough Formation (Unit 5). Photo: P. C. Buckland.

33x22mm (600 x 600 DPI)

Figure 5. Vertical face through units 3–5 in the south section at the Finningley Quarry. From base upwards is diamict (unit 3, dark grey), stratified sand (unit 4), and laminated clay–silt (unit 5, grey). Symmetrical ripple forms with rounded crests and overlain with silt–clay drapes in unit 4 are interpreted as wave ripples. 2-m high pole for scale. Photo: C. Frederick.

Figure 6. Schematic stratigraphy of the Finningley Quarry. Unit 1 = Older River Gravel (s.l.) of Gaunt (1994), 2 = channel deposits with lens of organic silts, 3 = diamict, 4 = Littoral Sands and Gravels (idem) and 5 = Lake Humber clay-silts (Hemingbrough Formation).

Figure 7. Vertical section through unit 2 at the Finningley Quarry: channel deposits with the contorted organic silt sampled for plant and invertebrate remains towards base. Photo: P. C. Buckland.

31x22mm (600 x 600 DPI)

Figure 8. Ice-wedge pseudomorph within sand and gravel (unit 1). The wedge is >1.45 m high and has a maximum true width (orthogonal to the axial plane of the wedge) of 28 cm. Note smooth downturning of host strata adjacent to the wedge and vertically aligned elongate pebbles within sandy gravel infill of the wedge. The top of the wedge is overlain by a continuous layer of grey sand and gravel at the top of the trowel, indicating that the wedge is intraformational within unit 1. October 2001. Photo: J. Murton.

Figure 9. Diamict of unit 3 Involved lower contact of diamict shows flame structures (indicated by arrows), inclined in various directions, of underlying sand and gravel extending up for as much as 0.5 m into the diamict. Diamict is texturally heterogeneous, with pockets of sand and dark grey silt-clay, and some clayey bands. Trowel for scale. August 2007. Photo: J. Murton.

Figure 10. (A) Sand and varved silty clay of unit 4 overlying diamict of unit 3. Note mud drape (dark grey) directly overlying diamict and flaser bedding within the overlying sand. Above this is a prominent layer 16–20 cm thick of varved silty clay. Ripple form sets are visible in the top of the overlying sand. October 2001. (B) Symmetrical ripple forms (wavy bedding) in sand of unit 4. Some ripples have peaked crests, some rounded crests, with both representing wave ripples. Photo: J. Murton, August 2007.

152x209mm (220 x 220 DPI)

Figure 11 Soft-sediment deformation structures in sand and silty clay of unit 2. (A) Sand dykes and sills in varved silty clay, beneath prominent sediment-injection / dewatering structures in overlying sand. Trowel for scale. (B) sediment-injection / dewatering structures. The structure on the left has a flat toe below scattered blocks of grey silty clay. Note smooth downturned strata on left side of structure and normal step faults on right side. Grey strata of silty clay extend across the top of the structure. Trowel for scale. (C) Load casts and sand diapirs. Coin and spade handle for scale. (D) Dish structures formed by water escape. All photographs J. Murton, October 2001.

107x246mm (220 x 220 DPI)

Figure 12. Beetle habitat trait-based environmental reconstruction using A) taxon occurrence only and B) species level abundance weighted data. A) provides a general reconstruction, giving equal weight to each taxon, based on the habitat preferences at all taxonomic resolutions; B) uses only taxa identified to species and weights the reconstruction according to the number of individuals found, thus providing a more constrained reconstruction for the most reliable habitat indications whilst ignoring evidence from poorly resolved identifications. Neither reconstruction is 'correct', but together they provide a tool for interpreting the environmental implications of the fauna. Note that a taxon may be included in more than one habitat class. MNI = Minimum Number of Individuals. (See [125] for classification and methodology, [89] for classification data).

330x156mm (299 x 299 DPI)

Figure 13. *Limnephilus algosus*, showing the frontoclypeal apotome from both modern and fossil material of this species. Photo: Dept. of Geography, Loughborough University

113x148mm (220 x 220 DPI)

1
2
3
4
5
6
7
8
9
10
11
12
13
14
15
16
17
18
19
20
21
22
23
24
25
26
27
28
29
30
31
32
33
34
35
36
37
38
39
40
41
42
43
44
45
46
47
48
49
50
51
52
53
54
55
56
57
58
59
60

Figure 14: Fossil specimens of *Limnephilus algosus* and map of modern distribution. Photo: Dept. of Geography, Loughborough University.

118x151mm (220 x 220 DPI)

Figure 15. OSL Palaeodose (D_e) replicate measurements for samples Finn 1-3 (for location see Figure 6) showing a high degree of convergence and low overdispersion (OD).

Figure 16. Beetle-based MCR reconstruction of the Finningley samples compared with other radiocarbon dated MIS 3 sites mentioned in the text. Modern Finningley climate is shown for comparison (leftmost bar and horizontal shaded areas). Samples with <3 MCR species have been omitted but samples with poor maximum species overlaps (<80%) have been included to provide more samples for comparison. The lower bar chart shows the number of species used in each reconstruction. Sample names are as stored in the BugsCEP database (Buckland & Buckland 2006) [89]. Site ages, in calibrated ka BP, are given for either the range of dated insect samples or the overall site where these are not directly dated (see supplemental data). The figure therefore does not represent a contiguous sequence. Dates calibrated with OxCal [140] using the IntCal 13 curve [141]. For sources and primary data, see Supplementary Table 1.

242x216mm (300 x 300 DPI)

Appendix B

Dear Editors and reviewers,

Please find below our collective response to your very helpful comments and suggestions. Where a response is not provided below, it is included in the commented version of the revised manuscript Word file.

Changes to the manuscript have been documented using track changes, and both commented (review file) and 'clean' (main document) versions have been uploaded.

Only revised or additional figures and tables have been uploaded a second time.

In response to reviewer 2, we have uploaded an additional supplementary figure.

We have also corrected some minor details in supplementary tables 1 and 2.

Kind regards,

Philip Buckland and colleagues

Reviewer comments to Author:

Reviewer: 1

This is basically a good paper that adds significantly to the corpus of data concerning the age and environment of deposition of the Older River Gravels, which are an important unit in the Middle to Late Pleistocene regional development of GB. My comments are all very minor and perhaps more food for thought. They are annotated on the attached manuscript.

Response to reviewer 1:

Thank you for your kind remarks and please see revisions in the annotated version of the manuscript. We have attempted to accommodate most of your suggestions.

Reviewer: 2

Comments by Dr Ian S. Evans, Durham University, on Buckland et al., Mid to Late Devensian landscape change in England.

A thorough analysis of stratigraphic and palaeontological/palaeoecological observations from this important site in eastern England is provided by a group of experienced scientists. Observations over a number of years as the quarry face progressed improve the reliability of interpretations. The numerous illustrations are well drawn and relevant: the photos are well selected.

The section 6.1 Discussion usefully compares the temperature ranges (summer and winter) implied at Finningley with those from other MIS 3 sites. Using the Greenland record, the authors correctly state that interstadials and stadials alternated within MIS 3, yet it is difficult to follow the implications of the various dates and temperatures in the text which (despite the conclusion) reads as if there is some expectation of uniform conditions throughout MIS 3. The dates tabulated below Fig. 16 do not put things into chronological order.

Authors' response: The sites are in order of youngest date, which is unfortunately as good a chronological order as the sample data permits (but see below)...

Section 6.1 would be aided by a summary temperature: chronology Figure plotting each observation as a box, with its range of calibrated dates and its range of Tmin and (separately) Tmax. (There are 13 date ranges mentioned in the text, i.e. 13 boxes.) The text could then be shortened and clarified.

Authors' response: This is something we have discussed back and forth repeatedly. Whilst we agree that this would be useful, the data are simply not yet good enough to allow the MCR results for individual samples to be placed at their correct point in a timeline (hence the use of site date ranges in figure 16). We have compromised by including an additional supplemental figure 3 which shows the site date ranges (calibrated 14C only) compared with the Greenland Ice core data.

In terms of adding MCR data to the new figure, the entomologists amongst us are adamant that this should not be done due to the limited and variable reliability of the dates for individual samples. Leaving out poorly/un-dated samples would give a misleading impression of the variability of climates at each site. Inserting these samples into a figure with the Greenland curve would give a false indication of accuracy, and just using the min-max values for Tmax/Tmin would remove indications of less variable climates. Leaving off Tmin simplifies, but removes the continentality (difference Tmax-Tmin) aspect of the story, which is very interesting and something for a later paper.

I think the title could be improved: it does not cover England and there is little on Late Devensian. I suggest 'Mid Devensian climate and landscape in eastern England' would be more accurate as the first sentence of the title.

Authors' response: A compromise is suggested, as we feel the results do have implications for a wider geographical region (see commented text).

Readers are expected to be familiar with detailed terminology in several fields: sedimentology, botany and zoology. Possibly terms such as 'flaser' and 'stenotherm' might be defined.

- Flaser clarified on first use

- 'stenotherm' clarified

Below I suggest a number of corrections and some cases where the expression can be polished. There is a tendency to use too many commas, making complicated sentences with too many qualifications:

- Acknowledged, and we have attempted to follow the reviewer's suggestion to help improve readability

some could be split, e.g.

lines 18-22

- split sentence

and 30-33 on page 6.

- split sentence

DETAILS:

Page 1: Line 39 'seek to' is anthropomorphic. 'data can ...'?

- Corrected

52 drop one of the 2 boths

- Corrected

55 perhaps 'Britain' (or even 'England') rather than UK?

- Our use of the UK as a geographical boundary is intentional. We do not cover Ireland, but refer to England, Scotland and include references on Wales.

Page 2: Line 1 'showed'

- Corrected

2 'the limit'

- Not corrected.

18 'setting, in'

- Corrected.

34 'deep-water' rather than profundal

- included both, to promote the continued use of scientific terminology

41 'from both'

- Corrected.

43 delete comma

- sentence changed

Page 3: 13 'were collected'

- sentence changed

13-14 delete present commas: insert comma before 'and'.

- sentence changed

14 'was revealed'

- Corrected.

18 'was placed'

- Corrected.

20 delete second 'then'

- Corrected.

25 'that for'

- adjusted

29 Should Stratigraphy be section 5.1? – and later sections renumbered.

- Yes, corrected.

31 delete first comma

- not corrected

49 delete first comma

- not corrected

Page 4: 21 'which are interpreted'

- corrected

Page 5: 2 'river Idle. It is interpreted as being'

- sentence restructured to avoid confusion

11 'non-preservation' !

- added 'poor'

48 'whereas others indicate'

- Corrected.

Page 6: 5.3.1 is italicised, unlike any later subsection headings...

- Corrected. Had issues with the template!

2 ': ' rather than ';

- I disagree! But have split the sentence anyway.

6 'appear'

- Corrected.

16 One comma too many! Delete last, '[82] and occurring'.

- Corrected.

26 'suggested'

- Corrected.

41 Give name before [99] ?

- Corrected.

Page 7: 28 remove comma. Split the following sentence?

- moved comma and split following sentence for clarity.

30 'it is found'

- sentence rewritten

39 delete last comma

- not corrected

39-40 "in papers published in" makes no sense, as the three following citations are normal papers, not collections.

- corrected

40 replace ';' with ','

- corrected

44 'BP at Clettnadal'

- corrected

54 'arctic-equivalent'

- corrected

Page 8: 13 'between ... and'

- corrected

58 'glance a similarly'

- corrected

Page 9: 11 6. Not 6.6.

- corrected

16-17 past tense, especially for 2001 !

- corrected, sad but true.

56 summer-winter-summer?? Rewrite sentence.

- rewritten

Page 10: 1 The sentence ending 'Queensford.' is a non-sentence. It should be combined with the following, or re-written.

- rewritten (removed the initial 'Whilst')

8: Replace second ';' with 'and'.

- removed first comma and replaced second with comma, as 'and' would change the referencing

This is another clumsy sentence.

- split the sentence for readability

24: 'near continent'

- corrected

39/40 ';' or ',' ?

- corrected '!

Page 11: top – excess '11'

- page number in template, I dare not touch it...

11 'suggests that around'

- corrected

14 'and 11'

- corrected

Page 14: 26-27 Why is 'Published for the Quaternary Research Association' stated here? It is not stated elsewhere JQS is cited. Also, italicise journal title.

- corrected